# Finding Stationary Points by Comparisons

**Helin Wang** [* 1]   **Chenyi Zhang** [* 2]   **Xiwen Tao** [1]   **Yexin Zhang** [3 4]   **Tongyang Li** [3 4]

## Abstract

We study the problem of finding stationary points of non-convex functions when access to the objective is provided only through a comparison oracle that, given two points, outputs which has the larger function value. For a twice differentiable $f\colon \mathbb{R}^n \to \mathbb{R}$ with Lipschitz gradient and Hessian, we develop an algorithm that outputs an $\epsilon$-stationary point using $\widetilde{O}(n^2/\epsilon^{1.5})$ queries. Our approach uses a subroutine that estimates the normalized Hessian to accuracy $\delta$ using $\widetilde{O}(n^2 \log(1/\delta))$ queries. We further study this problem with a quantum comparison oracle model where queries can be made in superpositions, and develop the first quantum algorithm that finds an $\epsilon$-stationary point, which takes $\widetilde{O}(n/\epsilon^{1.5})$ queries.

## 1. Introduction

We study the problem of finding a stationary point of a non-convex function $f\colon \mathbb{R}^n \to \mathbb{R}$, which is a point $\mathbf{x} \in \mathbb{R}^n$ satisfying $\|\nabla f(\mathbf{x})\| \leq \epsilon$, given an initial $\mathbf{x}^{(0)} \in \mathbb{R}^d$ with bounded initial function value gap, i.e., $f(\mathbf{x}^{(0)}) - \inf_{x \in \mathbb{R}^n} f(\mathbf{x}) \leq \Delta$. Finding a stationary point is a natural objective in non-convex optimization, since computing the global optimum is NP hard in the worst case. Moreover, in problems including tensor decomposition (Ge & Ma, 2017), matrix completion (Ge et al., 2016), and regression with non-convex regularization (Loh & Wainwright, 2015), global optimality can be obtained by finding a second-order stationary point, which is a generalization of stationary points.

As a foundational problem in optimization theory, finding a stationary point, also known as critical point computation (Agarwal et al., 2017; Adil et al., 2025) or making the gradient small (Allen-Zhu, 2018), has been extensively

---

[*]Equal contribution  [1]School of Electronics Engineering and Computer Science, Peking University [2]Computer Science Department, Stanford University [3]Center on Frontiers of Computing Studies, Peking University [4]School of Computer Science, Peking University. Correspondence to: Tongyang Li <tongyangli@pku.edu.cn>.

*Proceedings of the 43rd International Conference on Machine Learning*, Seoul, South Korea. PMLR 306, 2026. Copyright 2026 by the author(s).

studied. Given a gradient oracle, gradient descent finds an $\epsilon$-stationary point of $f$ using $O(1/\epsilon^2)$ iterations, provided that $f$ has Lipschitz gradient, i.e., $\|\nabla f(\mathbf{x}) - \nabla f(\mathbf{y})\| \leq L_1 \|\mathbf{x} - \mathbf{y}\|$ for any $\mathbf{x}, \mathbf{y} \in \mathbb{R}^d$ (Nesterov, 2013). More generally, if $f$ has $L_p$-Lipschitz $p$-th order derivative, Birgin et al. (2017) gives an algorithm using $O(L_p^{1/p} \Delta \epsilon^{-(p+1)/p})$ queries to a $p$-th order oracle, where a query at $\mathbf{x}$ returns all derivatives of $f$ at $\mathbf{x}$ up to order $p$. Furthermore, Carmon et al. (2020) establishes that these rates are optimal among dimension-independent algorithms. Following the rapid development of computing, the problem of finding stationary points has also been investigated in the setting of quantum computing, where both quantum algorithms (Zhang et al., 2021; Liu et al., 2023; Leng et al., 2023; Chen et al., 2025) and quantum lower bounds (Zhang & Li, 2023; Gong et al., 2025) were established.

More recently, training of machine learning models solicits for even simpler information. For example, it is known that taking only signs of gradient descents still demonstrates good performance in training neural networks (Liu et al., 2019; Li et al., 2023; Bernstein et al., 2018). Moreover, in the breakthrough of large language models (LLMs), reinforcement learning from human feedback (RLHF) (Christiano et al., 2017; Gaur et al., 2024; Fan et al., 2025) played an important rule in training these LLMs, for instance on GPT-3 (Ouyang et al., 2022). Compared to standard RL that applies function evaluation for rewards, RLHF is preference-based RL that only compares between options and determines which is better. There is emerging interest in preference-based RL, with a series of results (Chen et al., 2022; Saha et al., 2023; Novoseller et al., 2020; Xu et al., 2020; Zhu et al., 2023; Tang et al., 2023) establishing provable guarantees for learning a near-optimal policy from preference feedback. Furthermore, Wang et al. (2023) proved that preference-based RL can be solved with small or no extra costs compared to those of standard reward-based RL for a wide range of models.

There has been a line of research on solving optimization problems using a comparison oracle; see the survey by Larson et al. 2019. Formally, a function $f\colon \mathbb{R}^n \to \mathbb{R}$ is accessed through a comparison oracle $O_f^{\mathrm{comp}}\colon \mathbb{R}^n \times \mathbb{R}^n \to \{-1, 1\}$

that upon a pair of inputs $(\mathbf{x}, \mathbf{y}) \in \mathbb{R}^n \times \mathbb{R}^n$, we get output:

$$O_f^{\text{comp}}(\mathbf{x}, \mathbf{y}) = \begin{cases} 1, & f(\mathbf{x}) \geq f(\mathbf{y}) \\ -1. & f(\mathbf{x}) \leq f(\mathbf{y}) \end{cases} \quad (1)$$

with either output allowed when $f(\mathbf{x}) = f(\mathbf{y})$. Classical direct-search and pattern-search methods use such comparisons to accept or reject trial steps (Kolda et al., 2003; Audet & Dennis Jr, 2006), while the Nelder–Mead method (Nelder & Mead, 1965) relies on comparisons among candidate points to drive the search. However, Nelder–Mead may converge to a nonstationary point even on continuously differentiable convex examples (McKinnon, 1998). Based on that, Jamieson et al. (2012) studied derivative-free optimization with Boolean comparison feedback and comparison-based line searches. GradientLess Descent (Golovin et al., 2020) studies monotone-invariant zeroth-order optimization, and Tang et al. (2023) studied ranking-based feedback, where the oracle provides ranking information over multiple candidate points. Beyond noiseless pairwise comparisons in Euclidean space, Saha et al. (2024) studied batched and multiway preference feedback, and Ren et al. (2026) considered comparison-based optimization on Riemannian manifolds. Very recently, Scheinberg & Xiong (2026) proposed a function-free optimization framework in which the preference relation itself, rather than an underlying scalar objective, defines the optimization problem.

For convex objectives, several works give provable guarantees under comparison or order-type feedback. Karabag et al. (2021) proposed an ellipsoid-based method for smooth convex optimization using comparison oracles. Bergou et al. (2020); Gorbunov et al. (2020) developed the stochastic three-point methods using comparisons among randomly sampled trial points to select the next iterate. Dueling optimization (Saha et al., 2021; 2025) studies convex optimization from noisy pairwise comparisons, and Lobanov et al. (2024) developed accelerated methods in a similar setting.

For nonconvex objectives, fewer comparison-query guarantees are known. The stochastic three-point method of Bergou et al. (2020); Gorbunov et al. (2020) samples random search directions and chooses the next iterate using comparisons among trial points; for functions with Lipschitz gradients, this gives an $O(n/\epsilon^2)$ comparison-query implementation for finding an $\epsilon$-stationary point in expectation. These results leave open whether comparison access can exploit higher-order smoothness to improve the dependence on $\epsilon$ for finding stationary points.

Quantum algorithms for continuous optimization have been extensively studied, including semidefinite programming and linear programming (Brandão & Svore, 2017; Brandão et al., 2019; Augustino et al., 2023; Kerenidis & Prakash, 2018; van Apeldoorn & Gilyén, 2019; van Apeldoorn et al., 2020b), convex optimization (Chakrabarti et al., 2020; van

Apeldoorn et al., 2020a; Chakrabarti et al., 2023; Sidford & Zhang, 2023; Zhang et al., 2024; Augustino et al., 2025; Marsden et al., 2026), and nonconvex optimization (Zhang et al., 2021; Childs et al., 2022; Gong et al., 2025; Liu et al., 2023; Leng et al., 2023). Complementary lower bounds have also been obtained that establish limitations of quantum speedups for convex and nonconvex optimization in several oracle models (Garg et al., 2020; 2021; Zhang & Li, 2023).

**Our results.** In this paper, we develop an algorithm that makes $\widetilde{O}(n^2/\epsilon^{1.5})$ queries to $O_f^{\text{comp}}$ and guarantees that one of the queried points is an $\epsilon$-stationary point.[1]

**Theorem 1.1** (Informal). *Let $f \colon \mathbb{R}^n \to \mathbb{R}$ have $L_1$-Lipschitz gradient and $L_2$-Lipschitz Hessian. Given $\mathbf{x}_0 \in \mathbb{R}^n$ satisfying $f(\mathbf{x}_0) - \inf_{x \in \mathbb{R}^n}(\mathbf{x}) \leq \Delta$, with success probability at least $2/3$, Algorithm 5 visits an $\epsilon$-stationary point using $\widetilde{O}(\Delta\sqrt{L_2}n^2/\epsilon^{1.5})$ queries to $O_f^{\text{comp}}$.*

Up to poly-logarithmic factors, the $\epsilon$-dependence of our rate matches the optimal rate of second-order methods (Nesterov & Polyak, 2006; Carmon et al., 2020). Compared to Bergou et al. (2020); Gorbunov et al. (2020), our algorithm achieves an improved dependence on $\epsilon$, at the cost of a worse dependence on the dimension $n$.

It is unclear whether our bound, particularly the $n^2$ dependence, is asymptotically optimal in any non-trivial regime of $n$ and $\epsilon$. Although Carmon et al. (2020) give relevant lower bounds in the dimension-independent setting, the dimension-dependent complexity of finding stationary points is still not well understood, even under gradient- and Hessian-oracle access. We view developing matching lower bounds in the comparison setting as an interesting open problem.

Note that here we only obtain the guarantee that our algorithm visits an $\epsilon$-stationary point throughout the iterations instead of finding one, similar to prior works (Bergou et al., 2020; Gorbunov et al., 2020). This is due to the fact that in the comparison oracle model the algorithm only observes relative function values. Consequently, it is in general impossible to access the gradient norm or even to test whether a given point $\mathbf{x}$ is an $\epsilon$-stationary point. For applications where a single point is needed, a natural heuristic is to return the iterate with the smallest function value, which can be identified by comparisons. While this does not certify stationarity in general, in many structured nonconvex problems near-optimality is closely tied to stationarity or second-order stationarity, such as tensor decomposition (Ge & Ma, 2015), matrix completion (Ge et al., 2016), and regression with nonconvex regularization (Loh & Wainwright, 2015).

Furthermore, we study the problem of finding stationary points in the quantum setting, where we can query a quan-

---

[1]We use $\widetilde{O}(\cdot)$ to hide poly-logarithmic factors in $n$, $\epsilon^{-1}$, $L_1$, $L_2$, and $\Delta$.

tum comparison oracle

$$O_{f,q}^{\text{comp}}|\mathbf{x}\rangle|\mathbf{y}\rangle|b\rangle = |\mathbf{x}\rangle|\mathbf{y}\rangle|b \oplus \mathbb{1}\{f(\mathbf{x}) > f(\mathbf{y})\}\rangle \quad (2)$$

which performs comparison in quantum superpositions.[2] Extending upon our classical algorithm, we obtain *the first quantum algorithm for finding stationary points using a quantum comparison oracle.*

**Theorem 1.2** (Informal). *Let $f\colon \mathbb{R}^n \rightarrow \mathbb{R}$ have $L_1$-Lipschitz gradient and $L_2$-Lipschitz Hessian. Given $\mathbf{x}_0 \in \mathbb{R}^n$ satisfying $f(\mathbf{x}_0) - \inf_{x \in \mathbb{R}^n}(\mathbf{x}) \leq \Delta$, there exists a quantum algorithm that visits an $\epsilon$-stationary point using $\widetilde{O}(\Delta\sqrt{L_2}n/\epsilon^{1.5})$ queries to $O_{f,q}^{\text{comp}}$.*

**Techniques.** Our approach for obtaining Theorem 1.1 is summarized in Figure 1. Our techniques build upon Tao et al. (2026), which provides an algorithm for estimating gradient directions using only comparisons.

We begin by giving an algorithm ComparisonHessVec (Algorithm 1) that, for any given $\mathbf{x}, \mathbf{y} \in \mathbb{R}^d$, estimates the direction $\nabla^2 f(\mathbf{x}) \cdot \mathbf{y}$. This is done by first estimating the directions of $\nabla f(\mathbf{x})$, $\nabla f(\mathbf{x} + r\mathbf{y})$, and $\nabla f(\mathbf{x} - r\mathbf{y})$ using Tao et al. (2026). We then infer the direction of $\nabla f(\mathbf{x} + r\mathbf{y}) - \nabla f(\mathbf{x} - r\mathbf{y})$ using a geometric property: its intersection with $\nabla f(\mathbf{x})$ and $\nabla f(\mathbf{x} + r\mathbf{y})$ as well as its intersection with $\nabla f(\mathbf{x})$ and $\nabla f(\mathbf{x} - r\mathbf{y})$ give two segments of same length (see Figure 2). By choosing $r$ sufficiently small, this direction provides a good approximation of the direction of $\nabla^2 f(\mathbf{x}) \cdot \mathbf{y}$.

Then, we use ComparisonHessVec to implement an algorithm, ComparisonHE (Algorithm 2), that estimates the normalized Hessian $\nabla^2 f(\mathbf{x})$. Because a column of $\nabla^2 f(\mathbf{x})$ is given by applying the Hessian to a unit vector, we can use ComparisonHessVec to estimate the normalized direction of each column. The remaining challenge is to determine the relative norms of different columns. For any two columns $\mathbf{h}_i$ and $\mathbf{h}_j$ of $\nabla^2 f(\mathbf{x})$, this is accomplished by estimating the directions of $\mathbf{h}_i$, $\mathbf{h}_j$, and $\mathbf{h}_i + \mathbf{h}_j$, and then solving the resulting triangle. A corner case of the above argument arises when two columns $\mathbf{h}_i$ and $\mathbf{h}_j$ have nearly identical directions, in which case the error incurred by solving the triangle can be arbitrarily large. To address this issue, we introduce an additional perturbation along a direction sufficiently far from that of $\mathbf{h}_i$, which ensures that the resulting error remains controllable.

Furthermore, we give an algorithm ComparisonRatio (Algorithm 4) that estimates the ratio between the norm of the Hessian and the norm of the gradient using comparisons. This is done by identifying a vector $\mathbf{v}$ of the approximate Hessian whose corresponding eigenvalue has the largest magnitude, estimating the directions of $\nabla f(\mathbf{x})$ and $\nabla f(\mathbf{x} +$

$\mu\mathbf{v})$, and then solving the resulting triangle. One source of error in this approximation is that $\mathbf{v}$ may differ from $\mathbf{u}$, the eigenvector of the true Hessian $\nabla^2 f(\mathbf{x})$ corresponding to the eigenvalue with the largest magnitude. We control this discrepancy by applying the Davis–Kahan theorem to bound $\|\mathbf{u} - \mathbf{v}\|$, which yields a bound on the overall estimation error.

Finally, we design a trust-region–type algorithm (Algorithm 5) such that, at any iteration $\mathbf{x}_t$ which is not an $\epsilon$-stationary point, it produces a new point $\mathbf{x}_{t+1}$ that decreases the function value by $\Omega(\epsilon^{3/2})$. As shown in standard trust-region analysis, there exists a point $\hat{\mathbf{x}} \in \mathbb{R}^n$ with $\|\hat{\mathbf{x}} - \mathbf{x}_t\| = O(\sqrt{\epsilon})$ such that $f(\hat{\mathbf{x}}) - f(\mathbf{x}_t) \leq -\Omega(\epsilon^{3/2})$. Moreover, such a point can be found exactly given access to the normalized gradient, the normalized Hessian, and the ratio of their norms. We show that the approximate versions of these quantities produced by our previous algorithms are sufficiently accurate to recover a point $\hat{\mathbf{x}}'$ with the same asymptotic decrease in function value. Hence, after running this algorithm for $\Theta(1/\epsilon^{1.5})$ iterations, it decreases the function value for at least $\Omega(\Delta)$ in expectation, thus we get a contradiction with a constant probability. This contradiction means that we have visited at least one stationary point during the iteration steps. The above argument can fail in certain corner cases, e.g., when the Hessian is rank-one, positive semidefinite, or negative semidefinite, or when the eigenvector $v$ corresponding to the eigenvalue of largest magnitude is nearly aligned with the gradient. In these cases, we additionally apply normalized gradient descent together with a binary line search to guarantee an $\Omega(\epsilon^{-1.5})$ decrease in function value.

**Open questions.** Our work leaves several open questions for future investigation.

- First, it would be interesting to determine whether one can interpolate between our approach and the STP method (Bergou et al., 2020; Gorbunov et al., 2020) to obtain improved dependence on $n$ and $\epsilon$.

- Second, it is natural to extend our setting to stochastic comparison oracles, such as the standard model by Bradley & Terry (1952).

- Third, it would be valuable to study the application of our algorithm to practical machine learning scenarios, such as reinforcement learning from human feedback (RLHF).

- Fourth, it is promising to combine our approach and lazy-type method (Doikov & Grapiglia, 2025; Liu et al., 2026). We have known that lazy trust region method can reduce the $n$-dependence to $\tilde{O}(n^{1.5})$ in zeroth-order oracle setting, and it is natural to extend it to comparison setting.

---

[2]Quantum notation can be found in Section 2.

*Figure 1.* The structure of our stationary point finding algorithm. In this figure the arrow represents the implementation relationship. The three purple frames represent the novel techniques that we propose, the yellow frame represents the known technique from Tao et al. (2026), and the green frame represents our main theorem.

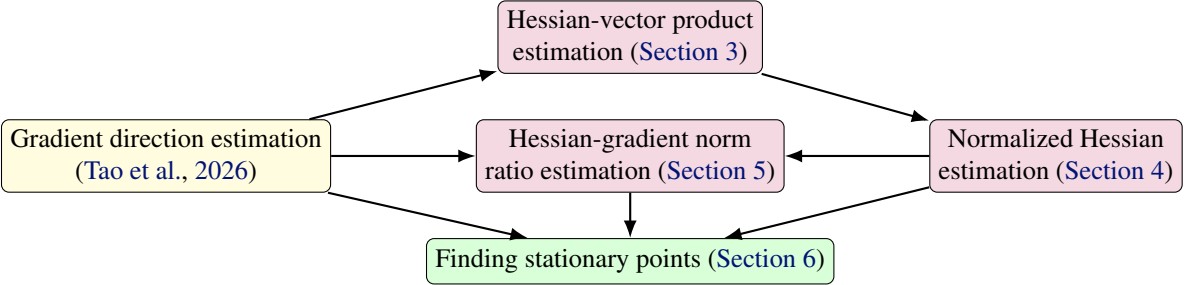

*Figure 2.* The intuition of ComparisonHessVec (Algorithm 1) for computing Hessian-vector products using gradient directions.

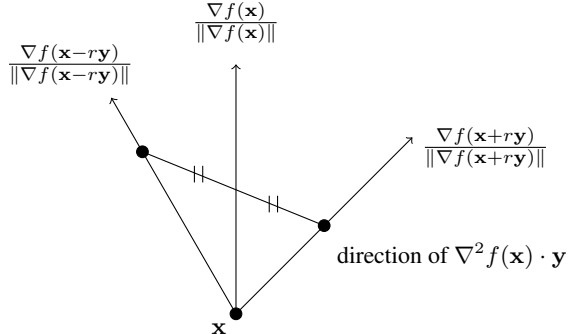

## 2. Preliminaries

**Basic notations.** Let $f\colon \mathbb{R}^n \to \mathbb{R}$ be a (possibly) nonconvex, twice differentiable function with Lipschitz continuous gradient and Hessian, with Lipschitz coefficients $L_1$ and $L_2$ respectively. We define $L_1$-gradient Lipschitz ($L_1$-smooth) function $f\colon \mathbb{R}^n \to \mathbb{R}$ as

$$\|\nabla f(\mathbf{x}) - \nabla f(\mathbf{y})\| \le L_1\|\mathbf{x} - \mathbf{y}\| \quad \forall \mathbf{x}, \mathbf{y} \in \mathbb{R}^n.$$

Similarly, we say $f$ is $L_2$-Hessian Lipschitz if

$$\|\nabla^2 f(\mathbf{x}) - \nabla^2 f(\mathbf{y})\| \le L_2\|\mathbf{x} - \mathbf{y}\| \quad \forall \mathbf{x}, \mathbf{y} \in \mathbb{R}^n.$$

Throughout the paper, for a fixed point $\mathbf{x} \in \mathbb{R}^n$, we denote $\mathbf{g} = \nabla f(\mathbf{x})$ and $H = \nabla^2 f(\mathbf{x})$ for convenience. For any vector $\mathbf{v}$ and matrix $A$, let $\|\mathbf{v}\|$ and $\|A\|$ be their $\ell_2$ norm. We denote $B_R^n(\mathbf{x}) \coloneqq \{\mathbf{y} \in \mathbb{R}^n\colon \|\mathbf{y} - \mathbf{x}\| \le R\}$ and $S_R^n(\mathbf{x}) \coloneqq \{\mathbf{y} \in \mathbb{R}^n\colon \|\mathbf{y} - \mathbf{x}\| = R\}$.

**Quantum notations.** We briefly introduce some basic notations of quantum computing that will be used in this paper. We use the Dirac notation $|\cdot\rangle$ to represent quantum states, which can be seen as column vectors. Qubit, which is the basic unit of quantum state, is represented as $|\phi\rangle = \alpha|0\rangle + \beta|1\rangle$ on the computational basis $\{|0\rangle, |1\rangle\}$ with $\alpha, \beta \in \mathbb{C}$ and $|\alpha|^2 + |\beta|^2 = 1$. An $n$-qubit system

is represented in a $2^n$-dimensional vector space, with basis states $\{|0\rangle, |1\rangle\}^{\otimes n}$. An $n$-qubit state can be written as $|\psi\rangle = \sum_{x \in \{0,1\}^n} \alpha_x|x\rangle$ with $\sum_x |\alpha_x|^2 = 1$. A quantum algorithm can access a function via queries to a quantum oracle. For a classical function $g$, the oracle $O_g$ is defined as a unitary transformation that maps $|x\rangle|b\rangle$ to $|x\rangle|b \oplus g(x)\rangle$. This allows the oracle to be queried on superpositions of inputs, producing corresponding superpositions of outputs.

**Gradient direction estimation using comparisons.** Recently, Tao et al. (2026) developed classical and quantum algorithms for gradient direction estimation using comparisons. These algorithms serve as basic building blocks in our algorithms, stated below:

**Theorem 2.1** (Classical gradient direction estimation, Theorem 3.3 in Tao et al. 2026). *Let $f\colon \mathbb{R}^n \to \mathbb{R}$ with $L_1$-Lipschitz gradient. Given query access to comparison oracle $\mathcal{O}_f^{\mathrm{comp}}$ (Eq. (1)) and a point $\mathbf{x} \in \mathbb{R}^n$, promising that $\|\nabla f(\mathbf{x})\| \ge \gamma$, for any precision $\epsilon$, Algorithm 3 in Tao et al. (2026) (here we denote as ComparisonGE$(\mathbf{x}, \epsilon, \gamma)$) outputs a unit vector $\widehat{\mathbf{g}}$ such that $\left\|\widehat{\mathbf{g}} - \frac{\nabla f(\mathbf{x})}{\|\nabla f(\mathbf{x})\|}\right\| \le \epsilon$ with success probability at least $2/3$, using $O(n\log(1/\epsilon))$ queries.*

**Theorem 2.2** (Quantum gradient direction estimation, Theorem 4.2 in Tao et al. 2026). *In setting of Theorem 2.1, for any precision $\epsilon$, Algorithm 5 in Tao et al. (2026) (here we denote as ComparisonQGE$(\mathbf{x}, \epsilon, \gamma)$) outputs a unit vector $\widehat{\mathbf{g}}$ such that $\left\|\widehat{\mathbf{g}} - \frac{\nabla f(\mathbf{x})}{\|\nabla f(\mathbf{x})\|}\right\| \le \epsilon$ with success probability at least $8/15 - 2\epsilon$, using $O(\log \frac{n}{\epsilon})$ queries.*

## 3. Hessian-Vector Product Estimation by Comparisons

In this section, we first introduce a useful subroutine for Hessian estimation, which is critical to Newton-type method. The subroutine is summarized as ComparisonHessVec (Algorithm 1), which approximates the direction of Hessian-vector products using comparisons.

**Theorem 3.1.** *Let $f\colon \mathbb{R}^n \to \mathbb{R}$ have $L_1$-Lipschitz gradient*

**Algorithm 1:** `ComparisonHessVec`

**Input:** $f \colon \mathbb{R}^n \to \mathbb{R}$, $\mathbf{x}, \mathbf{y} \in \mathbb{R}^n$, precision $\hat{\delta}$, $\gamma_{\mathbf{x}}$, $\gamma_{\mathbf{y}}$.

1 Set $r_0 \leftarrow \min\left\{ \frac{\gamma_{\mathbf{x}}}{100L_1}, \frac{\gamma_{\mathbf{x}}}{100L_2}, \frac{\sqrt{\gamma_{\mathbf{x}}\hat{\delta}}}{20\sqrt{L_2}}, \frac{\gamma_{\mathbf{y}}\hat{\delta}\sqrt{\epsilon}}{20\sqrt{L_2}} \right\}$.

2 $\hat{\mathbf{g}}_0 \leftarrow \text{ComparisonGE}(\mathbf{x}, \frac{L_2 r_0^2}{\gamma_{\mathbf{x}}}, \gamma_{\mathbf{x}})$,

$\hat{\mathbf{g}}_1 \leftarrow \text{ComparisonGE}(\mathbf{x} + r_0\mathbf{y}, \frac{\rho r_0^2}{\gamma_{\mathbf{x}}}, \gamma_{\mathbf{x}}/2)$,

$\hat{\mathbf{g}}_{-1} \leftarrow \text{ComparisonGE}(\mathbf{x} - r_0\mathbf{y}, \frac{L_2 r_0^2}{\gamma_{\mathbf{x}}}, \gamma_{\mathbf{x}}/2)$.

3 Set $\mathbf{g} = \sqrt{1 - \langle \hat{g}_{-1}, \hat{g}_0 \rangle^2}\,\hat{g}_1 - \sqrt{1 - \langle \hat{g}_1, \hat{g}_0 \rangle^2}\,\hat{g}_{-1}$.

4 **return** $\hat{\mathbf{u}}(\mathbf{x}, \mathbf{y}, \delta, \gamma_{\mathbf{x}}, \gamma_{\mathbf{y}}) = \mathbf{g}/\|\mathbf{g}\|$.

*and $L_2$-Lipschitz Hessian. For any $\gamma_{\mathbf{x}}, \gamma_{\mathbf{y}} > 0$ and $\mathbf{x}, \mathbf{y} \in \mathbb{R}^d$ satisfying*

$$\|\nabla f(\mathbf{x})\| \geq \gamma_{\mathbf{x}}, \quad \lambda_{\min}(\nabla^2 f(\mathbf{x})) \leq -\sqrt{L_2 \epsilon},$$
$$\|\mathbf{y}\| = 1, \qquad |\mathbf{y}_1| \geq \gamma_{\mathbf{y}},$$

*where $\mathbf{y}_1 = \langle \mathbf{y}, \mathbf{u}_1 \rangle$ with $\mathbf{u}_1$ being any eigenvector of $\nabla^2 f(\mathbf{x})$ with the smallest eigenvalue, Algorithm 1 outputs a vector $\hat{\mathbf{u}}$ satisfying*

$$\left\| \hat{\mathbf{u}} - \frac{\nabla^2 f(\mathbf{x}) \cdot \mathbf{y}}{\|\nabla^2 f(\mathbf{x}) \cdot \mathbf{y}\|} \right\| \leq \hat{\delta}$$

*using $\tilde{O}\big(n \log\big(1/(\gamma_{\mathbf{x}} \gamma_{\mathbf{y}}^2 \hat{\delta}^2)\big)\big)$ queries.*

The proof of Theorem 3.1 is deferred to Appendix B. Our intuition here is to use Taylor expansion $\nabla f(\mathbf{x} \pm r\mathbf{y}) \approx \nabla f(\mathbf{x}) \pm r\nabla^2 f(\mathbf{x}) \cdot \mathbf{y}$ and solving the triangle in Figure 3. This operation can be applied only with the three normalized gradients $\frac{\nabla f(\mathbf{x})}{\|\nabla f(\mathbf{x})\|}, \frac{\nabla f(\mathbf{x}_+)}{\|\nabla f(\mathbf{x}_+)\|}, \frac{\nabla f(\mathbf{x}_-)}{\|\nabla f(\mathbf{x}_-)\|}$.

## 4. Robust Hessian Estimation by Comparisons

In this section, we show how to obtain an estimate of the normalized Hessian at a given point using comparisons.

Denote the columns of $H$ as $H = (\mathbf{h}_1, \ldots, \mathbf{h}_n)$ with $\mathbf{h}_i = H\mathbf{e}_i$ and define the relative column norms $\hat{r}_i := \|\mathbf{h}_i\|/\|\mathbf{h}_1\| \in [0, 1]$. Here without loss of generality, we assume $\|H\mathbf{e}_1\| = \max_i \|H\mathbf{e}_i\|$. Otherwise, by testing $\left\langle \frac{\mathbf{h}_i}{\|\mathbf{h}_i\|}, \frac{\mathbf{h}_i + \mathbf{h}_j}{\|\mathbf{h}_i + \mathbf{h}_j\|} \right\rangle$ and $\left\langle \frac{\mathbf{h}_j}{\|\mathbf{h}_j\|}, \frac{\mathbf{h}_i + \mathbf{h}_j}{\|\mathbf{h}_i + \mathbf{h}_j\|} \right\rangle$, we can roughly tell which one is larger and then relabel columns. The error only incurs a constant overhead in the complexity. Our estimator first estimates the *directions* $\mathbf{h}_i/\|\mathbf{h}_i\|$ and $(\mathbf{h}_1 + \mathbf{h}_i)/\|\mathbf{h}_1 + \mathbf{h}_i\|$ via Hessian-vector product queries at (perturbed) $\mathbf{e}_i$ and $\mathbf{e}_1 + \mathbf{e}_i$, and then computes $\hat{r}_i$ by applying Lemma A.3 in parallelogram spanned by (perturbed) $\mathbf{h}_1$, $\mathbf{h}_i$ and $\mathbf{h}_1 + \mathbf{h}_i$ in non-degenerate cases that the direction of $\mathbf{h}_1$ and $\mathbf{h}_i$ are not close. Otherwise nearly-parallel cases are handled by a tiny perturbation.

**Algorithm 2:** `ComparisonHE`$(\mathbf{x}, \delta)$: Estimate Normalized Hessian at $\mathbf{x}$ with Precision $\delta$ by Comparisons

**Input:** Target point $\mathbf{x}$, target accuracy $\delta > 0$
**Output:** $\widehat{H}$

1 Set $\eta \leftarrow c_0\, \delta^4/n^2$, thresholds $\tau_\alpha, \tau_\beta \leftarrow \Theta(\sqrt{\eta})$, $\sigma \leftarrow \frac{1}{8}\eta^2$, sample $t \sim \text{Unif}(\{1, \ldots, n\})$.

2 Define $\mathbf{y}_i^{\pm} \leftarrow \frac{\mathbf{e}_i \pm \sigma \mathbf{e}_t}{\sqrt{1+\sigma^2}}$ for all $i \in [n]$.

3 Define $\mathbf{z}_i^{\pm} \leftarrow \frac{\mathbf{e}_1 + \mathbf{e}_i \pm \sigma \mathbf{e}_t}{\sqrt{2+\sigma^2}}$ for all $i \in \{2, \ldots, n\}$.

4 **for** $i = 1, \ldots, n$ **do**

5    Query $\mathbf{g}_i^0 \leftarrow \hat{\mathbf{u}}(\mathbf{x}, \mathbf{e}_i, \eta/4, \epsilon, \eta/2\sqrt{n})$,
     $\mathbf{g}_i^+ \leftarrow \hat{\mathbf{u}}(\mathbf{x}, \mathbf{y}_i^+, \eta/4, \epsilon, \eta/2\sqrt{n})$,
     $\mathbf{g}_i^- \leftarrow \hat{\mathbf{u}}(\mathbf{x}, \mathbf{y}_i^-, \eta/4, \epsilon, \eta/2\sqrt{n})$.

6    **if** $\langle \mathbf{g}_i^+, \mathbf{g}_i^- \rangle \leq 1 - \tau_\alpha$ **then**

7      Set $\mathbf{g}_i \leftarrow \mathbf{0}$.

8    **else**

9      $(\mathbf{v}_1, \mathbf{v}_2) \leftarrow$
     $\arg\max_{\mathbf{v}_1 \neq \mathbf{v}_2 \in \{\mathbf{g}_i^0, \mathbf{g}_i^+, \mathbf{g}_i^-\}} \langle \mathbf{v}_1, \mathbf{v}_2 \rangle$.

10      Set $\mathbf{g}_i \leftarrow \frac{\mathbf{v}_1 + \mathbf{v}_2}{\|\mathbf{v}_1 + \mathbf{v}_2\|}$.

11 **for** $i = 2, \ldots, n$ **do**

12    Query $\mathbf{g}_{1i}^0 \leftarrow \hat{\mathbf{u}}(\mathbf{x}, \mathbf{e}_1 + \mathbf{e}_i, \eta/4, \epsilon, \eta/2\sqrt{n})$,
     $\mathbf{g}_{1i}^+ \leftarrow \hat{\mathbf{u}}(\mathbf{x}, \mathbf{z}_i^+, \eta/4, \epsilon, \eta/2\sqrt{n})$,
     $\mathbf{g}_{1i}^- \leftarrow \hat{\mathbf{u}}(\mathbf{x}, \mathbf{z}_i^-, \eta/4, \epsilon, \eta/2\sqrt{n})$.

13    **if** $\langle \mathbf{g}_{1i}^+, \mathbf{g}_{1i}^- \rangle \leq 1 - \tau_\beta$ **then**

14      Set $\mathbf{g}_{1i} \leftarrow \mathbf{0}$.

15    **else**

16      $(\mathbf{v}_1, \mathbf{v}_2) \leftarrow$
     $\arg\max_{\mathbf{v}_1 \neq \mathbf{v}_2 \in \{\mathbf{g}_{1i}^0, \mathbf{g}_{1i}^+, \mathbf{g}_{1i}^-\}} \langle \mathbf{v}_1, \mathbf{v}_2 \rangle$.

17      Set $\mathbf{g}_{1i} \leftarrow \frac{\mathbf{v}_1 + \mathbf{v}_2}{\|\mathbf{v}_1 + \mathbf{v}_2\|}$.

18 Set $\hat{r}_1 \leftarrow 1$.

19 **for** $i \leftarrow 2$ **to** $n$ **do**

20    $\widehat{\alpha}_i \leftarrow \langle \mathbf{g}_i, \mathbf{g}_1 \rangle$, $\widehat{\beta}_i \leftarrow \langle \mathbf{g}_{1i}, \mathbf{g}_1 \rangle$.

21    **if** $\widehat{\beta}_i \geq 1 - \tau_\beta$ **then**

22      **if** $\widehat{\alpha}_i \leq 1 - \tau_\alpha$ **then**

23        $\hat{r}_i \leftarrow 0$.

24      **else**

25        $\hat{r}_i \leftarrow \text{PerturbAndSolve}(i, \eta)$.

26    **else**

27      Solve Eq. (4) with $(\alpha, \beta) = (\widehat{\alpha}_i, \widehat{\beta}_i)$ and take the unique root $\hat{r}_i \in [0, 1]$.

28 Form $\widetilde{H} \leftarrow [\hat{r}_1 \mathbf{g}_1, \ldots, \hat{r}_n \mathbf{g}_n]$.

29 $\widetilde{H} \leftarrow \frac{\widetilde{H} + \widetilde{H}^\top}{2}$.

30 **return** $\widehat{H} \leftarrow \widetilde{H}/\|\widetilde{H}\|$.

---

**Algorithm 3:** `PerturbAndSolve(i, η)`

---

1 Set $\rho \leftarrow \eta^{1/3}$.
2 **for** $j \leftarrow 2$ **to** $n$ **do**
3    **if** $j \neq i$ and $\widehat{\alpha}_j \leq 1 - \tau_\alpha$ and $\widehat{\alpha}_{1j} \leq 1 - \rho^2$ **then**
4       Query
      $\mathbf{g}_{i,j} \leftarrow \widehat{\mathbf{u}}(\mathbf{x}, \mathbf{e}_i + \rho\mathbf{e}_j, \eta/4, \epsilon, \eta/2\sqrt{n})$.
5       $\widehat{\alpha}_{i,j} \leftarrow \langle \mathbf{g}_{i,j}, \mathbf{g}_1 \rangle$.
6       Query $\mathbf{g}_{1,i,j} \leftarrow$
      $\widehat{\mathbf{u}}(\mathbf{x}, \mathbf{e}_1 + \mathbf{e}_i + \rho\mathbf{e}_j, \eta/4, \epsilon, \eta/2\sqrt{n})$.
7       $\widehat{\beta}_{i,j} \leftarrow \langle \mathbf{g}_{1,i,j}, \mathbf{g}_1 \rangle$.
8       **if** $\langle \mathbf{g}_{i,j}, \mathbf{g}_1 \rangle \leq 1 - \rho$ **then**
9          **return** $\hat{r}_i \leftarrow 0$.
10       Solve Eq. (4) with $(\alpha, \beta) = (\widehat{\alpha}_{i,j}, \widehat{\beta}_{i,j})$ and
      take $\hat{r}_{i,j} \in [0, 1 + \rho]$.
11       **return** $\hat{r}_i \leftarrow \hat{r}_{i,j}$.
12 **return** $\hat{r}_i \leftarrow 1$.

---

The computation of $\hat{r}_i$ can be roughly stated as follows: for each $i \geq 2$ with $\mathbf{h}_i \neq \mathbf{0}$ and $\mathbf{h}_1 + \mathbf{h}_i \neq 0$, define unit vectors

$$\mathbf{u}_1 := \frac{\mathbf{h}_1}{\|\mathbf{h}_1\|}, \quad \mathbf{u}_i := \frac{\mathbf{h}_i}{\|\mathbf{h}_i\|}, \quad \mathbf{u}_{1i} := \frac{\mathbf{h}_1 + \mathbf{h}_i}{\|\mathbf{h}_1 + \mathbf{h}_i\|}.$$

Let $\alpha_i := \langle \mathbf{u}_i, \mathbf{u}_1 \rangle$ and $\beta_i := \langle \mathbf{u}_{1i}, \mathbf{u}_1 \rangle$. A direct computation yields

$$\beta_i = F(\hat{r}_i, \alpha_i) := \frac{1 + \hat{r}_i \alpha_i}{\sqrt{1 + \hat{r}_i^2 + 2\hat{r}_i \alpha_i}}. \tag{3}$$

Squaring Eq. (3) gives the quadratic equation used by the solver:

$$(\beta_i^2 - \alpha_i^2)\hat{r}_i^2 + 2\alpha_i(\beta_i^2 - 1)\hat{r}_i + (\beta_i^2 - 1) = 0. \tag{4}$$

By solving this equation, we can roughly estimate $\hat{r}_i$, thus we can form our target $\widehat{H}$ (see Line 28 and Line 29).

**Theorem 4.1** (Informal; full statement in Theorem C.10). *Let $f \colon \mathbb{R}^n \to \mathbb{R}$ have $L_1$-Lipschitz gradient and $L_2$-Lipschitz Hessian. Given that $\|H\| \geq \sqrt{L_2\epsilon}$ and $\|\mathbf{g}\| \geq \epsilon$, with probability at least $2/3$, Algorithm 2 outputs a matrix $\widehat{H}$, satisfying $\left\| \widehat{H} - \frac{H}{\|H\|} \right\| \leq \delta$ for any given precision $\delta$. The query complexity of Algorithm 2 is $\tilde{O}(n^2 \log(1/\delta))$.*

The full proof is deferred to Appendix C; we present the proof intuition here.

*Proof sketch.* The analysis has two basic cases: *(i)* When $\mathbf{h}_1$ and $\mathbf{h}_i$ overlap relatively small in directions and their norms are not too small, i.e. $\alpha_i$ and $\beta_i$ are bounded away from 1, we can solve Eq. (4) with bounded error on $|\hat{r}_i -$

$r_i|$. *(ii)* Otherwise, when $\alpha_i$ and $\beta_i$ are close to 1, Eq. (4) becomes degenerate. There are two subcases that may cause degeneracy: $\|\mathbf{h}_i\|$ is much smaller than $\|\mathbf{h}_1\|$, or $\mathbf{h}_1$ and $\mathbf{h}_i$ have close directions.

With constant probability (guaranteed by Lemma A.6), our sampled $t$ satisfies $|\langle \mathbf{e}_t, \mathbf{v} \rangle| \geq 1/\sqrt{n}$, where $\mathbf{v}$ is the eigenvector corresponding to minimum eigenvalue of $H$, here we use the isotropy of $n$ coordinates. Once we can satisfy a $1/\sqrt{n}$-overlap, our choice of $\mathbf{g}_i$ and $\mathbf{g}_{1i}$ satisfies $\|\mathbf{g}_i - \mathbf{u}_i\| \leq \eta$ and $\|\mathbf{g}_{1i} - \mathbf{u}_{1i}\| \leq \eta$ according to Lemma C.2. Inner products satisfy $|\widehat{\alpha}_i - \alpha_i| \leq O(\eta)$ and $|\widehat{\beta}_i - \beta_i| \leq O(\eta)$.

When $\alpha_i$ is bounded away from 1 and $r_i$ is not tiny, the implicit map $\beta = F(r, \alpha)$ has slope $|\partial F/\partial r| = \Omega(1/r)$, so solving the quadratic Eq. (4) gives $|\hat{r}_i - r_i| = O(\eta/r_{\min}\tau_\alpha)$. (See Lemma C.4.)

If $\mathbf{u}_{1i} \approx \mathbf{u}_1$, the inversion is ill-conditioned. Note that it can be caused by two cases: *(i)* When $\|\mathbf{h}_i\|$ is relatively small, we can tell this case by testing if the angle between $\mathbf{u}_1$ and $\mathbf{u}_i$ is relatively large, and we can directly treat $\|\mathbf{h}_i\| = 0$ since the error of column vectors cause bounded error to Hessian estimation. *(ii)* When $\mathbf{h}_1$ and $\mathbf{h}_i$ have close directions, to avoid uncontrollable error, we apply Algorithm 3 to enforce an angle separation $1 - \alpha^2 = \Omega(\tau_\alpha)$, in which we use perturbed $\mathbf{e}_i + \rho\mathbf{e}_j$ to replace $\mathbf{e}_i$, and perturbed $\mathbf{e}_1 + \mathbf{e}_i + \rho\mathbf{e}_j$ to replace $\mathbf{e}_1 + \mathbf{e}_i$ to guarantee the non-degeneracy in the computation of the ratio $\hat{r}_i$. This makes the inverse map Lipschitz with coefficient $O(1/\rho^2)$, giving a noise term $O(\eta/\sqrt{\tau_\alpha}\rho^2)$. The perturbation changes the effective triangle by $O(\rho/\sqrt{\tau_\alpha})$, producing a bias term $O(\rho/\sqrt{\tau_\alpha})$. Hence

$$|\hat{r}_i - r_i| \leq \frac{1}{\sqrt{\tau_\alpha}}(C\frac{\eta}{\rho^2} + B\rho),$$

optimizing it gives $|\hat{r}_i - r_i| = O(\eta^{1/4})$. (See Lemma C.9.)

Finally, assembling columns $\hat{r}_i g_i$ yields a scale-free matrix $\widetilde{H}$ satisfying $\|\widetilde{H} - H^\star\| \leq O(\sqrt{n}\eta^{1/4})$, and normalization stability implies the same order bound for $\widehat{H} - \frac{H}{\|H\|}$. (See Lemma A.4 and Lemma A.5.) $\qquad \square$

## 5. Estimation of the Ratio between Hessian Norm and Gradient Norm

To apply Newton-type methods, we are supposed to give an algorithm to estimate the second-order Taylor expansion at any given point. We can achieve this by estimating the normalized gradient, Hessian, and the ratio between their norms. Without loss of generality, we assume:

**Assumption 5.1** (Properties of Hessian).

   1. The Hessian $H$ is not rank-one. (Otherwise, if $H$ is

**Algorithm 4:** `ComparisonRatio(x, δκ)`: Estimate Gradient-Hessian Ratio at point $\mathbf{x}$ with precision $\delta_\kappa$ by Comparisons

---

**Input:** Point $\mathbf{x}$, lower bound of gradient $\epsilon$ precision $\delta_\kappa$, degenerate parameters $\beta, \lambda$
**Output:** Estimate $\hat{\kappa} \approx \|\nabla^2 f(\mathbf{x})\| / \|\nabla f(\mathbf{x})\|$

**1** $\mu \leftarrow \tan^2 \beta \cdot \epsilon^2 \delta_\kappa^2$.

**2** $(\xi_1, \xi_2, \xi_3) \leftarrow (O(\epsilon\delta_\kappa), O(\epsilon\delta_\kappa), O(\epsilon^2\delta_\kappa^2/n))$.

**3** $\widehat{H} \leftarrow \texttt{ComparisonHE}(\mathbf{x}; \xi_3)$.

**4** Compute a unit eigenvector $v$ of $\widehat{H}$ associated with the eigenvalue $\lambda_v$ with the largest absolute value

**5** $\widehat{\mathbf{g}}_1 \leftarrow \texttt{ComparisonGE}(\mathbf{x}; \xi_1)$.

**6** $\widehat{\mathbf{g}}_2 \leftarrow \texttt{ComparisonGE}(\mathbf{x} + \mu\mathbf{v}; \xi_2)$.

**7** $\hat{\kappa} \leftarrow \dfrac{\sqrt{1 - \langle \widehat{\mathbf{g}}_1, \widehat{\mathbf{g}}_2 \rangle^2}}{\mu \sqrt{1 - \langle \widehat{\mathbf{g}}_2, \mathbf{v} \rangle^2}}$.

**8 return** $\hat{\kappa}$.

---

rank-one, it is separately handled in Lemma E.3.)

2. The Hessian $H$ is neither positive definite nor negative definite. (Otherwise, if $H$ is positive definite or negative definite, it is separately handled in Lemma E.4.)

Under Assumption 5.1, we estimate the ratio $\|\nabla^2 f(\mathbf{x})\| / \|\nabla f(\mathbf{x})\|$ using (i) one call to our normalized Hessian estimation routine and (ii) two calls to a normalized gradient direction estimator, stated as follows:

**Theorem 5.2.** *Given that $f$ have $L_1$-Lipschitz gradient and $L_2$-Lipschitz Hessian, for target point $\mathbf{x} \in \mathbb{R}^n$ with $\|\mathbf{g}\| = \|\nabla f(\mathbf{x})\| \geq \epsilon$. Let $H = \nabla^2 f(\mathbf{x})$ and assume Assumption 5.1. Suppose there exists a unit eigenvector $\mathbf{u}$ of $H$ with eigenvalue $\lambda_\mathbf{u}$ such that*

$$\arccos \left\langle \frac{\mathbf{g}}{\|\mathbf{g}\|}, \mathbf{u} \right\rangle \geq \beta \quad and \quad \|H\mathbf{u}\| = |\lambda_\mathbf{u}| \geq \lambda, \quad (5)$$

*for some $\beta \in (0, \pi/2)$ and $\lambda > 0$.*

*For any precision $\delta_\kappa > 0$, the output $\hat{\kappa}$ satisfies*

$$\left| \hat{\kappa} - \frac{\|H\|}{\|\mathbf{g}\|} \right| \leq \delta_\kappa. \quad (6)$$

*The query complexity of Algorithm 4 is $\tilde{O}(n^2 \log (1/\delta_\kappa))$.*

We sketch the intuition of our proof here, with full details deferred to Appendix D.

*Proof sketch. Step 1 (Taylor approximation)* By Lipschitzness of the Hessian, we have the second-order expansion for some small $\mu$,

$$\nabla f(\mathbf{x} + \mu\mathbf{v}) = \mathbf{g} + \mu H \mathbf{v} + r_\mu, \qquad \|r_\mu\| \leq \tfrac{1}{2} L_2 \mu^2. \quad (7)$$

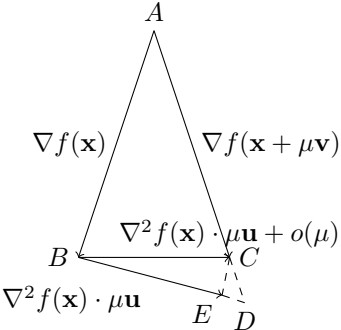

Here we choose $\mathbf{v}$ as an eigenvector of $\widehat{H}$ corresponding to the eigenvalue $\lambda_\mathbf{v}$ with largest absolute value satisfying $\arccos \left\langle \frac{\mathbf{g}_1}{\|\mathbf{g}_1\|}, \mathbf{v} \right\rangle \geq \beta$ and $\|\widehat{H}\mathbf{v}\| = |\lambda_\mathbf{v}| \geq \lambda$. Thus by law of sines, we can get an estimation of the Hessian-gradient norm ratio. We assume the non-degenerate case here. (Otherwise, when finding the stationary point, we apply line search on direction $\mathbf{v}$ to handle this case.)

*Step 2 (Neglect higher-order terms)* As shown in Figure 3 we can use line segment BE to replace BC, which allows us to compute the ratio $\frac{\|\nabla^2 f(\mathbf{x}) \cdot \mu\mathbf{u}\|}{\|\nabla f(\mathbf{x})\|} = \frac{\mu\lambda_\mathbf{u}}{\|\nabla f(\mathbf{x})\|} = \mu \frac{\|\nabla^2 f(\mathbf{x})\|}{\|\nabla f(\mathbf{x})\|}$.

*Step 3 (Solve the triangle)* We try to use $\hat{\mathbf{g}}_1, \widehat{H}$, and $\mathbf{v}$ to approximate real properties of function. To apply Lemma A.3 (equivalently, the law of sines), we need to extend the line segment BE to D by the ratio $\eta$. Approximately, we use our estimation result and bound the error. Once the rays are aligned, Lemma A.3 expresses $\|\eta'\mu H \mathbf{v}\| / \|\mathbf{g}\|$ using only angles between normalized directions. Algorithm 4 computes the same quantity with $\hat{\mathbf{g}}_1, \hat{\mathbf{g}}_2$ and $\mathbf{v}$.

*Step 4 (Bound the error)* A core lemma we use here is the Davis-Kahan Theorem (Theorem 1 in Yu et al. 2015), which bounds the distance of two subspaces spanned by eigenvectors under perturbation. We use this tool to bound $\|\mathbf{u} - \mathbf{v}\|$ and thus get an upper bound of the error $|\hat{\kappa} - \frac{\|H\|}{\|\mathbf{g}\|}|$. Note that in extreme cases, $\lambda_\mathbf{v}$ and its neighbor eigenvalues of $\widehat{H}$ are close enough to form a cluster. To handle this case, assume that all $n$ eigenvalues of $\widehat{H}$ are $\lambda'_1 \geq \cdots \geq \lambda'_n$ (correspond to eigenvectors $\mathbf{v}_1, \ldots, \mathbf{v}_n$) and all $n$ eigenvalues of $H$ are $\lambda_1 \geq \cdots \geq \lambda_n$ (correspond to eigenvectors $\mathbf{u}_1, \ldots, \mathbf{u}_n$). Find the first index $k$ satisfying $\lambda'_k - \lambda'_{k+1} \geq O(\epsilon\delta_\kappa/n)$, and define the cluster $\widehat{\Lambda} = \{\lambda'_1, \ldots, \lambda'_k\}$. We consider eigenvectors correspond to eigenvalues in this cluster, and denote the eigenspace spanned by them as $V$, similarly define $U$ for $H$. In *Step 1* and Figure 3, we actually take $\mathbf{u}$ as the projection of $\mathbf{v}$ in $U$. We claim that $\mathbf{u}$ has two properties:

*(i)* $\|\mathbf{u} - \mathbf{v}\| \leq O\left(\frac{n\xi_3}{\epsilon\delta_\kappa}\right)$;

---

**Algorithm 5:** ComparisonTR($\epsilon, \xi_1, \xi_2, \xi_3, T$):
Finding stationary point

**Input:** Initial point $\mathbf{x}_0$, precision $\epsilon$, tolerances
$\quad(\xi_1, \xi_2, \xi_3) = (1/100, 1/100, 1/100n)$,
$\quad$number of iterations $T$

**Output:** A stationary point $\mathbf{x}_T$

1 $\quad r \leftarrow c\sqrt{\epsilon/L_2}$.

2 $\quad \delta_\kappa \leftarrow c_0\sqrt{\frac{\epsilon}{L_2}}$.

3 **for** $t = 0, \dots, T-1$ **do**

4 $\qquad \hat{\mathbf{g}} \leftarrow$ ComparisonGE($\mathbf{x}_t; \xi_1$).

5 $\qquad \hat{H} \leftarrow$ ComparisonHE($\mathbf{x}_t; \xi_3$).

6 $\qquad \hat{\kappa} \leftarrow$ ComparisonRatio($\mathbf{x}_t; \delta_\kappa$).

7 $\qquad$ Compute unit eigenvector $\mathbf{v}$ of $\hat{H}$ with
$\qquad$ eigenvalue of largest magnitude.

8 $\qquad \mathbf{x}^{(1)} \leftarrow \mathbf{x}_t - r\hat{\mathbf{g}}$.

9 $\qquad$ Find $l^\star \in [-r, r]$ approximately minimizing
$\qquad f(\mathbf{x}_t + l\mathbf{v})$.

10 $\qquad \mathbf{x}^{(2)} \leftarrow \mathbf{x}_t + l^\star\mathbf{v}$.

11 $\qquad$ Define

$$m_{\mathbf{x}}(\mathbf{p}) = \langle \hat{\mathbf{g}}, \mathbf{p}\rangle + \frac{\hat{\kappa}}{2}\mathbf{p}^\top \hat{H}\mathbf{p}, \qquad \|\mathbf{p}\| \leq r.$$

$\qquad$ Let $\mathbf{p}^\star \in \arg\min_{\|\mathbf{p}\|\leq r} m_{\mathbf{x}}(\mathbf{p})$.

12 $\qquad \mathbf{x}^{(3)} \leftarrow \mathbf{x} + \mathbf{p}^\star$.

13 $\qquad \mathbf{x}_{t+1} = \arg\min\{f(\mathbf{x}^{(1)}), f(\mathbf{x}^{(2)}), f(\mathbf{x}^{(3)})\}$.

14 **return** $\mathbf{x}_T$

---

*(ii)* $\|H\| - \|H\mathbf{u}\| \leq \frac{\epsilon\delta_\kappa}{2} + 2\xi_3$.

We can take $\xi_3$ sufficiently small to apply Davis-Kahan Theorem (as shown in Figure 4 in Appendix D.2), which tells us $\|\sin(\Theta(U, V))\| \leq \frac{\xi_3}{\lambda'_k - \lambda'_{k+1}}$. With these two properties we can get an upper bound of $|\hat{\kappa} - \frac{\|H\|}{\|\mathbf{g}\|}|$. Full details are deferred to Appendix D. $\quad\square$

## 6. Finding Stationary Points

In this section, we present our algorithm for finding stationary points by comparisons in Algorithm 5.

If in some steps $\nabla f(\mathbf{x}_t) \leq \epsilon$, then we have already visited a $\epsilon$-stationary point. At any iterate $\mathbf{x}_t$ with $\|\nabla f(\mathbf{x}_t)\| \geq \epsilon$, Algorithm 5 finds the next iterate $\mathbf{x}_{t+1}$ that decreases the function value by $\Omega(\epsilon^{3/2})$. If after limited steps we cannot visit a stationary point, the function value will decrease more than $f(\mathbf{x}_0) - f(\mathbf{x}^\star)$, which is a contradiction. The iteration rule constructs three candidate steps (gradient step, trust-region step on a normalized quadratic model, and rank-one line-search step), and then selects the candidate with minimal function value by comparison. If the Hessian is rank-one,

line search performs well. Else if in other degenerate cases (the Hessian has too small norm, positive/negative definite, or the eigenvector with the largest absolute eigenvalue has significant overlap with the gradient), a normalized gradient descent step decreases $\Omega(\epsilon^{1.5})$. Else, in non-degenerate cases, trust-region step guarantees the decrease.

**Theorem 6.1.** *For any point $\mathbf{x}_t$ with $\|\mathbf{g}(\mathbf{x}_t)\| \geq \epsilon$, if we successfully estimate the Hessian, then a loop of Algorithm 5 returns a point $\mathbf{x}_{t+1}$ satisfying*

$$f(\mathbf{x}_{t+1}) \leq f(\mathbf{x}_t) - C\,\epsilon^{1.5} \qquad (8)$$

*for an absolute constant $C > 0$.*

The proof details are deferred to Appendix E; we present a proof sketch here.

*Proof sketch of Theorem 6.1.* If $\|\nabla f(\mathbf{x}_t)\| \leq \epsilon$, the algorithm has visited a stationary point and we are done. Assume $\|\mathbf{g}\| := \|\nabla f(\mathbf{x}_t)\| \geq \epsilon$ and set $r = c\sqrt{\epsilon/L_2}$.

First we bound the error between the target function and its Taylor reduction. Let

$$T_{\mathbf{x}_t}(\mathbf{x}_t + \mathbf{p}) = f(\mathbf{x}_t) + \langle \mathbf{g}, \mathbf{p}\rangle + \tfrac{1}{2}\mathbf{p}^\top H\mathbf{p}$$

be the quadratic form at $\mathbf{x}_t$ by truncating the Taylor series to the second power. By $L_2$-Lipschitz Hessian, Lemma E.1 gives $|f(\mathbf{x}_t + \mathbf{p}) - T_{\mathbf{x}_t}(\mathbf{x}_t + \mathbf{p})| \leq \frac{L_2}{6}\|\mathbf{p}\|^3$. Hence, for any $\|\mathbf{p}\| \leq r$,

$$f(\mathbf{x}_t + \mathbf{p}) - f(\mathbf{x}_t) \leq \left(T_{\mathbf{x}_t}(\mathbf{x}_t + \mathbf{p}) - T_{\mathbf{x}_t}(\mathbf{x}_t)\right) + \frac{L_2}{3}r^3$$
$$= \left(T_{\mathbf{x}_t}(\mathbf{x}_t + \mathbf{p}) - T_{\mathbf{x}_t}(\mathbf{x}_t)\right) + O(\epsilon^{3/2}).$$

Therefore it suffices to find a candidate step with $\|p\| \leq r$ such that $T_{\mathbf{x}_t}(\mathbf{x}_t + \mathbf{p}) - T_{\mathbf{x}_t}(\mathbf{x}_t) \leq -\Omega(r\epsilon)$.

Next, Algorithm 5 builds three candidates within the ball:

*(i)* normalized gradient descent along $-\hat{\mathbf{g}}$ on $[0, r]$,

*(ii)* a line search along an eigenvector $\mathbf{v}$ corresponding to the eigenvalue with the largest magnitude on $[-r, r]$,

*(iii)* a trust-region minimizer of the approximate quadratic $m_{\mathbf{x}_t}(\mathbf{p}) = \langle \hat{\mathbf{g}}, \mathbf{p}\rangle + \frac{\hat{\kappa}}{2}\mathbf{p}^\top\hat{H}\mathbf{p}$ in $\|\mathbf{p}\| \leq r$.

A standard trust-region argument (Lemma E.2) implies that *(iii)* either produces an $\epsilon$-stationary point or yields decrease $\Omega(r\|\hat{\mathbf{g}}\|) = \Omega(r\epsilon)$. The normalized gradient descent *(i)* and the line search *(ii)* ensure the same order decrease in the remaining degenerate cases (rank-one, near-colinearity or indefinite Hessian), up to an additive $O(\epsilon^{1.5})$ error, since we only need coarse precision at radius $r$. Note that at each time $t$, we should keep handling the degenerate cases, as in Line 13 we take the minimum of the three cases in each

iteration. This is inevitable since without the information of the gradient and Hessian norms, the algorithm cannot gauge which case is taking place.

The returned point $\mathbf{x}_{t+1}$ satisfies $f(\mathbf{x}_{t+1}) \leq \min\{f(\mathbf{x}^{(1)}), f(\mathbf{x}^{(2)}), f(\mathbf{x}^{(3)})\}$, hence inherits the $\Omega(\epsilon^{1.5})$ decrease from whichever candidate provides it. $\square$

**Theorem 6.2.** *Let $f: \mathbb{R}^n \to \mathbb{R}$ has $L_1$-Lipschitz gradient and $L_2$-Lipschitz Hessian. Denote $\mathbf{x}^\star = \arg\min f(\mathbf{x})$. Given $\mathbf{x}_0 \in \mathbb{R}^n$ satisfying $f(\mathbf{x}_0) - f(\mathbf{x}^\star) \leq \Delta$, with success probability at least $2/3$, Algorithm 5 outputs a list of $O(1/\epsilon^{1.5})$ points, in which at least one point is an $\epsilon$-stationary point, using $T = O(\frac{f(\mathbf{x}_0) - f(\mathbf{x}^\star)}{C'\epsilon^{1.5}})$ iterations. The overall query complexity is $\tilde{O}(\frac{\Delta\sqrt{L_2}n^2}{\epsilon^{1.5}})$.*

*Proof.* We use contradiction to prove the theorem. If the statement is not true and any iterate $\mathbf{x}_t$ is not an $\epsilon$-stationary point, by Theorem 6.1 we have

$$\mathbb{E}[f(\mathbf{x}_{t+1}) - f(\mathbf{x}_t)] = -C'\epsilon^{1.5},$$

where $C' = \Pr \cdot C$, $C$ is defined as in Theorem 6.1, $\Pr \geq 2/3$ is success probability in Theorem 4.1. Sum from $t = 0$ to $T - 1$:

$$\mathbb{E}[f(\mathbf{x}_T) - f(\mathbf{x}_0)] = \sum_{t=0}^{T-1} \mathbb{E}[f(\mathbf{x}_{t+1}) - f(\mathbf{x}_t)]$$
$$= -\sum_{t=0}^{T-1} C'\epsilon^{1.5}$$
$$= -TC'\epsilon^{1.5}.$$

By Markov's Inequality, if $X$ is a nonnegative random variable and $a > 0$, $\Pr(X \geq a) \leq \frac{\mathbb{E}[X]}{a}$, so $\Pr(X < a) \geq 1 - \frac{\mathbb{E}[X]}{a}$. Taking $a = 2TC'\epsilon^{1.5} > f(\mathbf{x}_0) - f(\mathbf{x}^\star)$, $X = a - f(\mathbf{x}_0) - f(\mathbf{x}_T)$, we have:

$$\Pr\left(f(\mathbf{x}_0) - f(\mathbf{x}_T) \geq \frac{1}{2}TC'\epsilon^{1.5}\right)$$
$$\geq \frac{a - \mathbb{E}[f(\mathbf{x}_0) - f(\mathbf{x}_T)]}{a - \frac{1}{2}TC'\epsilon^{1.5}} = \frac{2}{3}.$$

Therefore, with probability at least $2/3$, the total decrease of function value is larger than $f(\mathbf{x}_0) - f(\mathbf{x}^\star)$, which leads to a contradiction. Consequently, in $T$ steps, we have visited at least an $\epsilon$-stationary point. $\square$

**Corollary 6.3.** *If we add a negative curvature descent candidate in each iteration, we can visit a $(\epsilon, \sqrt{\epsilon})$-second order stationary point using $\tilde{O}(\frac{\Delta\sqrt{L_2}n^2}{\epsilon^{1.5}})$ queries. Here we define a $(\epsilon, \sqrt{\epsilon})$-second order stationary point as a point $\mathbf{x}$ satisfying $\|\nabla f(\mathbf{x})\| \leq \epsilon$ and $\lambda_{min}(\nabla^2 f(\mathbf{x})) \leq -\sqrt{L_2\epsilon}$.*

The formal statement and proof can be found in Appendix E.5.

Note that we have used `Comparison-GE` in Theorem 2.1 as a subroutine to estimate the direction of the gradient. The query complexity of `Comparison-GE` is $O(n\log\frac{1}{\epsilon})$, which incurs an $O(n)$ overhead – intuitively, classical algorithms by comparisons are limited by the fact that we need $\Omega(n)$ comparisons to explore an $n$-dimensional space. An idea that can significantly reduce the query complexity of our algorithm is by replacing `Comparison-GE` with `Comparison-QGE` in Theorem 2.2, a quantum algorithm with $\log n$ dependence for gradient estimation. This implies the following quantum algorithm result:

**Corollary 6.4.** *In quantum setting, we refer to algorithm `Comparison-QGE` in Theorem 2.2 here as a basic gradient estimation algorithm and apply Theorem 6.2, then with success probability at least $2/3$, Algorithm 5 visits an $\epsilon$-stationary point with query complexity $O(\frac{\Delta\sqrt{L_2}n}{\epsilon^{1.5}} \log\left(\frac{nL_1L_2}{\epsilon}\right))$.*

## Impact Statement

This paper presents work whose goal is to advance the field of machine learning, in particular on the topic of optimization theory. As the paper is theoretical, we do not foresee any potential societal consequences to be highlighted here.

## Acknowledgements

We thank the anonymous reviewers for their constructive feedback. HW, XT, YZ, and TL were supported by the National Natural Science Foundation of China (Grant Number 62372006).

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

# A. Auxiliary Lemmas

In this appendix, we collect all auxiliary lemmas needed for our proofs.

## A.1. Distance between normalized vectors

**Lemma A.1.** *If* $\mathbf{v}, \mathbf{v}' \in \mathbb{R}^n$ *are two vectors such that* $\|\mathbf{v}\| \geq \gamma$ *and* $\|\mathbf{v} - \mathbf{v}'\| \leq \tau$, *we have*

$$\left\| \frac{\mathbf{v}}{\|\mathbf{v}\|} - \frac{\mathbf{v}'}{\|\mathbf{v}'\|} \right\| \leq \frac{2\tau}{\gamma}.$$

*Proof.* By the triangle inequality, we have

$$\left\| \frac{\mathbf{v}}{\|\mathbf{v}\|} - \frac{\mathbf{v}'}{\|\mathbf{v}'\|} \right\| \leq \left\| \frac{\mathbf{v}}{\|\mathbf{v}\|} - \frac{\mathbf{v}'}{\|\mathbf{v}\|} \right\| + \left\| \frac{\mathbf{v}'}{\|\mathbf{v}\|} - \frac{\mathbf{v}'}{\|\mathbf{v}'\|} \right\|$$

$$= \frac{\|\mathbf{v} - \mathbf{v}'\|}{\|\mathbf{v}\|} + \frac{|\|\mathbf{v}\| - \|\mathbf{v}'\|| \|\mathbf{v}'\|}{\|\mathbf{v}\| \|\mathbf{v}'\|}$$

$$\leq \frac{\tau}{\gamma} + \frac{\tau}{\gamma} = \frac{2\tau}{\gamma}.$$

$\square$

**Lemma A.2.** *If* $\mathbf{v}_1, \mathbf{v}_2 \in \mathbb{R}^n$ *are two vectors such that* $\|\mathbf{v}_1\|, \|\mathbf{v}_2\| \geq \gamma$, *and* $\mathbf{v}_1', \mathbf{v}_2' \in \mathbb{R}^n$ *are another two vectors such that* $\|\mathbf{v}_1 - \mathbf{v}_1'\|, \|\mathbf{v}_2 - \mathbf{v}_2'\| \leq \tau$ *where* $0 < \tau < \gamma$, *we have*

$$\left| \left\langle \frac{\mathbf{v}_1}{\|\mathbf{v}_1\|}, \frac{\mathbf{v}_2}{\|\mathbf{v}_2\|} \right\rangle - \left\langle \frac{\mathbf{v}_1'}{\|\mathbf{v}_1'\|}, \frac{\mathbf{v}_2'}{\|\mathbf{v}_2'\|} \right\rangle \right| \leq \frac{6\tau}{\gamma}.$$

*Proof.* By the triangle inequality, we have

$$\left| \left\langle \frac{\mathbf{v}_1}{\|\mathbf{v}_1\|}, \frac{\mathbf{v}_2}{\|\mathbf{v}_2\|} \right\rangle - \left\langle \frac{\mathbf{v}_1'}{\|\mathbf{v}_1'\|}, \frac{\mathbf{v}_2'}{\|\mathbf{v}_2'\|} \right\rangle \right| \leq \left| \left\langle \frac{\mathbf{v}_1}{\|\mathbf{v}_1\|}, \frac{\mathbf{v}_2}{\|\mathbf{v}_2\|} \right\rangle - \left\langle \frac{\mathbf{v}_1'}{\|\mathbf{v}_1\|}, \frac{\mathbf{v}_2'}{\|\mathbf{v}_2\|} \right\rangle \right| + \left| \left\langle \frac{\mathbf{v}_1'}{\|\mathbf{v}_1\|}, \frac{\mathbf{v}_2'}{\|\mathbf{v}_2\|} \right\rangle - \left\langle \frac{\mathbf{v}_1'}{\|\mathbf{v}_1'\|}, \frac{\mathbf{v}_2'}{\|\mathbf{v}_2'\|} \right\rangle \right|.$$

On the one hand, by the triangle inequality and the Cauchy-Schwarz inequality,

$$\left| \left\langle \frac{\mathbf{v}_1}{\|\mathbf{v}_1\|}, \frac{\mathbf{v}_2}{\|\mathbf{v}_2\|} \right\rangle - \left\langle \frac{\mathbf{v}_1'}{\|\mathbf{v}_1\|}, \frac{\mathbf{v}_2'}{\|\mathbf{v}_2\|} \right\rangle \right| \leq \frac{1}{\|\mathbf{v}_1\| \|\mathbf{v}_2\|} (|\langle \mathbf{v}_1, \mathbf{v}_2 \rangle - \langle \mathbf{v}_1, \mathbf{v}_2' \rangle| + |\langle \mathbf{v}_1, \mathbf{v}_2' \rangle - \langle \mathbf{v}_1', \mathbf{v}_2' \rangle|)$$

$$\leq \frac{\|\mathbf{v}_2 - \mathbf{v}_2'\|}{\|\mathbf{v}_2\|} + \frac{\|\mathbf{v}_1 - \mathbf{v}_1'\| \|\mathbf{v}_2'\|}{\|\mathbf{v}_1\| \|\mathbf{v}_2\|}$$

$$\leq \frac{\tau}{\gamma} + \frac{\tau(\gamma + \tau)}{\gamma^2}.$$

On the other hand, by the Cauchy-Schwarz inequality, $|\langle \mathbf{v}_1', \mathbf{v}_2' \rangle| \leq \|\mathbf{v}_1'\| \|\mathbf{v}_2'\|$, and hence

$$\left| \left\langle \frac{\mathbf{v}_1'}{\|\mathbf{v}_1\|}, \frac{\mathbf{v}_2'}{\|\mathbf{v}_2\|} \right\rangle - \left\langle \frac{\mathbf{v}_1'}{\|\mathbf{v}_1'\|}, \frac{\mathbf{v}_2'}{\|\mathbf{v}_2'\|} \right\rangle \right| = |\langle \mathbf{v}_1', \mathbf{v}_2' \rangle| \left| \frac{1}{\|\mathbf{v}_1\| \|\mathbf{v}_2\|} - \frac{1}{\|\mathbf{v}_1'\| \|\mathbf{v}_2'\|} \right|$$

$$\leq \left| \frac{\|\mathbf{v}_1'\| \|\mathbf{v}_2'\|}{\|\mathbf{v}_1\| \|\mathbf{v}_2\|} - 1 \right|$$

$$\leq \left( \frac{\gamma + \tau}{\gamma} \right)^2 - 1.$$

In all, due to $\tau < \gamma$,

$$\left| \left\langle \frac{\mathbf{v}_1}{\|\mathbf{v}_1\|}, \frac{\mathbf{v}_2}{\|\mathbf{v}_2\|} \right\rangle - \left\langle \frac{\mathbf{v}_1'}{\|\mathbf{v}_1'\|}, \frac{\mathbf{v}_2'}{\|\mathbf{v}_2'\|} \right\rangle \right| \leq \frac{\tau}{\gamma} + \frac{\tau(\gamma + \tau)}{\gamma^2} + \left( \frac{\gamma + \tau}{\gamma} \right)^2 - 1 = \frac{2\tau(2\gamma + \tau)}{\gamma^2} \leq \frac{6\tau}{\gamma}.$$

$\square$

**Lemma A.3.** *For any nonzero vectors* $\mathbf{v}, \mathbf{g} \in \mathbb{R}^n$,

$$\sqrt{\frac{1 - \left\langle \frac{\mathbf{v}+\mathbf{g}}{\|\mathbf{v}+\mathbf{g}\|}, \frac{\mathbf{v}}{\|\mathbf{v}\|} \right\rangle^2}{1 - \left\langle \frac{\mathbf{v}-\mathbf{g}}{\|\mathbf{v}-\mathbf{g}\|}, \frac{\mathbf{v}}{\|\mathbf{v}\|} \right\rangle^2}} = \frac{\|\mathbf{v} - \mathbf{g}\|}{\|\mathbf{v} + \mathbf{g}\|}.$$

*Proof.* We have

$$\frac{1 - \left\langle \frac{\mathbf{v}+\mathbf{g}}{\|\mathbf{v}+\mathbf{g}\|}, \frac{\mathbf{v}}{\|\mathbf{v}\|} \right\rangle^2}{1 - \left\langle \frac{\mathbf{v}-\mathbf{g}}{\|\mathbf{v}-\mathbf{g}\|}, \frac{\mathbf{v}}{\|\mathbf{v}\|} \right\rangle^2} \cdot \frac{\|\mathbf{v} + \mathbf{g}\|^2}{\|\mathbf{v} - \mathbf{g}\|^2} = \frac{\|\mathbf{v} + \mathbf{g}\|^2 - \left\langle \mathbf{v} + \mathbf{g}, \frac{\mathbf{v}}{\|\mathbf{v}\|} \right\rangle^2}{\|\mathbf{v} - \mathbf{g}\|^2 - \left\langle \mathbf{v} - \mathbf{g}, \frac{\mathbf{v}}{\|\mathbf{v}\|} \right\rangle^2}$$

$$= \frac{\langle \mathbf{v} + \mathbf{g}, \mathbf{v} + \mathbf{g} \rangle - (\|\mathbf{v}\| + \frac{\langle \mathbf{v}, \mathbf{g} \rangle}{\|\mathbf{v}\|})^2}{\langle \mathbf{v} - \mathbf{g}, \mathbf{v} - \mathbf{g} \rangle - (\|\mathbf{v}\| - \frac{\langle \mathbf{v}, \mathbf{g} \rangle}{\|\mathbf{v}\|})^2}$$

$$= \frac{\|\mathbf{v}\|^2 + \|\mathbf{g}\|^2 + 2\langle \mathbf{v}, \mathbf{g} \rangle - (\|\mathbf{v}\|^2 + 2\langle \mathbf{v}, \mathbf{g} \rangle + \frac{\langle \mathbf{v}, \mathbf{g} \rangle^2}{\|\mathbf{v}\|^2})}{\|\mathbf{v}\|^2 + \|\mathbf{g}\|^2 - 2\langle \mathbf{v}, \mathbf{g} \rangle - (\|\mathbf{v}\|^2 - 2\langle \mathbf{v}, \mathbf{g} \rangle + \frac{\langle \mathbf{v}, \mathbf{g} \rangle^2}{\|\mathbf{v}\|^2})}$$

$$= 1.$$

$\square$

## A.2. Distance between normalized matrices

**Lemma A.4** (Normalization stability). *Let* $A, B \in \mathbb{R}^{n \times n}$ *be nonzero and assume* $\|A - B\| \leq \rho < \frac{1}{2}\|B\|$. *Then*

$$\left\| \frac{A}{\|A\|} - \frac{B}{\|B\|} \right\| \leq \frac{4\rho}{\|B\|}.$$

*Proof.* Write

$$\frac{A}{\|A\|} - \frac{B}{\|B\|} = \frac{A - B}{\|A\|} + B\left(\frac{1}{\|A\|} - \frac{1}{\|B\|}\right).$$

Since $\|A\| \geq \|B\| - \|A - B\| \geq \|B\| - \rho \geq \frac{1}{2}\|B\|$, we have $\|A\|^{-1} \leq 2\|B\|^{-1}$. Also,

$$\left| \frac{1}{\|A\|} - \frac{1}{\|B\|} \right| = \frac{|\|A\| - \|B\||}{\|A\|\|B\|} \leq \frac{\rho}{(\frac{1}{2}\|B\|)\|B\|} = \frac{2\rho}{\|B\|^2}.$$

Therefore,

$$\left\| \frac{A}{\|A\|} - \frac{B}{\|B\|} \right\| \leq \frac{\|A - B\|}{\|A\|} + \|B\| \left| \frac{1}{\|A\|} - \frac{1}{\|B\|} \right| \leq \frac{\rho}{\frac{1}{2}\|B\|} + \|B\| \frac{2\rho}{\|B\|^2} = \frac{4\rho}{\|B\|}.$$

$\square$

## A.3. Column norm bound implies spectral norm bound

**Lemma A.5.** *Let* $E \in \mathbb{R}^{n \times n}$. *If* $\|E\mathbf{e}_i\| \leq \varepsilon_c$ *for all* $i = 1, \ldots, n$, *then*

$$\|E\| \leq \varepsilon_c \sqrt{n}.$$

*Proof.* For any $\mathbf{v} \in \mathbb{R}^n$ with $\|\mathbf{v}\| = 1$,

$$E\mathbf{v} = \sum_{i=1}^{n} v_i (E\mathbf{e}_i),$$

so

$$\|E\mathbf{v}\| \leq \sum_{i=1}^{n} |v_i| \|E\mathbf{e}_i\| \leq \varepsilon_c \sum_{i=1}^{n} |v_i| \leq \varepsilon_c \sqrt{n} \left( \sum_{i=1}^{n} v_i^2 \right)^{1/2} = \varepsilon_c \sqrt{n}.$$

Taking the supremum over $\|\mathbf{v}\| = 1$ yields $\|E\| \leq \varepsilon_c \sqrt{n}$.

$\square$

### A.4. Inner Product Concentration for Random Vectors on Sphere

In this subsection, we give a lemma of the inner product concentration for random vectors on sphere proved in (Tao et al., 2026), stated below:

**Lemma A.6** (Lemma 3 of (Tao et al., 2026)). *Let $n \geq 5$. For any $\mathbf{x} \in \mathbb{R}^n$, $\mathbf{x} \neq \mathbf{0}$, and any constant $c > 0$, there exists constant $p_1$ and $p_2$ which is independent of $n$, such that*

$$p_1 \leq \Pr_{\mathbf{y} \sim S_n}\left[|\langle \mathbf{y}, \mathbf{x} \rangle| \leq \|\mathbf{x}\|/(c\sqrt{n})\right] \leq p_2.$$

*where $\mathbf{y}$ is chosen from $S_n$ uniformly at random. In particular, we have the inequality below:*

$$\Pr_{\mathbf{y} \sim S_n}\left[|\langle \mathbf{y}, \mathbf{x} \rangle| \leq \|\mathbf{x}\| \cdot \frac{24}{25\sqrt{n}}\right] \geq 3/5 \tag{9}$$

## B. Proof Details of Hessian-Vector Product Estimation by Comparisons

In this section, we analyze Algorithm 1 and present the proof of Theorem 3.1.

*Proof of Theorem 3.1.* Denote $\mathbf{x}_+ := \mathbf{x} + r_0\mathbf{y}$ and $\mathbf{x}_- := \mathbf{x} - r_0\mathbf{y}$. Since $f$ has $L_2$-Lipschitz Hessian,

$$\left\|\nabla f(\mathbf{x}_+) - \nabla f(\mathbf{x}) - r_0\nabla^2 f(\mathbf{x}) \cdot \mathbf{y}\right\| \leq L_2 r_0^2/2; \tag{10}$$

$$\left\|\nabla f(\mathbf{x}_-) - \nabla f(\mathbf{x}) + r_0\nabla^2 f(\mathbf{x}) \cdot \mathbf{y}\right\| \leq L_2 r_0^2/2. \tag{11}$$

Therefore, we have $\|\nabla f(\mathbf{x}_+) + \nabla f(\mathbf{x}_-) - 2\nabla f(\mathbf{x})\| \leq L_2 r_0^2$ and $\left\|\nabla^2 f(\mathbf{x}) \cdot \mathbf{y} - \frac{\nabla f(\mathbf{x}_+) - \nabla f(\mathbf{x}_-)}{2r_0}\right\| \leq L_2 r_0/2$. Furthermore, as $r_0 \leq \gamma_{\mathbf{x}}/(100L_1)$ and $f$ has $L_1$-Lipschitz gradient, both $\|\nabla f(\mathbf{x}_+)\|$ and $\|\nabla f(\mathbf{x}_-)\|$ is at least $\gamma_{\mathbf{x}} - L_1\gamma_{\mathbf{x}}/(100L_1) = 0.99\gamma_{\mathbf{x}}$. We first understand how to approximate $\nabla^2 f(\mathbf{x}) \cdot \mathbf{y}$ by normalized vectors $\frac{\nabla f(\mathbf{x})}{\|\nabla f(\mathbf{x})\|}, \frac{\nabla f(\mathbf{x}_+)}{\|\nabla f(\mathbf{x}_+)\|}, \frac{\nabla f(\mathbf{x}_-)}{\|\nabla f(\mathbf{x}_-)\|}$, and then analyze the approximation error due to using $\hat{\mathbf{g}}_0, \hat{\mathbf{g}}_1, \hat{\mathbf{g}}_{-1}$, respectively. By Lemma A.3, we have

$$\frac{1}{2\|\nabla f(\mathbf{x})\|} \frac{\|\nabla f(\mathbf{x}) - r_0\nabla^2 f(\mathbf{x}) \cdot \mathbf{y}\|}{\sqrt{1 - \left\langle \frac{\nabla f(\mathbf{x}) + r_0\nabla^2 f(\mathbf{x}) \cdot \mathbf{y}}{\|\nabla f(\mathbf{x}) + r_0\nabla^2 f(\mathbf{x}) \cdot \mathbf{y}\|}, \frac{\nabla f(\mathbf{x})}{\|\nabla f(\mathbf{x})\|} \right\rangle^2}}$$
$$= \frac{1}{2\|\nabla f(\mathbf{x})\|} \frac{\|\nabla f(\mathbf{x}) + r_0\nabla^2 f(\mathbf{x}) \cdot \mathbf{y}\|}{\sqrt{1 - \left\langle \frac{\nabla f(\mathbf{x}) - r_0\nabla^2 f(\mathbf{x}) \cdot \mathbf{y}}{\|\nabla f(\mathbf{x}) - r_0\nabla^2 f(\mathbf{x}) \cdot \mathbf{y}\|}, \frac{\nabla f(\mathbf{x})}{\|\nabla f(\mathbf{x})\|} \right\rangle^2}}.$$

We denote the value above as $\alpha$. Because $f$ is $L_2$-Hessian Lipschitz, $\|r_0\nabla^2 f(\mathbf{x}) \cdot \mathbf{y}\| \leq r_0 L_2$. Since $r_0 \leq \frac{\gamma_{\mathbf{x}}}{100L_2}$, $\|r_0\nabla^2 f(\mathbf{x}) \cdot \mathbf{y}\| \leq \frac{\gamma_{\mathbf{x}}}{100}$. Also by Lemma A.2 we have

$$\left\langle \frac{\nabla f(\mathbf{x}) + r_0\nabla^2 f(\mathbf{x}) \cdot \mathbf{y}}{\|\nabla f(\mathbf{x}) + r_0\nabla^2 f(\mathbf{x}) \cdot \mathbf{y}\|}, \frac{\nabla f(\mathbf{x})}{\|\nabla f(\mathbf{x})\|} \right\rangle \geq 0.94,$$

$$\left\langle \frac{\nabla f(\mathbf{x}) - r_0\nabla^2 f(\mathbf{x}) \cdot \mathbf{y}}{\|\nabla f(\mathbf{x}) - r_0\nabla^2 f(\mathbf{x}) \cdot \mathbf{y}\|}, \frac{\nabla f(\mathbf{x})}{\|\nabla f(\mathbf{x})\|} \right\rangle \geq 0.94,$$

which promises that $\alpha \geq \frac{0.99}{2\sqrt{1 - 0.94^2}} \geq 1$. In arguments next, we say a vector $\mathbf{u}$ is $\xi$-close to a vector $\mathbf{v}$ if $\|\mathbf{u} - \mathbf{v}\| \leq \xi$ for any $\xi \geq 0$. We prove that the vector

$$\tilde{\mathbf{g}}_1 := \frac{\nabla f(\mathbf{x})}{\|\nabla f(\mathbf{x})\|} + \alpha \cdot \left( \sqrt{1 - \left\langle \frac{\nabla f(\mathbf{x}_-)}{\|\nabla f(\mathbf{x}_-)\|}, \frac{\nabla f(\mathbf{x})}{\|\nabla f(\mathbf{x})\|} \right\rangle^2} \frac{\nabla f(\mathbf{x}_+)}{\|\nabla f(\mathbf{x}_+)\|} \right.$$
$$\left. - \sqrt{1 - \left\langle \frac{\nabla f(\mathbf{x}_+)}{\|\nabla f(\mathbf{x}_+)\|}, \frac{\nabla f(\mathbf{x})}{\|\nabla f(\mathbf{x})\|} \right\rangle^2} \frac{\nabla f(\mathbf{x}_-)}{\|\nabla f(\mathbf{x}_-)\|} \right) \tag{12}$$

is $\frac{7L_2r_0^2}{\gamma_{\mathbf{x}}}$-close to a vector proportional to $\nabla f(\mathbf{x}_+)$. This is because Eq. (10), Eq. (11), and Lemma A.1 imply that $\frac{\nabla f(\mathbf{x}_+)}{\|\nabla f(\mathbf{x}_+)\|}$ and $\frac{\nabla f(\mathbf{x})+r_0\nabla^2 f(\mathbf{x})\cdot\mathbf{y}}{\|\nabla f(\mathbf{x})+r_0\nabla^2 f(\mathbf{x})\cdot\mathbf{y}\|}$ are $\frac{L_2r_0^2}{0.99\gamma_{\mathbf{x}}}$-close to each other,

$$\sqrt{1-\left\langle\frac{\nabla f(\mathbf{x}_-)}{\|\nabla f(\mathbf{x}_-)\|},\frac{\nabla f(\mathbf{x})}{\|\nabla f(\mathbf{x})\|}\right\rangle^2}\frac{\nabla f(\mathbf{x}_+)}{\|\nabla f(\mathbf{x}_+)\|} \tag{13}$$

is proportional to $\nabla f(\mathbf{x}_+)$, and the definition of $\alpha$ implies

$$\frac{\nabla f(\mathbf{x})}{\|\nabla f(\mathbf{x})\|}-\alpha\cdot\frac{\nabla f(\mathbf{x})-r_0\nabla^2 f(\mathbf{x})\cdot\mathbf{y}}{\|\nabla f(\mathbf{x})-r_0\nabla^2 f(\mathbf{x})\cdot\mathbf{y}\|}\cdot\sqrt{1-\left\langle\frac{\nabla f(\mathbf{x})+r_0\nabla^2 f(\mathbf{x})\cdot\mathbf{y}}{\|\nabla f(\mathbf{x})+r_0\nabla^2 f(\mathbf{x})\cdot\mathbf{y}\|},\frac{\nabla f(\mathbf{x})}{\|\nabla f(\mathbf{x})\|}\right\rangle^2}$$
$$=\nabla f(\mathbf{x})+r_0\nabla^2 f(\mathbf{x})\cdot\mathbf{y}/(2\|\nabla f(\mathbf{x})\|). \tag{14}$$

The above vector is $\frac{L_2r_0^2}{4\gamma_{\mathbf{x}}}$-close to $\frac{\nabla f(\mathbf{x}_+)}{2\|\nabla f(\mathbf{x})\|}$ by Eq. (10), and the error in above steps cumulates by at most $\frac{6L_2r_0^2}{0.99\gamma_{\mathbf{x}}}$ using Lemma A.2. In total, the error is at most $\frac{6L_2r_0^2}{0.99\gamma_{\mathbf{x}}}+\frac{L_2r_0^2}{4\gamma_{\mathbf{x}}}\leq\frac{7L_2r_0^2}{\gamma_{\mathbf{x}}}$. Furthermore, this vector proportional to $\nabla f(\mathbf{x}_+)$ that is $\frac{L_2r_0^2}{4\gamma_{\mathbf{x}}}$-close to Eq. (12) has norm at least $(1-0.01)/2=0.495$ because the coefficient in Eq. (13) is positive, while in the equality above we have $\|r_0\nabla^2 f(\mathbf{x})\cdot\mathbf{y}\|\leq\frac{\gamma_{\mathbf{x}}}{100}$. Therefore, applying Lemma A.1, the vector $\tilde{\mathbf{g}}_1$ in Eq. (12) satisfies

$$\left\|\tilde{\mathbf{g}}_1/\|\tilde{\mathbf{g}}_1\|-\nabla f(\mathbf{x}_+)/\|\nabla f(\mathbf{x}_+)\|\right\|\leq 29L_2r_0^2/\gamma_{\mathbf{x}}. \tag{15}$$

Following the same proof, we can prove that the vector

$$\tilde{\mathbf{g}}_{-1}:=\frac{\nabla f(\mathbf{x})}{\|\nabla f(\mathbf{x})\|}-\alpha\sqrt{1-\left\langle\frac{\nabla f(\mathbf{x}_-)}{\|\nabla f(\mathbf{x}_-)\|},\frac{\nabla f(\mathbf{x})}{\|\nabla f(\mathbf{x})\|}\right\rangle^2}\frac{\nabla f(\mathbf{x}_+)}{\|\nabla f(\mathbf{x}_+)\|}$$
$$+\alpha\sqrt{1-\left\langle\frac{\nabla f(\mathbf{x}_+)}{\|\nabla f(\mathbf{x}_+)\|},\frac{\nabla f(\mathbf{x})}{\|\nabla f(\mathbf{x})\|}\right\rangle^2}\frac{\nabla f(\mathbf{x}_-)}{\|\nabla f(\mathbf{x}_-)\|} \tag{16}$$

satisfies

$$\left\|\tilde{\mathbf{g}}_{-1}/\|\tilde{\mathbf{g}}_{-1}\|-\nabla f(\mathbf{x}_-)/\|\nabla f(\mathbf{x}_-)\|\right\|\leq 29L_2r_0^2/\gamma_{\mathbf{x}}. \tag{17}$$

Furthermore, Eq. (14) implies that the vector $\tilde{\mathbf{g}}_1-\tilde{\mathbf{g}}_{-1}$ is $\frac{14L_2r_0^2}{\gamma_{\mathbf{x}}}$-close to

$$\frac{\nabla f(\mathbf{x})+r_0\nabla^2 f(\mathbf{x})\cdot\mathbf{y}}{2\|\nabla f(\mathbf{x})\|}-\frac{\nabla f(\mathbf{x})-r_0\nabla^2 f(\mathbf{x})\cdot\mathbf{y}}{2\|\nabla f(\mathbf{x})\|}$$
$$=r_0\nabla^2 f(\mathbf{x})\cdot\mathbf{y}/\|\nabla f(\mathbf{x})\|. \tag{18}$$

Given that $\lambda_{\min}(\nabla^2 f(\mathbf{x}))\leq-\sqrt{L_2\epsilon}$ and $|y_1|\geq\gamma_{\mathbf{y}}$, $\|\nabla^2 f(\mathbf{x})\cdot\mathbf{y}\|\geq\sqrt{L_2\epsilon}\gamma_{\mathbf{y}}$. Therefore, the RHS of Eq. (18) has norm at least $\frac{r_0\sqrt{L_2\epsilon}\gamma_{\mathbf{y}}}{\gamma_{\mathbf{x}}}$, and by Lemma A.1 we have

$$\left\|\frac{\tilde{\mathbf{g}}_1-\tilde{\mathbf{g}}_{-1}}{\|\tilde{\mathbf{g}}_1-\tilde{\mathbf{g}}_{-1}\|}-\frac{\nabla^2 f(\mathbf{x})\cdot\mathbf{y}}{\|\nabla^2 f(\mathbf{x})\cdot\mathbf{y}\|}\right\|\leq\frac{14L_2r_0^2}{\gamma_{\mathbf{x}}}\bigg/\bigg(\frac{r_0\sqrt{L_2\epsilon}\gamma_{\mathbf{y}}}{\gamma_{\mathbf{x}}}\bigg)=\frac{14r_0\sqrt{L_2}}{\sqrt{\epsilon}\gamma_{\mathbf{y}}}. \tag{19}$$

Finally, by Theorem 2.1 and our choice of the precision parameter, the error terms coming from running ComparisonGE, $\left\|\hat{\mathbf{g}}_0-\frac{\nabla f(\mathbf{x})}{\|\nabla f(\mathbf{x})\|}\right\|$, $\left\|\hat{\mathbf{g}}_1-\frac{\nabla f(\mathbf{x}_+)}{\|\nabla f(\mathbf{x}_+)\|}\right\|$, and $\left\|\hat{\mathbf{g}}_{-1}-\frac{\nabla f(\mathbf{x}_-)}{\|\nabla f(\mathbf{x}_-)\|}\right\|$, are all upper bounded by $\frac{L_2r_0^2}{\gamma_{\mathbf{x}}}$. Combined with Eq. (15) and Eq. (17), we know that the vector $\mathbf{g}$ we obtained in Algorithm 1 is $\frac{61L_2r_0^2}{\gamma_{\mathbf{x}}}$ close to $\frac{\tilde{\mathbf{g}}_1-\tilde{\mathbf{g}}_{-1}}{2\alpha}$. Since $\alpha\geq 1$, by Lemma A.1 we have

$$\left\|\frac{\mathbf{g}}{\|\mathbf{g}\|}-\frac{\tilde{\mathbf{g}}_1-\tilde{\mathbf{g}}_{-1}}{\|\tilde{\mathbf{g}}_1-\tilde{\mathbf{g}}_{-1}\|}\right\|\leq\frac{61L_2r_0^2}{\gamma_{\mathbf{x}}}. \tag{20}$$

In total, all the errors we have are Eq. (19) and Eq. (20):

$$\left\| \frac{\mathbf{g}}{\|\mathbf{g}\|} - \frac{\nabla^2 f(\mathbf{x}) \cdot \mathbf{y}}{\|\nabla^2 f(\mathbf{x}) \cdot \mathbf{y}\|} \right\| \leq \frac{61 L_2 r_0^2}{\gamma_{\mathbf{x}}} + \frac{14 r_0 \sqrt{L_2}}{\sqrt{\epsilon} \gamma_{\mathbf{y}}}. \tag{21}$$

Our choice of $r_0 = \min\left\{ \frac{\gamma_{\mathbf{x}}}{100 L_1}, \frac{\gamma_{\mathbf{x}}}{100 L_2}, \frac{\sqrt{\gamma_{\mathbf{x}} \hat{\delta}}}{20\sqrt{L_2}}, \frac{\gamma_{\mathbf{y}} \hat{\delta} \sqrt{\epsilon}}{20\sqrt{L_2}} \right\}$ guarantees that Eq. (21) is at most $\hat{\delta}$. In terms of query complexity, we make 3 calls to ComparisonGE. By Theorem 2.1 and that our precision is $L_2 r_0^2 / \gamma_{\mathbf{x}} = \Omega(\gamma_{\mathbf{x}} \gamma_{\mathbf{y}}^2 \epsilon \hat{\delta}^2 / (L_2 L_1^2))$, the total query complexity is $O\left( n \log \left( n L_2 L_1^2 / \gamma_{\mathbf{x}} \gamma_{\mathbf{y}}^2 \epsilon \hat{\delta}^2 \right) \right)$. $\qquad\square$

## C. Proof Details of Robust Hessian Estimation by Comparisons

In this section, we provide complete statements and proofs for the guaranties claimed in Section 4. Throughout, we assume that $\|H\mathbf{e}_1\| = \max_i \|H\mathbf{e}_i\| > 0$. We can make this assumption since we can find a column with maximum norm after obtaining all the column norm ratios, and it suffices to denote this column as index 1.

### C.1. Notations

Let $H \in \mathbb{R}^{n \times n}$ be symmetric, $H \neq 0$, with columns $H = (\mathbf{h}_1, \dots, \mathbf{h}_n)$, $\mathbf{h}_i := H\mathbf{e}_i$. Define

$$r_i := \frac{\|\mathbf{h}_i\|}{\|\mathbf{h}_1\|} \in [0, 1], \qquad r_1 = 1.$$

When $\mathbf{h}_i \neq 0$, define the unit column direction $u_i := \mathbf{h}_i / \|\mathbf{h}_i\|$. When $\mathbf{h}_1 + \mathbf{h}_i \neq 0$, define the unit sum direction $u_{1i} := (\mathbf{h}_1 + \mathbf{h}_i) / \|\mathbf{h}_1 + \mathbf{h}_i\|$.

Define the scale-free target matrix

$$H^\star := [r_1 \mathbf{u}_1, \dots, r_n \mathbf{u}_n] \in \mathbb{R}^{n \times n}. \tag{22}$$

Then $H = \|\mathbf{h}_1\| H^\star$ and hence $H/\|H\| = H^\star / \|H^\star\|$.

**Lemma C.1.** *Let $H^\star$ be defined by Eq. (22). Then $\|H^\star\| \geq 1$.*

*Proof.* Because $r_1 = 1$ and $\mathbf{u}_1$ is a unit vector, the first column of $H^\star$ equals $H^\star \mathbf{e}_1 = \mathbf{u}_1$ and thus

$$\|H^\star\| \geq \|H^\star \mathbf{e}_1\| = \|\mathbf{u}_1\| = 1.$$

$\qquad\square$

### C.2. Error bound of column vector estimation

In this subsection, we give a detailed proof of the fact that the small perturbation $\mathbf{h}_i + \rho \mathbf{h}_j$ can only affect the estimation of $\mathbf{h}_i$ under a bounded error when $\rho$ is bounded.

**Lemma C.2.** *Our choice of $\mathbf{g}_i$ and $\mathbf{g}_{1i}$ in Algorithm 2, Line 7, Line 10, Line 14 and Line 17 satisfies $\|\mathbf{g}_i - \mathbf{u}_i\| \leq \eta$ and $\|\mathbf{g}_{1i} - \mathbf{u}_{1i}\| \leq \eta$.*

*Proof.* We analyze $\mathbf{g}_i$ here, $\mathbf{g}_{1i}$ case is similar. Assume that $\mathbf{v}$ is an eigenvector of $H$ corresponding to minimum eigenvalue, and $|\langle \mathbf{e}_t, \mathbf{v} \rangle| \geq \frac{1}{\sqrt{n}}$. Fix an index $i$. Denote

$$\mathbf{y}_i^0 = \mathbf{e}_i, \ \mathbf{y}_i^1 = \mathbf{e}_i + \sigma \mathbf{e}_t, \ \mathbf{y}_i^2 = \mathbf{e}_i - \sigma \mathbf{e}_t.$$

Observe that for any $j \neq k \in \{0, 1, 2\}$,

$$|\langle \mathbf{y}_i^j - \mathbf{y}_i^k, \mathbf{v} \rangle| \geq \frac{\sigma}{\sqrt{n}}.$$

This implies at least two of three queries satisfies

$$|\langle \mathbf{y}_i^j, \mathbf{v} \rangle| \geq \frac{\sigma}{2\sqrt{n}}. \tag{23}$$

For each index $j$ satisfies Eq. (23), by taking $\gamma_\mathbf{y} = \frac{\sigma}{2\sqrt{n}}$, $\gamma_\mathbf{x} = \epsilon$, accuracy $\hat{\delta} = \frac{\eta}{4}$ in Algorithm 1, we know at least two queries in the setting $\{\mathbf{y}_i^0, \mathbf{y}_i^1, \mathbf{y}_i^2\}$ satisfies the condition $|\langle \mathbf{y}_i^j, \mathbf{v}\rangle| \geq \gamma_\mathbf{y}$. Therefore, according to Theorem 3.1, at least two of three inequalities hold:

$$\left\| \mathbf{g}_i^0 - \frac{\mathbf{h}_i}{\|\mathbf{h}_i\|} \right\| \leq \frac{\eta}{4}, \quad \left\| \mathbf{g}_i^+ - \frac{\mathbf{h}_i + \sigma\mathbf{h}_t}{\|\mathbf{h}_i + \sigma\mathbf{h}_t\|} \right\| \leq \frac{\eta}{4}, \quad \left\| \mathbf{g}_i^- - \frac{\mathbf{h}_i - \sigma\mathbf{h}_t}{\|\mathbf{h}_i - \sigma\mathbf{h}_t\|} \right\| \leq \frac{\eta}{4}.$$

In non-degenerate case, our choice $(\mathbf{v}_1, \mathbf{v}_2) \leftarrow \arg\max_{\mathbf{v}_1 \neq \mathbf{v}_2 \in \{\mathbf{g}_i^0, \mathbf{g}_i^+, \mathbf{g}_i^-\}} \langle \mathbf{v}_1, \mathbf{v}_2 \rangle$ tells $\mathbf{v}_1$ and $\mathbf{v}_2$ are two success queries to Algorithm 1, take $\mathbf{v}_1 = \mathbf{g}_i^+$, $\mathbf{v}_2 = \mathbf{g}_i^-$ for example, other case is the same, from Lemma A.1 we have:

$$\left\| \frac{\mathbf{v}_1 + \mathbf{v}_2}{\|\mathbf{v}_1 + \mathbf{v}_2\|} - \frac{\mathbf{h}_i}{\|\mathbf{h}_i\|} \right\| \leq \left\| \mathbf{g}_i^+ - \frac{\mathbf{h}_i + \sigma\mathbf{h}_t}{\|\mathbf{h}_i + \sigma\mathbf{h}_t\|} \right\| + \left\| \mathbf{g}_i^- - \frac{\mathbf{h}_i - \sigma\mathbf{h}_t}{\|\mathbf{h}_i - \sigma\mathbf{h}_t\|} \right\| + \left\| \frac{\mathbf{h}_i + \sigma\mathbf{h}_t}{\|\mathbf{h}_i + \sigma\mathbf{h}_t\|} - \frac{\mathbf{h}_i - \sigma\mathbf{h}_t}{\|\mathbf{h}_i - \sigma\mathbf{h}_t\|} \right\|$$
$$\leq \frac{\eta}{4} + \frac{\eta}{4} + \frac{2\eta}{4} = \eta.$$

Here the last inequality holds because $\langle \mathbf{g}_i^+, \mathbf{g}_i^- \rangle \geq 1 - \tau_\alpha$ tell us $\|\mathbf{h}_i\|/\|\sigma\mathbf{h}_t\| \geq \frac{4}{\eta}$.

In the degenerate case $\langle \mathbf{g}_i^+, \mathbf{g}_i^- \rangle \leq 1 - \tau_\alpha$, we have $\|\mathbf{h}_i\|/\|\sigma\mathbf{h}_t\| \leq \frac{4}{\eta}$, i.e.,

$$\|\mathbf{h}_i\| \leq \frac{\eta}{32\sqrt{n}}.$$

In this case, taking $\mathbf{g}_i = 0$ guarantees error on this column is $O\left(\frac{\eta}{\sqrt{n}}\right)$. $\qquad\square$

### C.3. Stability in the non-degenerate branch

In this subsection, we give a deterministic bound for the branch that directly solves Eq. (25) without perturbation.

**Assumption C.3** (Non-degenerate branch). Fix an index $i \geq 2$ such that $\mathbf{h}_i \neq \mathbf{0}$ and $\mathbf{h}_1 + \mathbf{h}_i \neq \mathbf{0}$. Assume:

1. (Not too parallel) $\alpha_i \leq 1 - \gamma$ for some $\gamma \in (0, 1)$.

2. (Not too small) $r_i \geq r_{\min}$ for some $r_{\min} \in (0, 1]$.

Fix $i \geq 2$ with $\mathbf{h}_i \neq \mathbf{0}$ and $\mathbf{h}_1 + \mathbf{h}_i \neq \mathbf{0}$. Define

$$\alpha_i := \langle \mathbf{u}_i, \mathbf{u}_1 \rangle, \qquad \beta_i := \langle \mathbf{u}_{1i}, \mathbf{u}_1 \rangle.$$

As shown in the main text,

$$\beta_i = F(r_i, \alpha_i) := \frac{1 + r_i\alpha_i}{\sqrt{1 + r_i^2 + 2r_i\alpha_i}}. \tag{24}$$

Squaring yields the quadratic (used by the solver):

$$(\beta^2 - \alpha^2)r^2 + 2\alpha(\beta^2 - 1)r + (\beta^2 - 1) = 0. \tag{25}$$

**Lemma C.4** (Non-degenerate inversion stability). *Under Assumption C.3, let $(\hat{\alpha}, \hat{\beta})$ satisfy $|\hat{\alpha} - \alpha_i| \leq \varepsilon$ and $|\hat{\beta} - \beta_i| \leq \varepsilon$, with $\varepsilon \leq \gamma/10$. Let $\hat{r}$ be the (unique) solution in $[0, 1]$ of $\hat{\beta} = F(r, \hat{\alpha})$. Then*

$$|\hat{r} - r_i| \leq \frac{C_0}{r_{\min}\gamma} \varepsilon,$$

*where $C_0$ is a universal constant.*

*Proof.* By the mean value theorem,

$$|F(\hat{r}, \hat{\alpha}) - F(r_i, \hat{\alpha})| = |\frac{\partial F}{\partial r}(\xi, \hat{\alpha})| |\hat{r} - r_i|$$

for some $\xi$ between $\hat{r}$ and $r_i$. Therefore,

$$|\hat{r} - r_i| = \frac{|\hat{\beta} - F(r_i, \hat{\alpha})|}{|\frac{\partial F}{\partial r}(\xi, \hat{\alpha})|}.$$

We bound numerator and denominator respectively. By triangle inequality, we can bound the numerator as:

$$|\hat{\beta} - F(r_i, \hat{\alpha})| \leq |\hat{\beta} - \beta_i| + |F(r_i, \alpha_i) - F(r_i, \hat{\alpha})| \leq \epsilon + L_\alpha \epsilon, \tag{26}$$

where $L_\alpha$ is a Lipschitz constant of $F$ in $\alpha$ on the compact set $r \in [0, 1]$, $\alpha \in [-1, 1]$. We can bound $L_\alpha$ explicitly by computing $\partial F / \partial \alpha$ and taking a supremum; a crude universal bound $L_\alpha \leq 2$ suffices (indeed, one can check $|\partial F / \partial \alpha| \leq 2$ on this domain). Thus the numerator $\leq 3\epsilon$.

On the other hand, We have

$$\left| \frac{\partial F}{\partial r}(\xi, \hat{\alpha}) \right| = \frac{\xi(1 - \hat{\alpha}^2)}{(1 + \xi^2 + 2\xi\hat{\alpha})^{3/2}}.$$

Since $\xi$ lies between $\hat{r} \in [0, 1]$ and $r_i \geq r_{\min}$, we have $\xi \geq r_{\min}/2$ provided $\epsilon$ is small enough; we can ensure this by restricting to $\epsilon \leq \gamma/10$ and using continuity of the inverse map. Also, $\hat{\alpha} \leq \alpha_i + \epsilon \leq 1 - \gamma + \gamma/10 = 1 - \frac{9}{10}\gamma$, so

$$1 - \hat{\alpha}^2 = (1 - \hat{\alpha})(1 + \hat{\alpha}) \geq \left( \tfrac{9}{10}\gamma \right) \cdot 1 = \tfrac{9}{10}\gamma.$$

Therefore

$$|\frac{\partial F}{\partial r}(\xi, \hat{\alpha})| \geq \frac{(r_{\min}/2) \cdot (\frac{9}{10}\gamma)}{8} \geq c\, r_{\min}\gamma$$

for a universal constant $c > 0$. Combining numerator and denominator gives

$$|\hat{r} - r_i| \leq \frac{3\varepsilon}{c\, r_{\min}\gamma} = \frac{C_0}{r_{\min}\gamma} \varepsilon$$

with $C_0 := 3/c$. $\qquad\square$

**Corollary C.5** (Non-degenerate branch under oracle tolerance $\eta$). *Under Assumption C.3, Algorithm 2 computes $\widehat{\alpha}_i = \langle \mathbf{g}_i, \mathbf{g}_1 \rangle \leq 1 - \tau_\alpha$ and $\widehat{\beta}_i = \langle \mathbf{h}_i, \mathbf{g}_1 \rangle$ satisfying $|\widehat{\alpha}_i - \alpha_i| \leq 3\eta$ and $|\widehat{\beta}_i - \beta_i| \leq 3\eta$, and thus*

$$|\hat{r}_i - r_i| \leq \frac{C_1}{r_{\min}\tau_\alpha}\eta$$

*for a constant $C_1$.*

### C.4. Perturbation branch: detailed bias–amplification bound

Here we prove Lemma C.9 with explicit steps. In perturbation (but $\|\mathbf{h}_i\|$ is large enough) case, we can find an index $j$ such that the direction of $\mathbf{h}_j$ and the direction of $\mathbf{h}_1$ (so to $\mathbf{h}_i$) are sufficiently separated for the perturbation step. Note that if we cannot find the index $j$, it means that the rank of $H$ is 1 up to negligible error, which would be dealt with by line search step, so we have the following assumption.

**Assumption C.6** (Perturbation separation). Fix an index $i$ such that $\mathbf{h}_i \neq \mathbf{0}$ and $\mathbf{h}_1 + \mathbf{h}_i \neq \mathbf{0}$. Assume $r_i \geq r_{\min} > 0$. Algorithm 3 finds an index $j$ such that

$$1 - \left\langle \frac{\mathbf{h}_i + \rho\mathbf{h}_j}{\|\mathbf{h}_i + \rho\mathbf{h}_j\|}, \frac{\mathbf{h}_1}{\|\mathbf{h}_1\|} \right\rangle^2 \geq c_0\rho^2 \tag{27}$$

for some absolute $c_0 > 0$.

Define $\alpha := \langle \mathbf{v}, \mathbf{u}_1 \rangle$, $\mathbf{w} := \frac{\mathbf{u}_1 + r_i\mathbf{v}}{\|\mathbf{u}_1 + r_i\mathbf{v}\|}$, $\beta := \langle \mathbf{w}, \mathbf{u}_1 \rangle$. Then $\beta = F(r_i, \alpha)$. Algorithm 3 estimates $\mathbf{v}$ and $\mathbf{w}$ by oracle queries:

$$\hat{\mathbf{v}} := \widehat{\mathbf{u}}(\mathbf{e}_i + \rho\mathbf{e}_j; \eta), \qquad \hat{\mathbf{w}} := \widehat{\mathbf{u}}(\mathbf{e}_1 + \mathbf{e}_i + \rho\mathbf{e}_j; \eta),$$

and computes

$$\hat{\alpha} := \langle \hat{\mathbf{v}}, \mathbf{g}_1 \rangle, \qquad \hat{\beta} := \langle \hat{\mathbf{w}}, \mathbf{g}_1 \rangle,$$

then returns $\hat{r}$ as the solution in $[0, 1 + \rho]$ to $\hat{\beta} = F(\hat{r}, \hat{\alpha})$.

**Lemma C.7.** *Under Assumption C.6, there exists a constant $C$ such that for $\eta$ sufficiently small,*

$$|\hat{r} - r_i| \leq C \frac{\eta}{r_{\min}\rho^2}.$$

*Proof.* By Theorem 3.1, $\|\hat{\mathbf{v}} - \mathbf{v}\| \leq \eta$, $\|\hat{\mathbf{w}} - \mathbf{w}\| \leq \eta$, and also $\|\mathbf{g}_1 - \mathbf{u}_1\| \leq \eta$. Apply Lemma A.2 to obtain

$$|\hat{\alpha} - \alpha| \leq 3\eta, \qquad |\hat{\beta} - \beta| \leq 3\eta. \tag{28}$$

By $r_i \geq r_{\min}$, we get

$$\left| \frac{\partial F}{\partial r}(r_i, \alpha) \right| = \frac{r_i(1 - \alpha^2)}{(1 + r_i^2 + 2r_i\alpha)^{3/2}} \geq \frac{r_{\min}(1 - \alpha^2)}{8}.$$

Using the separation $1 - \alpha^2 \geq c_0\rho^2$, we obtain

$$\left| \frac{\partial F}{\partial r}(r_i, \alpha) \right| \geq \frac{r_{\min}c_0}{8}\rho^2 \tag{29}$$

Again by the mean value theorem,

$$|\hat{r} - r_i| = \frac{|\hat{\beta} - F(r_i, \hat{\alpha})|}{|\frac{\partial F}{\partial r}(\xi)|}$$

for some $\xi$ between $r_i$ and $\hat{r}$.

For $\eta$ sufficiently small, $\hat{\alpha}$ is close to $\alpha$ and thus $1 - \hat{\alpha}^2 \geq \frac{1}{2}(1 - \alpha^2) \geq \frac{1}{2}c_0\rho^2$, and $\xi$ remains in a compact interval containing $[r_{\min}/2, 1 + \rho]$. Therefore the lower bound Eq. (29) (possibly with a factor $1/2$) holds for $|\frac{\partial F}{\partial r}(\xi)|$ as well:

$$\left| \frac{\partial F}{\partial r}(\xi, \hat{\alpha}) \right| \geq \frac{r_{\min}c_0}{16}\rho^2.$$

Hence

$$|\hat{r} - r_i| \leq \frac{16}{r_{\min}c_0\rho^2} |\hat{\beta} - F(r_i, \hat{\alpha})|.$$

Similar to Eq. (26), we can bound error during estimating $\beta = F(r_i, \alpha)$:

$$|\hat{\beta} - F(r_i, \hat{\alpha})| \leq |\hat{\beta} - \beta| + |F(r_i, \alpha) - F(r_i, \hat{\alpha})| \leq 3\eta + 3L_\alpha\eta.$$

where $L_\alpha$ is a Lipschitz constant of $F$ in $\alpha$ on the compact set $r \in [0, 1]$, $\alpha \in [-1, 1]$. Conclude that the numerator is $\leq C'\eta$, and therefore

$$|\hat{r} - r_i| \leq C \frac{\eta}{r_{\min}\rho^2}.$$

$\square$

**Lemma C.8.** *Let $\mathbf{a}, \mathbf{c} \in \mathbb{R}^n$ with $\mathbf{a} \neq 0$. For $\rho \in (0, 1/2)$ define*

$$\mathbf{v} := \frac{\mathbf{a} + \rho\mathbf{c}}{\|\mathbf{a} + \rho\mathbf{c}\|}, \qquad \mathbf{u} := \frac{\mathbf{a}}{\|\mathbf{a}\|}.$$

*If $\|\mathbf{c}\| \leq \|\mathbf{h}_1\|$ and $\|\mathbf{a}\| \geq r_{\min}\|\mathbf{h}_1\|$, then*

$$\|\mathbf{v} - \mathbf{u}\| \leq \frac{4}{r_{\min}}\rho.$$

*Proof.* According to triangle inequality, $\|\mathbf{a} + \rho\mathbf{c}\| \geq \|\mathbf{a}\| - \rho\|\mathbf{c}\| \geq \|\mathbf{a}\| - \rho\|\mathbf{h}_1\|$. Using $\|\mathbf{a}\| \geq r_{\min}\|\mathbf{h}_1\|$ and $\rho \leq r_{\min}/2$, we get $\|\mathbf{a} + \rho\mathbf{c}\| \geq \frac{1}{2}\|\mathbf{a}\|$.

Now write

$$\mathbf{v} - \mathbf{u} = \frac{\mathbf{a} + \rho\mathbf{c}}{\|\mathbf{a} + \rho\mathbf{c}\|} - \frac{\mathbf{a}}{\|\mathbf{a}\|} = \mathbf{a}\left( \frac{1}{\|\mathbf{a} + \rho\mathbf{c}\|} - \frac{1}{\|\mathbf{a}\|} \right) + \frac{\rho}{\|\mathbf{a} + \rho\mathbf{c}\|}\mathbf{c}.$$

Hence

$$\|\mathbf{v} - \mathbf{u}\| \le \|\mathbf{a}\| \left| \frac{1}{\|\mathbf{a} + \rho\mathbf{c}\|} - \frac{1}{\|\mathbf{a}\|} \right| + \frac{\rho}{\|\mathbf{a} + \rho\mathbf{c}\|} \|\mathbf{c}\|.$$

Also

$$\left| \frac{1}{\|\mathbf{a} + \rho\mathbf{c}\|} - \frac{1}{\|\mathbf{a}\|} \right| = \frac{|\|\mathbf{a} + \rho\mathbf{c}\| - \|\mathbf{a}\||}{\|\mathbf{a} + \rho\mathbf{c}\|\|\mathbf{a}\|} \le \frac{\rho\|\mathbf{c}\|}{\|\mathbf{a} + \rho\mathbf{c}\|\|\mathbf{a}\|}.$$

Therefore

$$\|\mathbf{v} - \mathbf{u}\| \le \|\mathbf{a}\| \frac{\rho\|\mathbf{c}\|}{\|\mathbf{a} + \rho\mathbf{c}\|\|\mathbf{a}\|} + \frac{\rho}{\|\mathbf{a} + \rho\mathbf{c}\|} \|\mathbf{c}\| = \frac{2\rho\|\mathbf{c}\|}{\|\mathbf{a} + \rho\mathbf{c}\|} \le \frac{2\rho\|\mathbf{h}_1\|}{\frac{1}{2}\|\mathbf{a}\|} = \frac{4\rho}{r_{\min}}.$$

$\square$

**Lemma C.9.** *Under Assumption C.6 and assuming $r_i \ge r_{\min}$, the perturbation branch estimator satisfies*

$$|\hat{r} - r_i| \le \frac{1}{r_{min}} \left( C \frac{\eta}{\rho^2} + B\rho \right)$$

*for constants $C, B$ depending only on $c_0$.*

*Proof.* The first term $C\eta/\rho^2$ follows from Lemma C.7. The second term accounts for the fact that the perturbation uses $\mathbf{v}$ (direction of $\mathbf{h}_i + \rho\mathbf{h}_j$) rather than $\mathbf{u}_i$ (direction of $\mathbf{h}_i$). By Lemma C.8, $\|\mathbf{v} - \mathbf{u}_i\| \le O(\rho)$ (with explicit constant $4/r_{\min}$). On the non-degenerate set enforced by $1 - \alpha^2 \ge c_0\rho^2$, the mapping from input directions to the recovered ratio is Lipschitz; therefore this $O(\rho)$ direction change induces an $O(\rho)$ bias in $r$. Combining the two terms of error yields the stated bound. $\square$

## C.5. Error bound for $H/\|H\|$

We now complete the proof of the main theorem.

**Theorem C.10** (Full statement of Theorem 4.1). *For target point $x$, assuming that $\|H\| \ge \sqrt{L_2\epsilon}$ and $\|g\| \ge \epsilon$, run Algorithm 2 with tolerance $\eta = c_0\delta^4/n^2$ and perturbation $\rho = \eta^{1/3}$. Then with probability at least $2/3$, there exists a constant $K$ such that the output $\widehat{H}$ satisfies*

$$\left\| \widehat{H} - \frac{H}{\|H\|} \right\| \le K\sqrt{n}\,\eta^{1/4}.$$

*In particular, choosing $\eta \le (\delta/(K\sqrt{n}))^4$ guarantees $\|\widehat{H} - H/\|H\|\| \le \delta$. The query complexity of Algorithm 2 is $O(n^2 \log\left(\frac{nL_1L_2}{\epsilon\delta}\right))$.*

*Proof.* From Lemma A.6, we have $\Pr_{\mathbf{u}\sim\mathrm{Unif}(S_1(0))}[|\langle \mathbf{u}, \mathbf{v}\rangle| \ge 1/\sqrt{n}] \ge 2/5$ where $\mathbf{v}$ is the eigenvalue of $H$ with the largest absolute eigenvalue. Because of the isotropy of n coordinates, $\mathbf{v} \sim \mathrm{Unif}(S_1(0))$. So we have:

$$\Pr_{\mathbf{u}\sim\mathrm{Unif}(S_1(0))}[|\langle \mathbf{u}, \mathbf{v}\rangle| \ge 1/\sqrt{n}] = \Pr_{\mathbf{u}\sim\mathrm{Unif}(S_1(0)),\mathbf{v}\sim\mathrm{Unif}(S_1(0))}[|\langle \mathbf{u}, \mathbf{v}\rangle| \ge 1/\sqrt{n}]$$

$$= \Pr_{\mathbf{u}\sim\mathrm{Unif}(\{\mathbf{e}_1,\ldots,\mathbf{e}_n\}),\mathbf{v}\sim\mathrm{Unif}(S_1(0))}[|\langle \mathbf{u}, \mathbf{v}\rangle| \ge 1/\sqrt{n}]$$

$$= \Pr_{t\sim\mathrm{Unif}(\{1,\ldots,n\}),\mathbf{v}\sim\mathrm{Unif}(S_1(0))}[|\langle \mathbf{e}_t, \mathbf{v}\rangle| \ge 1/\sqrt{n}]$$

$$\ge 2/5.$$

Repeat for $O(1)$ times and our success probability can be more than $2/3$. Let $H^\star$ be the scale-free target matrix Eq. (22), so that $H/\|H\| = H^\star/\|H^\star\|$. Let $\widetilde{H} = [\hat{r}_1\mathbf{g}_1, \ldots, \hat{r}_n\mathbf{g}_n]$ and define $E := \widetilde{H} - H^\star$. Fix $i$. If $\hat{r}_i$ is produced by the non-degenerate solver, then Corollary C.5 yields

$$|\hat{r}_i - r_i| \le \frac{C}{r_{\min}\tau_\alpha}\eta$$

for some constant $C$. here $r_{\min}$ relates to $\tau_\beta$. Geometry computation tell us:

$$r_{\min} = \tan\left(\arccos(1 - \tau_\beta)\right).$$

When $\eta$ is sufficiently small, through Taylor expansion, $f(x) = \tan(\arccos(1-x)) \sim \sqrt{2x} + O(x^{3/2})$, i.e. $r_{\min} \sim \sqrt{\tau_\beta}$. If $\hat{r}_i$ is produced by perturbation, then Lemma C.9 yields

$$|\hat{r}_i - r_i| \le \frac{1}{r_{\min}}(C\,\frac{\eta}{\rho^2} + B\rho)$$

If $\hat{r}_i$ is set to 0, which means the algorithm enters this branch only when the column is treated as negligible, i.e. $r_i \le r_{\min}$. We have

$$|\hat{r}_i - r_i| \le r_{\min}.$$

Combine these three cases and take the best parameters $\rho = \Theta(\eta^{1/3})$, $\tau_\alpha = \tau_\beta = \Theta(\sqrt{\eta})$, we get:

$$|\hat{r}_i - r_i| \le O(\eta^{1/4})$$

Now in all cases,

$$\|E\mathbf{e}_i\| = \|\hat{r}_i\mathbf{g}_i - r_i\mathbf{u}_i\| \le \|(\hat{r}_i - r_i)\mathbf{g}_i\| + \|r_i(\mathbf{g}_i - \mathbf{u}_i)\| \le |\hat{r}_i - r_i|\cdot\|\mathbf{g}_i\| + r_i\eta \le 2|\hat{r}_i - r_i| + \eta,$$

since $\|\mathbf{g}_i\| \le 1 + \eta \le 2$ and $r_i \le 1$. Thus $\|E\mathbf{e}_i\| \le K_1\eta^{1/4}$ for a constant $K_1$.

By Lemma A.5,

$$\|E\| \le K_1\sqrt{n}\,\eta^{1/4}.$$

By Lemma C.1, $\|H^\star\| \ge 1$. If $\eta$ is small enough that $\|E\| \le 1/2$, then Lemma A.4 gives

$$\left\|\frac{\widetilde{H}}{\|\widetilde{H}\|} - \frac{H^\star}{\|H^\star\|}\right\| \le 4\|E\| \le 4K_1\sqrt{n}\,\eta^{1/4}.$$

Using $H/\|H\| = H^\star/\|H^\star\|$ yields the stated bound with $K := 4K_1$. $\qquad\square$

# D. Proof Details of Estimation of Ratio between Hessian Norm and Gradient Norm

We provide proof details of the claims in Section 5 in this appendix.

## D.1. Basic lemmas for estimating the ratio

We first introduce the core lemma in our proof:

**Lemma D.1** (Davis-Kahan Theorem, see e.g., Theorem 1 of Yu et al. 2015). *Let $A, \widetilde{A} \in \mathbb{R}^{d\times d}$ be two symmetric matrices satisfying $\|A - \widetilde{A}\| \le \xi$ for some $\xi > 0$. For any $a < b$, denote $S = \{v_1, \ldots, v_k\}$ and $\widetilde{S} = \{\tilde{v}_1, \ldots, \tilde{v}_k\}$ as the set of normalized eigenvectors of $A$ and $\widetilde{A}$ associated with eigenvalues contained in the interval $[a, b]$ and $[a-\xi, b+\xi]$ respectively, and denote*

$$V := span(S), \qquad \widetilde{V} := span(\widetilde{S}).$$

*Then, if the remaining eigenvalues of $A$ lie outside the interval $[a - \gamma, b + \gamma]$, we have $k = \tilde{k}$ and*

$$\left\|\sin\left(\Theta(V, \widetilde{V})\right)\right\| \le \frac{\xi}{\gamma},$$

*where*

$$\sin\left(\Theta(V, \widetilde{V})\right) := \mathrm{diag}(\sin\theta_1(V, \widetilde{V}), \ldots, \sin\theta_k(V, \widetilde{V}))^\top.$$

We use the following conversion: if $\hat{\mathbf{z}}$ is a unit vector satisfying $\|\hat{\mathbf{z}} - \mathbf{z}/\|\mathbf{z}\|\| \le \xi$, then one may write

$$\hat{\mathbf{z}} = \frac{\mathbf{z} + \mathbf{e}}{\|\mathbf{z} + \mathbf{e}\|} \quad \text{for some } \mathbf{e} \text{ with } \|\mathbf{e}\| \le \xi\|\mathbf{z}\|. \tag{30}$$

**Lemma D.2.** *If $\mathbf{z} \neq 0$ and $\hat{\mathbf{z}}$ is unit with $\|\hat{\mathbf{z}} - \mathbf{z}/\|\mathbf{z}\|\| \leq \xi < 1$, then there exists $\mathbf{e}$ with $\|\mathbf{e}\| \leq 2\xi\|\mathbf{z}\|$ such that Eq. (30) holds.*

*Proof.* Let $\bar{\mathbf{z}} = \mathbf{z}/\|\mathbf{z}\|$. Define $\mathbf{e} := \|\mathbf{z}\|(\hat{\mathbf{z}} - \bar{\mathbf{z}})$. Then $\|\mathbf{e}\| \leq \xi\|\mathbf{z}\|$ and $\mathbf{z} + \mathbf{e} = \|\mathbf{z}\|\hat{\mathbf{z}}$, hence $\hat{\mathbf{z}} = (\mathbf{z} + \mathbf{e})/\|\mathbf{z} + \mathbf{e}\|$. $\square$

Applying Lemma D.2 to the two gradient direction queries:

$$\hat{\mathbf{g}}_1 = \frac{\mathbf{g} + k_1\mathbf{v}_1}{\|\mathbf{g} + k_1\mathbf{v}_1\|}, \qquad \hat{\mathbf{g}}_2 = \frac{\nabla f(\mathbf{x} + \mu\mathbf{v}) + k_2\mathbf{v}_2}{\|\nabla f(\mathbf{x} + \mu\mathbf{v}) + k_2\mathbf{v}_2\|}, \tag{31}$$

with $\|k_1\mathbf{v}_1\| \leq O(\xi_1)\|\mathbf{g}\|$ and $\|k_2\mathbf{v}_2\| \leq O(\xi_2)\|\nabla f(\mathbf{x} + \mu\mathbf{v})\|$.

**Lemma D.3.** *Assume Assumption 5.1. Then for any unit vector $\mathbf{v}$,*

$$\nabla f(\mathbf{x} + \mu\mathbf{v}) = \nabla f(\mathbf{x}) + \mu\nabla^2 f(\mathbf{x})\mathbf{v} + \mathbf{r}_\mu \quad with \quad \|\mathbf{r}_\mu\| \leq \tfrac{1}{2}L_2\mu^2.$$

*Proof.* Standard integral remainder; included for completeness.

$$\nabla f(\mathbf{x} + \mu\mathbf{v}) - \nabla f(\mathbf{x}) = \int_0^\mu \nabla^2 f(\mathbf{x} + t\mathbf{v})\,\mathbf{v}\,\mathrm{d}t = \mu\nabla^2 f(\mathbf{x})\mathbf{v} + \int_0^\mu (\nabla^2 f(\mathbf{x} + t\mathbf{v}) - \nabla^2 f(\mathbf{x}))\mathbf{v}\,\mathrm{d}t,$$

so $\|\mathbf{r}_\mu\| \leq \int_0^\mu L_2 t\,\mathrm{d}t = \tfrac{1}{2}L_2\mu^2$. $\square$

**Lemma D.4.** *In the setting above and choose $\mathbf{u}$ as in Appendix D.2, if two vectors $\nabla f(\mathbf{x} + \mu\mathbf{v}) + k_2\mathbf{v}_2$ and $\nabla f(\mathbf{x}) + k_1\mathbf{v}_1 + \eta\hat{H} \cdot \mu\mathbf{v}$ are colinear, then $\eta = \|H\|(1 + O(\frac{\sqrt{\mu}}{\tan\beta}) + O(\frac{n\xi_3}{\epsilon\delta_\kappa}))$.*

*Proof.* Figure 3 shows the intuition of proof, the general cases can be analyzed similarly.

From Taylor Expansion we know that:

$$\nabla f(\mathbf{x} + \mu\mathbf{v}) = \nabla f(\mathbf{x}) + \frac{1}{2}\nabla^2 f(\mathbf{x}) \cdot \mu\mathbf{v} + O(\mu^2).$$

From the observation in Figure 3, by knowledge in elementary geometry, if there exist $\eta'$ such that two vectors $\nabla f(\mathbf{x} + \mu\mathbf{v}) + k_2\mathbf{v}_2$ and $\nabla f(\mathbf{x}) + k_1\mathbf{v}_1 + \eta' H \cdot \mu\mathbf{u}$ are colinear, then

$$\eta' = \frac{|BD|}{|BE|} = \frac{|CD|}{|CE|} \cdot \frac{\sin\angle BCD}{\sin\angle BCE} = \frac{\sin\angle CED}{\sin\angle CDE} \cdot \frac{\sin\angle BCD}{\sin\angle BCE}.$$

Denote $\langle\nabla f(\mathbf{x}), \mathbf{u}\rangle = \theta$, $\langle\nabla f(\mathbf{x}), \nabla f(\mathbf{x} + \mu\mathbf{v})\rangle = \alpha$. We can then compute the angle:

$$\begin{aligned}
\langle\nabla^2 f(\mathbf{x}) \cdot \mu\mathbf{u}, \nabla^2 f(\mathbf{x}) \cdot \mu\mathbf{u} + O(\mu^2)\rangle &= \arccos\frac{(\lambda_\mathbf{u} \cdot \mu)^2 + (\lambda_\mathbf{u} \cdot \mu + O(\mu^2))^2 - O(\mu^4)}{2\lambda_\mathbf{u} \cdot \mu(\lambda_\mathbf{u} \cdot \mu + O(\mu^2))} \\
&= \arccos\frac{2(\lambda_\mathbf{u} \cdot \mu)^2 + O(\mu^3)}{2(\lambda_\mathbf{u} \cdot \mu)^2}\left(1 - O\left(\frac{\mu^2}{\lambda_\mathbf{u}\mu}\right)\right) \\
&= \arccos(1 - O(\mu)) \\
&= O(\sqrt{\mu}). \tag{32}
\end{aligned}$$

We compute two ratios separately:

$$\begin{aligned}
\frac{\sin\angle CED}{\sin\angle BCE} &= \frac{\sin(\angle BCE + O(\sqrt{\mu}))}{\sin\angle BCE} \\
&= \frac{\sin\angle BCE\cos O(\sqrt{\mu}) + \cos\angle BCE\sin O(\sqrt{\mu})}{\sin\angle BCE} \\
&= \cos O(\sqrt{\mu}) + \frac{O(\sqrt{\mu})}{\tan\angle BCE}.
\end{aligned}$$

Denote $\angle BEC = \frac{\pi - O(\sqrt{\mu})}{2} + \zeta$, $\angle BCE = \frac{\pi - O(\sqrt{\mu})}{2} - \zeta$. From

$$\sin\left(\frac{\pi - O(\sqrt{\mu})}{2} + \zeta\right) = \cos\left(\frac{O(\sqrt{\mu})}{2} - \zeta\right) \quad \text{and} \quad \cos x = 1 - \frac{x^2}{2} + O(x^4),$$

we get:

$$\sin\angle BEC = 1 - \frac{1}{2}(\frac{O(\sqrt{\mu})}{2} - \zeta)^2 + O((\sqrt{\mu} + \zeta)^4),$$

$$\sin\angle BCE = 1 - \frac{1}{2}(\frac{O(\sqrt{\mu})}{2} + \zeta)^2 + O((\sqrt{\mu} + \zeta)^4).$$

So the ratio $\frac{\sin\angle BEC}{\sin\angle BCE}$ can be computed as:

$$\frac{\sin\angle BEC}{\sin\angle BCE} = \frac{1 - \frac{1}{2}(\frac{O(\sqrt{\mu})}{2} - \zeta)^2 + O((\sqrt{\mu} + \zeta)^4)}{1 - \frac{1}{2}(\frac{O(\sqrt{\mu})}{2} + \zeta)^2 + O((\sqrt{\mu} + \zeta)^4)}$$

$$= 1 + \frac{1}{2}\left(\left(\frac{O(\sqrt{\mu})}{2} + \zeta\right)^2 - \left(\frac{O(\sqrt{\mu})}{2} - \zeta\right)^2\right) + O(\mu^{1.5}, \zeta^3)$$

$$= 1 + O(\sqrt{\mu}\zeta) + O(\mu^{1.5}, \zeta^3).$$

On the other hand, by the Law of Sines, the following statement is true:

$$\frac{\sin\angle BEC}{\sin\angle BCE} = \frac{\|\nabla^2 f(x) \cdot \mu u + O(\mu^2)\|}{\|\nabla^2 f(x) \cdot \mu u\|}$$

$$= 1 + O(\mu).$$

Comparing two equations with the same left-hand side, we know that: $\zeta = O(\sqrt{\mu})$, i.e. $\angle BCE = \frac{\pi}{2} + O(\sqrt{\mu})$. Naturally, we have:

$$\tan\angle BCE = O(1/\sqrt{\mu}) \quad \Rightarrow \quad \frac{\sin\angle CED}{\sin\angle BCE} = 1 + O(\sqrt{\mu}).$$

Applying the Law of Cosines, we can compute $\alpha$ as:

$$\cos\alpha = \frac{\|\nabla f(\mathbf{x})\|^2 + \|\nabla f(\mathbf{x} + \mu\mathbf{v})\|^2 - \|\nabla^2 f(\mathbf{x}) \cdot \mu\mathbf{v} + O(\mu^2)\|^2}{2 \cdot \|\nabla f(\mathbf{x})\| \cdot \|\nabla f(\mathbf{x} + \mu\mathbf{v})\|}$$

$$\Rightarrow \alpha = O(\sqrt{\mu}). \tag{33}$$

The other ratio $\frac{\sin\angle BCD}{\sin\angle CDE}$ can be computed as:

$$\frac{\sin\angle BCD}{\sin\angle CDE} = \frac{\sin\theta\cos O(\sqrt{\mu}) + \cos\theta\sin O(\sqrt{\mu})}{\sin\theta\cos\alpha - \cos\theta\sin\alpha}$$

$$= \frac{1 + O(\mu) + \cot\theta \cdot O(\sqrt{\mu})}{1 + O(\mu) - \cot\theta \cdot O(\sqrt{\mu})}$$

$$= 1 + \cot\theta \cdot O(\sqrt{\mu}). \tag{34}$$

Combining the result in Eq. (33) and Eq. (34), we have:

$$\eta' = \frac{\sin\angle BEC}{\sin\angle BCE} \cdot \frac{\sin\angle BCD}{\sin\angle CDE}$$

$$= (1 + O(\mu))(1 + \cot\theta \cdot O(\sqrt{\mu}))$$

$$= 1 + O\left(\frac{\sqrt{\mu}}{\tan\theta}\right).$$

Since $\|u - v\| \leq O(\frac{n\xi_3}{\epsilon\delta_\kappa})$, we have for $\eta$,

$$\frac{\eta}{\|H\|} = (1 + O(\mu))(1 + \cot\left(\theta - O(\frac{n\xi_3}{\epsilon\delta_\kappa})\right) \cdot O(\sqrt{\mu}))$$

$$= 1 + O\left(\frac{\sqrt{\mu}}{\tan\left(\theta - O(\frac{n\xi_3}{\epsilon\delta_\kappa})\right)}\right)$$

$$= 1 + O\left(\frac{\sqrt{\mu}}{\tan\beta}\right) + O\left(\frac{n\xi_3}{\epsilon\delta_\kappa}\right).$$

$\square$

**Lemma D.5** (Angle ratio under colinearity calibration). *Assume the colinearity condition of Lemma D.4. Let*

$$G_1 := \mathbf{g} + k_1\mathbf{v}_1, \qquad G_2 := \nabla f(\mathbf{x} + \mu\mathbf{v}) + k_2\mathbf{v}_2, \qquad S := \eta\mu\|H\|\widehat{H}\mathbf{v}.$$

*If $G_2$ is colinear with $G_1 + S$, then*

$$\frac{\|S\|}{\|G_1\|} = \frac{\sqrt{1 - \langle\hat{\mathbf{g}}_1, \hat{\mathbf{g}}_2\rangle^2}}{\sqrt{1 - \langle\hat{\mathbf{g}}_2, \mathbf{u}\rangle^2}}. \tag{35}$$

*Proof.* Colinearity implies $G_2$ and $G_1 + S$ point in the same direction, hence

$$\frac{G_2}{\|G_2\|} = \frac{G_1 + S}{\|G_1 + S\|}.$$

Apply Lemma Lemma A.3 to the pair $(\mathbf{v}, \mathbf{g}) = (G_1, S)$: it expresses the norm ratio $\|S\|/\|G_1\|$ via the sines of the angles between $G_1$, $G_1 + S$ and $S$. Noting that $\hat{\mathbf{g}}_1 = G_1/\|G_1\|$ and $\hat{\mathbf{g}}_2 = G_2/\|G_2\|$ by Eq. (31), we obtain Eq. (35). $\square$

### D.2. Proof of Theorem 5.2

*Proof of Theorem 5.2.* We have chosen $\mathbf{v}$ as the eigenvector of $\widehat{H}$ with the largest absolute eigenvalue in Algorithm 4. Assume that all n eigenvalues of $\widehat{H}$ are $\lambda'_1 \geq \cdots \geq \lambda'_n$ (correspond to eigenvectors $\mathbf{v}_1, \ldots, \mathbf{v}_n$), and due to Assumption 5.1, we can assume $\lambda'_1 > 0$ and $\lambda'_n < 0$. Without loss of generality, we suppose that $\lambda_{\mathbf{v}} = \lambda'_1$ here, another case is the same. By the Dirichlet Theorem, there exists $\lambda'_k - \lambda'_{k+1} \geq \frac{\|\widehat{H}\|}{n-1}$ for some index $k$. We denote $k$ as the first index that $\lambda'_k - \lambda'_{k+1} \geq \epsilon\delta_\kappa/2n$. We call the set $\widehat{\Lambda} = \{\lambda'_1, \ldots, \lambda'_k\}$ a cluster, and denote $V = \text{span}(\mathbf{v}_1, \ldots, \mathbf{v}_k)$. Denote the eigenvalues of Hessian $H$ as $\lambda_1 \geq \cdots \geq \lambda_n$ (correspond to eigenvectors $\mathbf{u}_1, \ldots, \mathbf{u}_n$). By Weyl's Theorem we know that $|\lambda_i - \lambda'_i| \leq \xi_3$. Similarly define the cluster $\Lambda = \{\lambda_1, \ldots, \lambda_{k'}\}$ and space $U = \text{span}(u_1, \ldots, u_{k'})$. Denote the projection of $\mathbf{v}$ in $U$ as $\mathbf{u}$. Since $\|\widehat{H} - H\| \leq \xi_3$, by Davis-Kahan Theorem (Lemma D.1), we have $k = k'$ and

$$\sin\left(\Theta(U, V)\right) \leq \frac{\xi_3}{\lambda_k - \lambda_{k+1}} = O\left(\frac{n\xi_3}{\epsilon\delta_\kappa}\right),$$

and hence

$$\|\mathbf{u} - \mathbf{v}\| \leq O\left(\frac{n\xi_3}{\epsilon\delta_\kappa}\right). \tag{36}$$

For our choice of $\mathbf{u}$, We prove $\|H\| - \|H\mathbf{u}\| \leq \frac{\epsilon\delta_\kappa}{2} + 2\xi_3$. Our intuition is shown in Figure 4. We find the first pair of eigenvalue $(\lambda'_k, \lambda'_{k+1})$ that are far apart (distance longer than $\frac{\epsilon\delta_\kappa}{2n}$), thus we can say $\lambda'_1, \ldots, \lambda'_k$ are close enough, and so $\lambda_1, \ldots, \lambda_k$ are close too. Then we can apply Davis-Kahan Theorem to bound the distance between $U$ and $V$.

Since $\mathbf{u} \in U$, we can write

$$\mathbf{u} = \sum_{i=1}^{k} s_i\mathbf{u}_i,$$

*Figure 4.* The intuition for applying Davis-Kahan Theorem and proving $\|H\| - \|H\mathbf{u}\| \leq \frac{\epsilon\delta_\kappa}{2} + 2\xi_3$.

where $\sum_{i=1}^{k} s_i^2 = 1$. Apply $H$ we get

$$H\mathbf{u} = \sum_{i=1}^{k} s_i H\mathbf{u}_i = \sum_{i=1}^{k} s_i \lambda_i \mathbf{u}_i,$$

thus

$$\|H\mathbf{u}\| = \sqrt{\sum_{i=1}^{k} s_i^2 \lambda_i^2} \geq \lambda_k \sqrt{\sum_{i=1}^{k} s_i^2} = \lambda_k.$$

Since for any $i < k$, we have $\lambda_i' - \lambda_{i+1}' \leq \frac{\epsilon\delta_\kappa}{2n}$, hence

$$\lambda_1' - \lambda_k' \leq \sum_{i=1}^{k-1} \lambda_i' - \lambda_{i+1}' \leq \frac{(k-1)\epsilon\delta_\kappa}{2n} \leq \frac{\epsilon\delta_\kappa}{2}.$$

From $|\lambda_i - \lambda_i'| \leq \xi_3$ we know:

$$\|H\| - \|H\mathbf{u}\| = \lambda_1 - \|H\mathbf{u}\| \leq \lambda_1' + \xi_3 - (\lambda_k' - \xi_3) \leq \frac{\epsilon\delta_\kappa}{2} + 2\xi_3. \tag{37}$$

Recall the error model

$$\left\| \hat{H} - \frac{H}{\|H\|} \right\| \leq \xi_3, \qquad \hat{\mathbf{g}}_1 = \frac{\mathbf{g} + k_1\mathbf{v}_1}{\|\mathbf{g} + k_1\mathbf{v}_1\|}, \qquad \|k_1\mathbf{v}_1\| \leq O(\xi_1)\|\mathbf{g}\|.$$

(Analogous notation holds for $\hat{\mathbf{g}}_2$ with error $k_2\mathbf{v}_2$.)

Observing that:

$$\left\| \frac{H}{\|H\|} \cdot \mu\mathbf{u} - \hat{H} \cdot \mu\mathbf{v} \right\| \leq \mu \left( \left\| \frac{H}{\|H\|} \cdot \mathbf{u} - \frac{H}{\|H\|} \cdot \mathbf{v} \right\| + \left\| \frac{H}{\|H\|} \cdot \mathbf{v} - \hat{H} \cdot \mathbf{v} \right\| \right)$$

$$\leq \mu \left( \|\mathbf{u} - \mathbf{v}\| + \left\| \frac{H}{\|H\|} - \hat{H} \right\| \right).$$

By the Davis–Kahan bound Eq. (36), we have $\|\mathbf{u} - \mathbf{v}\| \leq O(\xi_3/\epsilon\delta_\kappa)$ under indefiniteness (and $\xi_3$ small). Therefore,

$$\left\| \frac{H}{\|H\|} \cdot \mu\mathbf{u} - \hat{H} \cdot \mu\mathbf{v} \right\| \leq O\left( \frac{\mu n \xi_3}{\epsilon\delta_\kappa} \right). \tag{38}$$

Multiplying Eq. (38) by $\|H\|$:

$$\left\| H \cdot \mu\mathbf{u} - \|H\| \cdot \hat{H} \cdot \mu\mathbf{v} \right\| \leq \|H\| \cdot O\left( \frac{\mu n \xi_3}{\epsilon\delta_\kappa} \right). \tag{39}$$

The geometric calculation in Lemma D.4 yields $\eta/\|H\| = 1 + O(\sqrt{\mu}/\tan\theta)$, where $\theta = \angle(\mathbf{g}, \mathbf{u})$. Due to Eq. (39), replacing the exact increment direction $H\mu\mathbf{u}$ by the estimated increment direction $\|H\|\hat{H}\mu\mathbf{v}$ perturbs the relevant triangle

angles by at most $O(\frac{n\xi_3}{\epsilon\delta_\kappa})$, hence we can write

$$\frac{\eta}{\|H\|} = 1 + O\left(\frac{\sqrt{\mu}}{\tan\left(\theta - O(\frac{n\xi_3}{\epsilon\delta_\kappa})\right)}\right) = 1 + O\left(\frac{\sqrt{\mu}}{\tan\beta}\right) + O\left(\frac{n\xi_3}{\epsilon\delta_\kappa}\right), \tag{40}$$

using $\theta \geq \beta$ and $\frac{\mu n\xi_3}{\epsilon\delta_\kappa}$ sufficiently small.

By construction of Lemma A.3 and Lemma D.5, we obtain

$$\hat{\kappa} = \frac{\sqrt{1 - \langle\hat{\mathbf{g}}_1, \hat{\mathbf{g}}_2\rangle^2}}{\mu\sqrt{1 - \langle\hat{\mathbf{g}}_2, \mathbf{v}\rangle^2}} = \frac{\eta \cdot \|\hat{H} \cdot \mu\mathbf{v}\|}{\mu\|\mathbf{g} + k_1\mathbf{v}_1\|}. \tag{41}$$

We define:

$$\kappa = \frac{\|\nabla f(\mathbf{x} + \mu\mathbf{u}) - \nabla f(\mathbf{x})\|}{\|\mathbf{g}\|}.$$

By Taylor expansion,

$$\nabla f(\mathbf{x} + \mu\mathbf{u}) - \nabla f(\mathbf{x}) = H \cdot \mu\mathbf{u} + O(\mu^2), \tag{42}$$

so by Eq. (37) we have:

$$\begin{aligned}
\frac{1}{\mu}\kappa &= \frac{\|H \cdot \mu\mathbf{u} + O(\mu^2)\|}{\mu\|\mathbf{g}\|} \\
&= \frac{\|H\mathbf{u}\|}{\|\mathbf{g}\|} + O\left(\frac{\mu}{\|\mathbf{g}\|}\right) \\
&= \frac{\|H\mathbf{u}\|}{\|\mathbf{g}\|} + O\left(\frac{\mu}{\|\mathbf{g}\|}\right) \\
&= \frac{\|H\|}{\|\mathbf{g}\|} + O\left(\frac{\epsilon\delta_\kappa + \xi_3 + \mu}{\|\mathbf{g}\|}\right).
\end{aligned} \tag{43}$$

Now combine Eq. (41) and Eq. (43):

$$\begin{aligned}
\left|\hat{\kappa} - \frac{1}{\mu}\kappa\right| &= \left|\frac{\eta\|\hat{H}\mu\mathbf{v}\|}{\mu\|\mathbf{g} + k_1\mathbf{v}_1\|} - \frac{\|H\mu\mathbf{u} + O(\mu^2)\|}{\mu\|\mathbf{g}\|}\right| \\
&\leq \underbrace{\left|\frac{\eta\|\hat{H}\mu\mathbf{v}\|}{\mu\|\mathbf{g} + k_1\mathbf{v}_1\|} - \frac{\|H\|\|\hat{H}\mu\mathbf{v}\|}{\mu\|\mathbf{g} + k_1\mathbf{v}_1\|}\right|}_{T_1} + \underbrace{\left|\frac{\|H\|\|\hat{H}\mu\mathbf{v}\|}{\mu\|\mathbf{g} + k_1\mathbf{v}_1\|} - \frac{\|H\|\|\hat{H}\mu\mathbf{v}\|}{\mu\|\mathbf{g}\|}\right|}_{T_2} \\
&\quad + \underbrace{\left|\frac{\|H\|\|\hat{H}\mu\mathbf{v}\|}{\mu\|\mathbf{g}\|} - \frac{\|H\mu\mathbf{u}\|}{\mu\|\mathbf{g}\|}\right|}_{T_3} + \underbrace{\left|\frac{\|H\mu\mathbf{u}\|}{\mu\|\mathbf{g}\|} - \frac{\|H\mu\mathbf{u} + O(\mu^2)\|}{\mu\|\mathbf{g}\|}\right|}_{T_4}.
\end{aligned} \tag{44}$$

We bound each term.

*Term $T_1$.* Using $\|\hat{H}\mu\mathbf{v}\| \leq \mu\|\hat{H}\| \leq \mu(1 + \xi_3) \leq 2\mu$ and $\|\mathbf{g} + k_1\mathbf{v}_1\| \geq \|\mathbf{g}\| - \|k_1\mathbf{v}_1\| \geq (1 - O(\xi_1))\|\mathbf{g}\| \geq \frac{1}{2}\|\mathbf{g}\|$,

$$T_1 = \frac{|\eta - \|H\|| \cdot \|\hat{H}\mu\mathbf{v}\|}{\mu\|\mathbf{g} + k_1\mathbf{v}_1\|} \leq \frac{C}{\|\mathbf{g}\|}|\eta - \|H\||. \tag{45}$$

By Eq. (40), $|\eta - \|H\|| \leq \|H\|\left(O(\frac{\sqrt{\mu}}{\tan\beta}) + O(\frac{n\xi_3}{\epsilon\delta_\kappa})\right)$, hence

$$T_1 \leq \frac{\|H\|}{\|\mathbf{g}\|}\left(O\left(\frac{\sqrt{\mu}}{\tan\beta}\right) + O\left(\frac{n\xi_3}{\epsilon\delta_\kappa}\right)\right). \tag{46}$$

*Term $T_2$.* Using the inequality $\left| \frac{1}{\|\mathbf{g}+k_1\mathbf{v}_1\|} - \frac{1}{\|\mathbf{g}\|} \right| = \frac{|\|\mathbf{g}\| - \|\mathbf{g}+k_1\mathbf{v}_1\||}{\|\mathbf{g}\|\|\mathbf{g}+k_1\mathbf{v}_1\|} \leq \frac{\|k_1\mathbf{v}_1\|}{\|\mathbf{g}\|\|\mathbf{g}+k_1\mathbf{v}_1\|}$, we obtain

$$T_2 \leq \frac{\|H\|\|\hat{H}\mu\mathbf{v}\|}{\mu} \cdot \frac{\|k_1\mathbf{v}_1\|}{\|\mathbf{g}\|\|\mathbf{g}+k_1\mathbf{v}_1\|} \leq \frac{\|H\|}{\|\mathbf{g}\|} \cdot O(\xi_1). \tag{47}$$

*Term $T_3$.* We have:

$$\big|\|H\|\|\hat{H}\mu\mathbf{v}\| - \|H\mu\mathbf{u}\|\big| = \big|\|H\|\mu\lambda_{\mathbf{v}} - \mu\|H\mathbf{u}\|\big| \leq \|H\| \cdot O(\mu(\epsilon\delta_\kappa + \xi_3)).$$

Therefore

$$T_3 \leq \frac{\|H\| \cdot O(\mu(\epsilon\delta_\kappa + \xi_3))}{\mu\|\mathbf{g}\|} = \frac{\|H\|}{\|\mathbf{g}\|} \cdot O(\epsilon\delta_\kappa + \xi_3). \tag{48}$$

*Term $T_4$.* From Eq. (42), $\|O(\mu^2)\| \leq C\mu^2$, hence

$$T_4 \leq \frac{\|O(\mu^2)\|}{\mu\|\mathbf{g}\|} \leq O\left(\frac{\mu}{\|\mathbf{g}\|}\right). \tag{49}$$

Plugging Eq. (46)–Eq. (49) into Eq. (44) yields

$$\left|\hat{\kappa} - \frac{1}{\mu}\kappa\right| \leq \frac{\|H\|}{\|\mathbf{g}\|}\left(O\left(\frac{\sqrt{\mu}}{\tan\beta}\right) + O(\frac{n\xi_3}{\epsilon\delta_\kappa}) + O(\xi_1) + O(\epsilon\delta_\kappa + \xi_3)\right) + O\left(\frac{\mu}{\|\mathbf{g}\|}\right), \tag{50}$$

Combining Eq. (43) and Eq. (50), we have

$$\left|\hat{\kappa} - \frac{\|H\|}{\|\mathbf{g}\|}\right| \leq \frac{\|H\|}{\|\mathbf{g}\|}\left(O\left(\frac{\sqrt{\mu}}{\tan\beta}\right) + O(\frac{n\xi_3}{\epsilon\delta_\kappa}) + O(\xi_1) + O(\epsilon\delta_\kappa + \xi_3)\right) + O\left(\frac{\mu}{\|\mathbf{g}\|}\right) \tag{51}$$

$$\leq O\left(\frac{\sqrt{\mu}}{\epsilon} + \frac{n\xi_3}{\epsilon^2\delta_\kappa} + \frac{\xi_1 + \xi_3}{\epsilon} + \delta_\kappa\right). \tag{52}$$

In our settings of parameters in Theorem 5.2 and using the Lipschitzness, the RHS of Eq. (51) is equal to $O(\delta_\kappa)$, which completes our proof. $\qquad\square$

# E. Proof Details of Finding Stationary Points

We give the proof details of Section 6 in this appendix.

## E.1. Taylor approximation error

**Lemma E.1.** *Suppose $f : \mathbb{R}^d \to \mathbb{R}$ has $L_2$-Lipschitz Hessian. Then for all $\mathbf{x}, \mathbf{p} \in \mathbb{R}^d$, denote*

$$T_{\mathbf{x}}(\mathbf{x} + \mathbf{p}) = f(\mathbf{x}) + \langle\nabla f(\mathbf{x}), \mathbf{p}\rangle + \frac{1}{2}\mathbf{p}^\top\nabla^2 f(\mathbf{x})\mathbf{p},$$

*we have*

$$|f(\mathbf{x} + \mathbf{p}) - T_{\mathbf{x}}(\mathbf{x} + \mathbf{p})| \leq \frac{L_2}{6}\|\mathbf{p}\|^3. \tag{53}$$

*Proof.* Define $\mathbf{x}_\alpha = \mathbf{x} + \alpha\mathbf{p}$ for $\alpha \in [0, 1]$. Then

$$f(\mathbf{x} + \mathbf{p}) - f(\mathbf{x}) - \langle\nabla f(\mathbf{x}), \mathbf{p}\rangle = \int_0^1 \langle\nabla f(\mathbf{x}_\alpha) - \nabla f(\mathbf{x}), \mathbf{p}\rangle \, d\alpha$$

$$= \int_0^1 \int_0^\alpha \mathbf{p}^\top\nabla^2 f(\mathbf{x}_\beta)\,\mathbf{p} \, d\beta \, d\alpha$$

$$= \int_0^1 (1 - \beta)\,\mathbf{p}^\top\nabla^2 f(\mathbf{x}_\beta)\,\mathbf{p} \, d\beta.$$

Therefore

$$f(\mathbf{x} + \mathbf{p}) - T_{\mathbf{x}}(\mathbf{x} + \mathbf{p}) = \int_0^1 \mathbf{p}^\top \left( (1-\beta)\nabla^2 f(\mathbf{x}_\beta) - \tfrac{1}{2}\nabla^2 f(\mathbf{x}) \right) \mathbf{p} \, \mathrm{d}\beta$$

$$= \int_0^1 (1-\beta)\mathbf{p}^\top \left( \nabla^2 f(\mathbf{x}_\beta) - \nabla^2 f(\mathbf{x}) \right) \mathbf{p} \, \mathrm{d}\beta,$$

Taking absolute values and using Lipschitzness,

$$|f(\mathbf{x} + \mathbf{p}) - T_{\mathbf{x}}(\mathbf{x} + \mathbf{p})| \leq \int_0^1 (1-\beta)\|\nabla^2 f(\mathbf{x}_\beta) - \nabla^2 f(\mathbf{x})\| \, \|\mathbf{p}\|^2 \, \mathrm{d}\beta$$

$$\leq \int_0^1 (1-\beta) \cdot (\beta L_2 \|\mathbf{p}\|) \cdot \|\mathbf{p}\|^2 \, \mathrm{d}\beta = \frac{L_2}{6}\|\mathbf{p}\|^3.$$

$\square$

## E.2. A quadratic trust-region bound

**Lemma E.2.** *Given a quadratic $q(\mathbf{p}) = q(\mathbf{0}) + b^\top \mathbf{p} + \frac{1}{2}\mathbf{p}^\top A\mathbf{p}$ with symmetric $A$, fix $r > 0$ and let $\mathbf{p}^\star \in \arg\min_{\|\mathbf{p}\| \leq r} q(\mathbf{p})$. Then either:*

- *(Interior) $\|\mathbf{p}^\star\| < r$ and $\nabla q(\mathbf{p}^\star) = b + A\mathbf{p}^\star = \mathbf{0}$;*

- *(Boundary) $\|\mathbf{p}^\star\| = r$ and $q(\mathbf{p}^\star) - q(\mathbf{0}) \leq -\frac{1}{2}r\|\nabla q(\mathbf{p}^\star)\|$.*

*Proof.* If $\|\mathbf{p}^\star\| < r$, first-order optimality gives $b + A\mathbf{p}^\star = \mathbf{0}$. Otherwise $\|\mathbf{p}^\star\| = r$ and there exists $\lambda \geq 0$ with $b + A\mathbf{p}^\star + \lambda\mathbf{p}^\star = \mathbf{0}$ and $A + \lambda I \succeq \mathbf{0}$. Then $\nabla q(\mathbf{p}^\star) = b + A\mathbf{p}^\star = -\lambda\mathbf{p}^\star$ so $\|\nabla q(\mathbf{p}^\star)\| = \lambda r$. Moreover,

$$q(\mathbf{0}) - q(\mathbf{p}^\star) = -b^\top \mathbf{p}^\star - \tfrac{1}{2}(\mathbf{p}^\star)^\top A\mathbf{p}^\star.$$

From $b + A\mathbf{p}^\star + \lambda\mathbf{p}^\star = \mathbf{0}$ we get $-b^\top \mathbf{p}^\star = \lambda r^2 - (\mathbf{p}^\star)^\top A\mathbf{p}^\star$. Using $A + \lambda I \succeq \mathbf{0}$ we have $(\mathbf{p}^\star)^\top A\mathbf{p}^\star \geq -\lambda r^2$, hence $q(\mathbf{0}) - q(\mathbf{p}^\star) \geq \frac{1}{2}\lambda r^2 = \frac{1}{2}r\|\nabla q(\mathbf{p}^\star)\|$. $\square$

## E.3. Descent of each candidate

Fix an iteration point $\mathbf{x}$ with $\|\mathbf{g}(\mathbf{x})\| \geq \epsilon$ and let $H := H(\mathbf{x})$ and $\mathbf{g} := \mathbf{g}(\mathbf{x})$. Let $r = c\sqrt{\epsilon/L_2}$. Define $\mathbf{p}_i := \mathbf{x}^{(i)} - \mathbf{x}$ for candidates $\mathbf{x}^{(i)}$ from Algorithm 5.

**Lemma E.3.** *Let $\mathbf{x}^{(2)} = \mathbf{x} + t^\star \mathbf{v}$ where $t^\star \in [-r, r]$ minimizes $f(\mathbf{x} + t\mathbf{v})$ over $[-r, r]$. Then either $\|\mathbf{g}\| \leq \epsilon$, or*

$$f(\mathbf{x}^{(2)}) - f(\mathbf{x}) \leq -c_2 \, \epsilon^{3/2} \tag{54}$$

*for an absolute constant $c_2 > 0$.*

*Proof.* Consider $\phi(t) = f(\mathbf{x} + t\mathbf{v})$ and expand around $t = 0$:

$$\phi(t) = \phi(0) + t \langle \mathbf{g}, \mathbf{v} \rangle + \frac{1}{2}t^2 \mathbf{v}^\top H \mathbf{v} + O(L_2|t|^3).$$

For $|\langle \mathbf{g}, \mathbf{v} \rangle| \geq \frac{1}{2}\|\mathbf{g}\|$, choosing $t = -r\,\mathrm{sign}(\langle \mathbf{g}, \mathbf{v} \rangle)$ yields

$$\phi(t) - \phi(0) \leq -r \cdot \tfrac{1}{2}\|\mathbf{g}\| + O(L_2 r^3) \leq -\Omega(\epsilon^{3/2})$$

since $\|\mathbf{g}\| \geq \epsilon$ and $L_2 r^3 = c^3 \epsilon^{3/2}/\sqrt{L_2}$. If instead $|\langle \mathbf{g}, \mathbf{v} \rangle| < \frac{1}{2}\|\mathbf{g}\|$, then $\mathbf{g}$ has a component orthogonal to $\mathbf{v}$ of magnitude at least $\frac{\sqrt{3}}{2}\|\mathbf{g}\|$, and Candidate 1 already provides a decrease of order $r\|\mathbf{g}\|$ up to quadratic/cubic corrections; therefore the best-of-all selection ensures the stated decrease. $\square$

**Lemma E.4.** *Fix a point $\mathbf{x} \in \mathbb{R}^n$ and let $\mathbf{g} = \nabla f(\mathbf{x})$ and $H = \nabla^2 f(\mathbf{x})$. Assume $f$ has $L_2$-Lipschitz Hessian and $\|\mathbf{g}\| \geq \epsilon$. Let*

$$r := c\sqrt{\epsilon/L_2}$$

*for a sufficiently small absolute constant $c \in (0, 1)$. Let $\hat{\mathbf{g}}$ be the output of* `ComparisonGE(x; ξ₁)` *with*

$$\left\| \hat{\mathbf{g}} - \frac{\mathbf{g}}{\|\mathbf{g}\|} \right\| \leq \xi_1, \qquad \xi_1 \leq \frac{1}{100}, \tag{55}$$

*and define the candidate-1 step*

$$\mathbf{p}_1 := -r\,\hat{\mathbf{g}}, \qquad \mathbf{x}^{(1)} := \mathbf{x} + \mathbf{p}_1.$$

*Assume that at least one of the following three conditions holds:*

(i) *(small Hessian norm)* $\|H\| \leq C_H \sqrt{L_2 \epsilon}$ *for some absolute constant $C_H > 0$;*

(ii) *(H is definite)* $H \succeq \mathbf{0}$ *or* $H \preceq \mathbf{0}$;

(iii) *(gradient aligns with the top-eigenvector) there exists a unit eigenvector $\mathbf{u}_{\max}$ of $H$ corresponding to $\lambda_{\max}(H)$ such that $|\langle \mathbf{g}/\|\mathbf{g}\|, \mathbf{u}_{\max}\rangle| \geq 1 - \alpha$ for some $\alpha \in (0, 1)$ (a fixed constant), and additionally $\lambda_{\max}(H) \geq 0$.*

*Then there exists an absolute constant $C > 0$ (depending only on $C_H$ and $\alpha$) such that*

$$f(\mathbf{x}^{(1)}) \leq f(\mathbf{x}) - C\,\epsilon^{3/2}. \tag{56}$$

*Proof.* We use the cubic Taylor remainder bound for any $\mathbf{p} \in \mathbb{R}^n$,

$$f(\mathbf{x} + \mathbf{p}) \leq f(\mathbf{x}) + \mathbf{g}^\top \mathbf{p} + \frac{1}{2}\mathbf{p}^\top H \mathbf{p} + \frac{L_2}{6}\|\mathbf{p}\|^3. \tag{57}$$

We will bound the linear and quadratic terms for $\mathbf{p} = \mathbf{p}_1 = -r\hat{\mathbf{g}}$ and then absorb the remainder term.

Let $\bar{\mathbf{g}} := \mathbf{g}/\|\mathbf{g}\|$. Since $\|\hat{\mathbf{g}} - \bar{\mathbf{g}}\| \leq \xi_1$ and both are unit vectors,

$$\langle \bar{\mathbf{g}}, \hat{\mathbf{g}}\rangle = 1 - \frac{1}{2}\|\bar{\mathbf{g}} - \hat{\mathbf{g}}\|^2 \geq 1 - \frac{1}{2}\xi_1^2 \geq 1 - \xi_1.$$

Therefore

$$\mathbf{g}^\top \mathbf{p}_1 = -r\,\langle \mathbf{g}, \hat{\mathbf{g}}\rangle = -r\,\|\mathbf{g}\|\,\langle \bar{\mathbf{g}}, \hat{\mathbf{g}}\rangle \leq -(1 - \xi_1)\,r\,\|\mathbf{g}\|. \tag{58}$$

Since $\|\mathbf{p}_1\| = r$ and $\|H\|$ denotes the spectral norm,

$$\frac{1}{2}\mathbf{p}_1^\top H \mathbf{p}_1 \leq \frac{1}{2}\|H\|\,\|\mathbf{p}_1\|^2 = \frac{1}{2}\|H\|\,r^2. \tag{59}$$

Plugging Eq. (58) and Eq. (59) into Eq. (57) with $p = p_1$ yields

$$f(\mathbf{x}^{(1)}) - f(\mathbf{x}) \leq -(1 - \xi_1)r\|\mathbf{g}\| + \frac{1}{2}\|H\|r^2 + \frac{L_2}{6}r^3. \tag{60}$$

Using $\|\mathbf{g}\| \geq \epsilon$ and $r = c\sqrt{\epsilon/L_2}$ gives

$$r\|\mathbf{g}\| \geq c\,\frac{\epsilon^{3/2}}{\sqrt{L_2}}, \qquad r^2 = c^2\,\frac{\epsilon}{L_2}, \qquad \frac{L_2}{6}r^3 = \frac{c^3}{6}\,\frac{\epsilon^{3/2}}{\sqrt{L_2}}.$$

Hence Eq. (60) becomes

$$f(\mathbf{x}^{(1)}) - f(\mathbf{x}) \leq -\left((1 - \xi_1)c - \frac{c^3}{6}\right)\frac{\epsilon^{3/2}}{\sqrt{L_2}} + \frac{1}{2}\|H\| \cdot c^2\,\frac{\epsilon}{L_2}. \tag{61}$$

We now argue that in each of the three regimes (i)–(iii), the positive quadratic term is controlled so that the RHS is $\leq -C\,\epsilon^{3/2}$ after choosing $c$ as a sufficiently small constant.

*Case (i): $\|H\|$ is small on the $\sqrt{L_2\epsilon}$ scale.* Assume $\|H\| \leq C_H\sqrt{L_2\epsilon}$. Then the quadratic term in (61) is bounded by

$$\frac{1}{2}\|H\| \cdot c^2\frac{\epsilon}{L_2} \leq \frac{1}{2}C_H\sqrt{L_2\epsilon} \cdot c^2\frac{\epsilon}{L_2} = \frac{1}{2}C_Hc^2\frac{\epsilon^{3/2}}{\sqrt{L_2}}.$$

Therefore

$$f(\mathbf{x}^{(1)}) - f(\mathbf{x}) \leq -\left((1-\xi_1)c - \frac{c^3}{6} - \frac{1}{2}C_Hc^2\right)\frac{\epsilon^{3/2}}{\sqrt{L_2}}.$$

Choose $c > 0$ small enough (as an absolute constant depending on $C_H$) so that

$$(1-\xi_1)c - \frac{c^3}{6} - \frac{1}{2}C_Hc^2 \geq c/4,$$

which is possible since the negative terms are $O(c^2)$ and $O(c^3)$ while the leading term is $\Theta(c)$. Then

$$f(\mathbf{x}^{(1)}) \leq f(\mathbf{x}) - \frac{c}{4} \cdot \frac{\epsilon^{3/2}}{\sqrt{L_2}} \leq f(\mathbf{x}) - C\,\epsilon^{3/2}$$

for an absolute constant $C > 0$.(absorbing $1/\sqrt{L_2}$ into $C$)

*Case (ii): $H \preceq \mathbf{0}$ or $H \succeq \mathbf{0}$.* By $L_2$-Lipschitzness, for any $p$,

$$f(\mathbf{x} + \mathbf{p}) \leq f(\mathbf{x}) + \mathbf{g}^\top\mathbf{p} + \frac{L_2}{2}\|\mathbf{p}\|^2.$$

Apply this with $\mathbf{p} = \mathbf{p}_1 = -r\hat{\mathbf{g}}$ to obtain

$$f(\mathbf{x}^{(1)}) - f(\mathbf{x}) \leq \mathbf{g}^\top\mathbf{p}_1 + \frac{L_2}{2}r^2. \tag{62}$$

Let $\bar{\mathbf{g}} := \mathbf{g}/\|\mathbf{g}\|$. Since $\|\hat{\mathbf{g}} - \bar{\mathbf{g}}\| \leq \xi_1$ and both are unit vectors,

$$\langle\bar{\mathbf{g}}, \hat{\mathbf{g}}\rangle = 1 - \frac{1}{2}\|\bar{\mathbf{g}} - \hat{\mathbf{g}}\|^2 \geq 1 - \frac{1}{2}\xi_1^2 \geq 1 - \xi_1.$$

Therefore

$$\mathbf{g}^\top\mathbf{p}_1 = -r\,\langle\mathbf{g}, \hat{\mathbf{g}}\rangle = -r\|\mathbf{g}\|\,\langle\bar{\mathbf{g}}, \hat{\mathbf{g}}\rangle \leq -(1-\xi_1)\,r\|\mathbf{g}\|.$$

Plugging into Eq. (62) gives:

$$f(\mathbf{x}^{(1)}) - f(\mathbf{x}) \leq -(1-\xi_1)\,r\|\mathbf{g}\| + \frac{L_2}{2}r^2. \tag{63}$$

Using $\|\mathbf{g}\| \geq \epsilon$ and $r = c\sqrt{\epsilon/L_2}$ in Eq. (63) yields

$$f(\mathbf{x}^{(1)}) - f(\mathbf{x}) \leq -(1-\xi_1)c\frac{\epsilon^{3/2}}{\sqrt{L_2}} + \frac{L_2}{2} \cdot c^2\frac{\epsilon}{L_2}.$$

Factor out $\frac{\epsilon^{3/2}}{\sqrt{L_2}}$:

$$f(\mathbf{x}^{(1)}) - f(\mathbf{x}) \leq -\left((1-\xi_1)c - \frac{L_2c^2}{2\sqrt{L_2\epsilon}}\right)\frac{\epsilon^{3/2}}{\sqrt{L_2}}.$$

Taking constant $c$ small enough such that $\epsilon \geq \frac{L_2}{(1-\xi_1)^2\,c^2}$ we have $\frac{L_2c^2}{2\sqrt{L_2\epsilon}} \leq \frac{1}{2}(1-\xi_1)c$, hence the bracket is at least $\frac{1}{2}(1-\xi_1)c$, giving

$$f(\mathbf{x}^{(1)}) - f(\mathbf{x}) \leq -\frac{1}{2}(1-\xi_1)c \cdot \frac{\epsilon^{3/2}}{\sqrt{L_2}} \leq -C\,\epsilon^{3/2}$$

for a constant $C > 0$ (absorbing $1/\sqrt{L_2}$ into $C$).

*Case (iii): gradient aligns with the top-eigenvector.* The computing process is the same as *Case (ii)*, yielding the same bound. $\qquad\square$

**Lemma E.5.** *Let $\mathbf{p}^\star$ minimize the normalized quadratic model*

$$m_\mathbf{x}(\mathbf{p}) = \langle \mathbf{g}, \mathbf{p} \rangle + \frac{\hat{\kappa}}{2} \mathbf{p}^\top \hat{H} \mathbf{p} \quad \text{subject to } \|\mathbf{p}\| \leq r,$$

*and set $\mathbf{p}_3 = \mathbf{p}^\star$. Then either*

$$\|\nabla m_\mathbf{x}(\mathbf{p}_3)\| \leq \epsilon \tag{64}$$

*or*

$$f(\mathbf{x}^{(3)}) - f(\mathbf{x}) = m_\mathbf{x}(\mathbf{p}_3) - m_\mathbf{x}(\mathbf{0}) \leq -\frac{1}{2} r \|\nabla m_\mathbf{x}(\mathbf{p}_3)\|. \tag{65}$$

*In particular, in the non-stationary case Eq. (65) implies $m_\mathbf{x}(\mathbf{p}_3) - m_\mathbf{x}(\mathbf{0}) \leq -\Omega(r\epsilon) = -\Omega(\epsilon^{3/2})$.*

*Proof.* This is a direct corollary of Lemma E.2, taking $q = m_\mathbf{x}$, $b = \mathbf{g}$ and $A = \hat{\kappa}\hat{H}$. $\qquad\square$

### E.4. Complete proof of Theorem 6.1

*Proof of Theorem 6.1.* Assume $\|\mathbf{g}(x)\| \geq \epsilon$ (otherwise Algorithm 5 returns $\mathbf{x}$). Let $r = c\sqrt{\epsilon/L_2}$. Let $\mathbf{x}_{t+1} = \arg\min\{f(\mathbf{x}^{(1)}), f(\mathbf{x}^{(2)}), f(\mathbf{x}^{(3)})\}$ in Algorithm 5.

By Lemma E.1, for each candidate $\mathbf{p}_i = \mathbf{x}^{(i)} - \mathbf{x}$ with $\|\mathbf{p}_i\| \leq r$,

$$f(\mathbf{x}^{(i)}) - f(\mathbf{x}) \leq \left( T_\mathbf{x}(\mathbf{x} + \mathbf{p}_i) - T_\mathbf{x}(\mathbf{x}) \right) + \frac{L_2}{3} r^3. \tag{66}$$

We show that at least one candidate has $T_\mathbf{x}(\mathbf{x}+\mathbf{p}_i) - T_\mathbf{x}(\mathbf{x}) \leq -c'r\epsilon$ for a constant $c' > 0$; then since $\frac{L_2}{3}r^3 = \frac{1}{3}c^3\epsilon^{3/2}/\sqrt{L_2}$, for $c$ sufficiently small, we obtain $f(\mathbf{x}^{(i)}) - f(\mathbf{x}) \leq -\Omega(\epsilon^{3/2})$ and thus the same for $\mathbf{x}_{t+1}$.

If the Hessian is rank-one or colinear, Candidate 2 ensures the same order decrease by Lemma E.3.

Else, for cases in Lemma E.4,

$$T_\mathbf{x}(\mathbf{x} + \mathbf{p}_1) - T_\mathbf{x}(\mathbf{x}) \leq -\frac{99}{100} r \|\mathbf{g}\| + \frac{1}{2} r^2 \|H\|.$$

Using the ratio $\|H\|/\|\mathbf{g}\| \approx \hat{\kappa}$, we have $\frac{1}{2}r^2\|H\| \leq c''r\|\mathbf{g}\| \cdot r\hat{\kappa} \leq c'''r\|\mathbf{g}\|$ at the scale $r = c\sqrt{\epsilon/L_2}$ with $c$ sufficiently small, so the linear term dominates for small $\epsilon$. Hence $T_\mathbf{x}(\mathbf{x} + \mathbf{p}_1) - T_\mathbf{x}(\mathbf{x}) \leq -\Omega(r\|\mathbf{g}\|) \leq -\Omega(r\epsilon) = -\Omega(\epsilon^{3/2})$. Then Eq. (66) yields $f(\mathbf{x}^{(1)}) - f(\mathbf{x}) \leq -\Omega(\epsilon^{3/2})$.

In the non-degenerate case: If Eq. (64) fails, then by Lemma E.5, $m_\mathbf{x}(\mathbf{p}_3) - m_\mathbf{x}(\mathbf{0}) \leq -\frac{1}{2}r\|\nabla m_\mathbf{x}(\mathbf{p}_3)\|$. Since failure of Eq. (64) means $\|\nabla m_\mathbf{x}(\mathbf{p}_3)\| > \epsilon$, we get $m_\mathbf{x}(\mathbf{p}_3) - m_\mathbf{x}(\mathbf{0}) \leq -\frac{1}{2}r\epsilon = -\Omega(\epsilon^{3/2})$. By the constant-factor accuracy of $\hat{\kappa}$ and $\hat{H}$, the model $m_\mathbf{x}$ is a constant-factor approximation of the quadratic model $T_\mathbf{x}$ on $\|\mathbf{p}\| \leq r$ (we omit this routine norm-equivalence step; it only changes constants). Thus $T_\mathbf{x}(\mathbf{x} + \mathbf{p}_3) - T_\mathbf{x}(\mathbf{x}) \leq -\Omega(\epsilon^{3/2})$ and Eq. (66) gives $f(\mathbf{x}_3) - f(\mathbf{x}) \leq -\Omega(\epsilon^{3/2})$.

In all cases, at least one candidate $x^{(i)}$ satisfies $f(\mathbf{x}^{(i)}) \leq f(\mathbf{x}) - C\epsilon^{3/2}$. Since $\mathbf{x}_{t+1} = \arg\min\{f(\mathbf{x}^{(1)}), f(\mathbf{x}^{(2)}), f(\mathbf{x}^{(3)})\}$, we have $f(\mathbf{x}_{t+1}) \leq f(\mathbf{x}^{(i)})$ for that index $i$, proving Eq. (8). $\qquad\square$

### E.5. Discussion about Corollary 6.3

In this subsection, we naturally promote Algorithm 5 so we can go further to visit a $(\epsilon, \sqrt{\epsilon})$-second order stationary point.

Without using comparison oracle, we can get an estimation of the negative curvature direction $\mathbf{v}_{\min}$ from normalized Hessian $H/\|H\|$, satisfying $H\mathbf{v}_{\min} = \lambda_{\min}\mathbf{v}_{\min}$. When $\|\mathbf{g}\| \leq \epsilon$ and $\lambda_{\min} \leq -\sqrt{L_2\epsilon}$, the current point is nearly a saddle point. Step along the negative curvature direction gives a $\Omega(\epsilon^{3/2})$ descent in sphere with radius $r = \Theta(\sqrt{\epsilon})$.

**Lemma E.6.** *Fix a point $\mathbf{x} \in \mathbb{R}^n$ and let $\mathbf{g} = \nabla f(\mathbf{x})$ and $H = \nabla^2 f(\mathbf{x})$. Suppose that $\mathbf{v}_{min}$ is the unit eigenvector corresponding to the minimum eigenvalue of $H$, denoted as $\lambda_{min}$. If $\langle \mathbf{v}_{min}, \mathbf{g} \rangle \geq 0$, set $\mathbf{v}_{min} = -\mathbf{v}_{min}$. Let $\mathbf{x}^{(4)} = \mathbf{x} + r\mathbf{v}_{min}$, where $r = c\sqrt{\epsilon/L_2}$. If $\|\mathbf{g}\| \leq \epsilon$ and $\lambda_{min} \leq -\sqrt{L_2\epsilon}$, then*

$$f(\mathbf{x}^{(4)}) - f(\mathbf{x}) \leq -c_4\epsilon^{3/2} \tag{67}$$

*for an absolute constant $c_4 > 0$.*

*Proof.* We use the cubic Taylor remainder bound for any $\mathbf{p} \in \mathbb{R}^n$,

$$f(\mathbf{x} + \mathbf{p}) \leq f(\mathbf{x}) + \mathbf{g}^\top \mathbf{p} + \frac{1}{2}\mathbf{p}^\top H \mathbf{p} + \frac{L_2}{6}\|\mathbf{p}\|^3.$$

Plugging in $\mathbf{p}_4 = r\mathbf{v}_{\min}$ we have

$$\begin{aligned}
f(\mathbf{x}^{(4)}) &\leq f(\mathbf{x}) + r\mathbf{g}^\top \mathbf{v}_{\min} + \frac{1}{2}r^2 \mathbf{v}_{\min}^\top H \mathbf{v}_{\min} + \frac{L_2}{6}r^3 \\
&= f(\mathbf{x}) + c\sqrt{\frac{\epsilon}{L_2}}\mathbf{g}^\top \mathbf{v}_{\min} + \frac{1}{2}c^2 \frac{\epsilon}{L_2} \mathbf{v}_{\min}^\top H \mathbf{v}_{\min} + \frac{L_2}{6} \cdot c^3 \frac{\epsilon^{3/2}}{L_2^{3/2}} \\
&\leq f(\mathbf{x}) + c\sqrt{\frac{\epsilon^3}{L_2}}(-\frac{1}{2}c + \frac{1}{6}c^2)
\end{aligned}$$

Obviously we can choose $c = 1$ to obtain that there exists a constant $c_4 = 1/3\sqrt{L_2}$, satisfying

$$f(\mathbf{x}^{(4)}) - f(\mathbf{x}) \leq -c_4 \epsilon^{3/2}. \tag{68}$$

$\square$

With 4 candidates $\mathbf{x}^{(1)}, \mathbf{x}^{(2)}, \mathbf{x}^{(3)}, \mathbf{x}^{(4)}$, we can use a technique similar to Appendix E.4 to prove that we can find a $(\epsilon, \sqrt{\epsilon})$-second order stationary point in same query complexity $\tilde{O}(\frac{\Delta\sqrt{L_2}n^2}{\epsilon^{1.5}})$ as in Theorem 6.1.

*Proof of Corollary 6.3.* For any point $\mathbf{x} \in \mathbb{R}^n$, if $\|\nabla f(\mathbf{x})\| \geq \epsilon$, according to Theorem 6.1, we know there exists a constant $c_1$ such that
$$\min f(\mathbf{x}^{(1)}), f(\mathbf{x}^{(2)}), f(\mathbf{x}^{(3)}) \leq f(\mathbf{x}) - c_1 \epsilon^{3/2}.$$

Else if $\|\nabla f(\mathbf{x})\| \leq \epsilon$ and $\|\nabla^2 f(\mathbf{x})\| \leq -\sqrt{L_2 \epsilon}$, according to Lemma E.6, there exists a constant $c_2$ such that

$$f(\mathbf{x}^{(4)}) \leq f(\mathbf{x}) - c_2 \epsilon^{3/2}.$$

Combining these two fact, we have

$$\min\{f(\mathbf{x}^{(1)}), f(\mathbf{x}^{(2)}), f(\mathbf{x}^{(3)}), f(\mathbf{x}^{(4)})\} \leq f(\mathbf{x}) - \min\{c_1, c_2\}\epsilon^{3/2}.$$

For any step $\mathbf{x}_t$, choose $\mathbf{x}_{t+1} = \min f(\mathbf{x}^{(1)}), f(\mathbf{x}^{(2)}), f(\mathbf{x}^{(3)}), f(\mathbf{x}^{(4)})$ we have a $\Omega(\epsilon^{3/2})$ descent, which tells us that after at most $O(\frac{\Delta}{\epsilon^{3/2}})$ iterations we can find a $(\epsilon, \sqrt{L_2 \epsilon})$-second order stationary point. The total query complexity is $O(\frac{\Delta\sqrt{L_2}n^2}{\epsilon^{1.5}} \log\left(\frac{nL_1 L_2}{\epsilon}\right))$. $\square$

