# OpenReview forum: "Finding Stationary Points by Comparisons"
_ICML.cc/2026/Conference — ICML 2026 regular_

### Official Review · Reviewer_occa · 2026-02-21

**Soundness:** 3
**Presentation:** 2
**Significance:** 3
**Originality:** 3
**Overall Recommendation:** 5
**Confidence:** 3

**Summary:**

This paper considers the problem of finding approximate stationary points in high-dimensional spaces under non-convex, but $L_2$-Hessian Lipschitz, objective functions $f$. They study algorithms for this problem that only get pairwise comparison queries: given $x,y\in \mathbb{R}^n$, the oracle outputs whether $f(x) > f(y)$. The main result is a classical algorithm that makes roughly $\Delta L_2 n^2/\epsilon^{1.5}$ queries to the oracle and has the guarantee that one of the queries is an $\epsilon$-stationary point. This is under the assumption that the initial given query point has value at most $\Delta$ away from the true optimum. Note that this problem is invariant under scaling $f$, and this shows up in the query complexity because scaling will cancel.

There is also a quantum version of the algorithm that quadratically improves the dependence on $n$.

Previous works used standard Lipschitz bounds on $f$ to achieve a quadratic dependence on the ratio between $\epsilon$ and the Lipschitz/global closeness parameters. This paper adds the stronger hypothesis of $L_2$-Hessian Lipschitzness and improves this bound on the accuracy ratio.

The main technical contribution of this paper is a way to estimate the principal direction of the Hessian of the function $f$ around a given point using comparison queries. The way they do this is essentially a reduction to estimating the gradient directions of a function using comparisons by applying that algorithm to different points forming a triangle around the point of interest.

**Compliance With Llm Reviewing Policy:**

Affirmed.

**Final Justification:**

I will keep my positive evaluation given that the authors fully resolved my questions in the rebuttal.

**Key Questions For Authors:**

Do you know of any lower bounds on the query complexity?

What additional assumptions do you need to verify approximate stationary points?

Should it be easy to extend your work to higher-order information (beyond Hessians)?

**Limitations:**

yes

**Strengths And Weaknesses:**

Strengths: The paper is sound and studies a natural problem that captures many other important computational problems. The trick of estimating the Hessian directions is nice and seems to be technically novel. The additional assumption on the Lipschitzness of the Hessian is natural.

Weaknesses: Much of the paper is very technical, and the technical overview seems a bit lacking in terms of contextualizing the new insights in terms of the design of the previous algorithms on this problem. The technical content is quite dense and could use some exposition in the middle. Also, the inability to even verify fixed points is somewhat strange.

---

> ### Author Rebuttal · Authors · 2026-03-31
>
> Thank you for your positive feedback and detailed comments!
>
> Thank you for your question on query complexity lower bounds. We would like to kindly note that dimension-dependent lower bounds for stationary point computation are not completely understood, even in the gradient oracle and Hessian oracle model. For example, [1] shows that even with gradient and Hessian oracles, at least $\Omega(\epsilon^{-1.5})$ queries are required to find an $\epsilon$-stationary point, but only when $n \ge \Omega(\epsilon^{-1.5})$. The $\epsilon$-dependence of our algorithm matches this lower bound. However, it is unclear if this dependence is tight when $n=o(\epsilon^{-1.5})$. As for dependency on $n$, at least $\Omega(n)$ queries are needed. Indeed, finding an $\epsilon$-stationary point is no easier than estimating the normalized gradient direction at a given point, which in turn requires $\Omega(n)$ function value queries to achieve even constant accuracy in the worst case.
>
> Thank you for your great question on how to verify stationarity. In the comparison setting, the algorithm only have access to relative ordering of function values across queried points. Consequently, it cannot access any magnitude information about derivatives at any order, including the function value, gradient, or Hessian. Hence, an additional assumption that the algorithm has certain access to the magnitude information is needed in the worst case.
> Nevertheless, we note that for many machine learning problems of interest, stationarity is closely aligned with global optimality. For example, in tensor decomposition [3], matrix completion [4], and regression with nonconvex regularization [5], it has been shown that global optimality can be obtained by finding a second-order stationary point, which is a generalization of stationary points. In these settings, the iterate in our algorithm with the smallest function value would be is close to optimal and therefore approximately stationary with high probability. We will add further discussion on this in the final version.
>
> Thank you for the insightful question on extending our technique to higher-order methods. A natural idea would be to develop an algorithm that (i) estimates higher-order normalized derivatives, (ii) estimates the relative scaling between the gradient and these higher-order derivatives, and (iii) performs an approximate higher-order trust-region method whose update can be computed from these normalized derivatives together with their relative scalings. Theoretically, we believe this approach is feasible and would achieve query complexity $\tilde O(n^p\epsilon^{-\frac{1+p}{p}})$ if the function $f$ has $O(1)$-Lipschitz $p$-th order derivatives. This would match the $\epsilon$-dependence of optimal $p$-th order methods up to polylogarithmic factors [1].
> Nevertheless, this introduces additional technical challenges. For example, our current procedure for estimating the normalized Hessian already requires careful handling of certain corner cases, particularly when the Hessian is ill-conditioned. Moving to normalized third-order derivatives is more delicate: the associated tensor structure can introduce further degeneracies, and controlling the resulting errors would require a more refined analysis. We will include a more detailed discussion on this in the final version.
>
> Thank you for your comments on the presentation. We will include more qualitative explanations of the algorithms and proofs in the final version to improve clarity.
>
> [1] Yair Carmon, John C. Duchi, Oliver Hinder, and Aaron Sidford. Lower bounds for finding stationary points I.
>
> [2] El Houcine Bergou, Eduard Gorbunov, Peter Richtárik. Stochastic Three Points Method for Unconstrained Smooth Minimization.
>
> [3] Rong Ge and Tengyu Ma. Decomposing overcomplete tensors using sum-of-squares algorithms.
>
> [4] Rong Ge, Furong Huang, Chi Jin, and Yang Yuan. Escaping from saddle points—online stochastic gradient for tensor decomposition.
>
> [5] Po-Ling Loh and Martin J. Wainwright. Regularized M-estimators with nonconvexity: Statistical and algorithmic theory for local optima.

---

> > ### Author Rebuttal · Reviewer_occa · 2026-03-31
> >
> > I thank the authors for their well thought out rebuttal.

---

> > > ### Author Response · Authors · 2026-04-01
> > >
> > > Thank you so much for your very positive feedback. We are delighted that our response fully resolved your concerns.

---

### Official Review · Reviewer_zvyp · 2026-02-24

**Soundness:** 3
**Presentation:** 3
**Significance:** 3
**Originality:** 4
**Overall Recommendation:** 5
**Confidence:** 3

**Summary:**

This paper studies nonconvex optimization when the algorithm cannot access function values or gradients, and instead only has a comparison oracle that tells which of two points has larger objective value. Under standard smoothness assumptions (Lipschitz gradient and Hessian), the authors develop a comparison-based method (Algorithm 5) that is guaranteed to visit an ε-stationary point with query complexity O(Δsqrt(L2​​)n^2/ε^1.5), improving the ε-dependence over prior comparison-based nonconvex methods. The technical core is a sequence of geometric subroutines that estimate gradient directions and a normalized Hessian (via a Hessian-vector direction estimator), which then enable a progress guarantee of Ω(𝜀^3/2) decrease per iteration unless an ε-stationary point has already been encountered. The paper also extends the framework to a quantum comparison oracle model and gives (to the best of my knowledge) the first quantum algorithm in this setting for visiting an ε-stationary point.

**Compliance With Llm Reviewing Policy:**

Affirmed.

**Final Justification:**

Overall, the paper delivers a technically original and meaningful advance in comparison-based nonconvex optimization, and these strengths outweigh the limitations in output guarantees and dimension dependence.

**Key Questions For Authors:**

1. Does the main guarantee assume deterministic/noiseless comparisons, or is there a formal tolerance model under which the bounds still hold? If robustness requires additional assumptions, please summarize them clearly and indicate how the complexity changes.

2. Is the O(e^-1.5) dependence information-theoretically tight for comparison oracles under the stated smoothness assumptions, or is there still a gap to known lower bounds?

3. Can you clarify whether the n^2 dependence is inherent to estimating a normalized Hessian via comparisons, or might it be reduced via sketched Hessian structure without degrading ε-dependence?

4. Since the algorithm can only guarantee it visited an ε-stationary point, do you have a principled strategy or heuristic to pick a single returned point that empirically correlates with stationarity in practical problems?

5. For the quantum comparison oracle result, can you provide additional intuition on when such an oracle is plausible, and whether the stated quantum improvement is optimal relative to existing quantum lower bounds in related models?

**Limitations:**

Not fully. I suggest adding a short limitations/impact paragraph explicitly noting that the algorithm can only output a list and cannot certify which point is 𝜀 ε-stationary in the comparison model, the n^2 dependence may be prohibitive in high dimensions, and assumptions (twice differentiability, Lipschitz Hessian, bounded initial gap) restrict applicability. Societal impact appears limited given the theoretical nature, but the paper could briefly mention potential connections to preference-based decision systems.

**Strengths And Weaknesses:**

Strengths

The main claims are stated under clear smoothness assumptions (Lipschitz gradient and Hessian) and are backed by theorem statements with explicit query complexity.

The proof strategy is coherent and shows a per-iteration expected decrease unless stationarity was visited, and then argue by contradiction using a bounded function gap, yielding the stated iteration and query bounds and a success probability. The algorithm design is careful about the comparison-only restriction by selecting among candidate points by comparisons.

The introduction positions the work relative to prior comparison-oracle results in both convex and nonconvex settings, and clearly states what improves (the ε rate) and what worsens (dimension dependence). The paper provides a roadmap of techniques  starting from gradient-direction estimation to Hessian direction estimation to normalized Hessian estimation to stationary-point procedure.

The comparison-oracle model is relevant to dueling optimization and some RLHF-style formulations where only rankings are observable. The improved ε-dependence to match second-order rates is a meaningful theoretical advance. That said, the setting is specialized and the “list output” nature reduces immediate applicability, so the impact is strongest for theory/foundations.

The paper’s main originality is the higher-order comparison-based pathway to improved stationarity rates, enabled by a novel way to estimate Hessian directional information using only comparisons, and then using that inside a stationary-point routine. The quantum comparison-oracle extension is also a distinctive contribution.

Weaknesses

The guarantee is that the method visits an ε-stationary point during its trajectory and outputs a list of candidates, because in the comparison model one cannot generally test ∥∇𝑓(𝑥)∥ or certify stationarity at a given point.

This is a real limitation for downstream use. Also, the improved ε-dependence comes with an n^2 dimension dependence vs. some prior n-linear dependencies, which may limit practicality in high dimensions.

The analysis appears technically dense with many parameters and multi-branch arguments. A short parameter glossary and a single table summarizing oracle calls per subroutine would improve readability and reproducibility. It would also help to more explicitly separate what is information theoretically necessary in the comparison model with what is an artifact of the current construction.

---

> ### Author Rebuttal · Authors · 2026-03-31
>
> Thank you for your positive feedback and detailed comments!
>
> Thank you for your question on our assumption about basic oracle setting. The assumption used in our paper is a deterministic noiseless comparison oracle, i.e. given two points $x$ and $y$, the oracle $O_f^{\text{comp}}$ outputs 1 iff $f(x)\ge f(y)$ and $-1$ otherwise deterministically. Our methods naturally extends to the case where $O_f^{\text{comp}}$ is stochastic and outputs the correct answer with success probability $2/3$ via majority vote by introducing an additional logarithmic factor. If you have a particular setting or noise model in mind, we would be very interested in discussing it further during the discussion period.
>
> Regarding your question on whether the $O(\epsilon^{-1.5})$ dependence is tight, we think it is a great question. We would like to kindly note that dimension-dependent lower bounds for stationary point computation are not completely understood, even in the gradient oracle and Hessian oracle model. For example, [1] shows that even with gradient and Hessian oracles, at least $\Omega(\epsilon^{-1.5})$ queries are required to find an $\epsilon$-stationary point, but only when $n \ge \Omega(\epsilon^{-1.5})$. The $\epsilon$-dependence of our algorithm matches this lower bound. However, it is unclear if this dependence is tight when $n=o(\epsilon^{-1.5})$. This lower bound also has not ruled out the possibility that one can reduce the dependence on $\epsilon$ by increasing the dependence on the dimension $n$. We view the development of lower bounds in the comparison model as a very interesting future direction, and will include a more detailed discussion of lower bounds in our final version.
>
> Thank you for the great question on the $n^2$ dependence. A better dependence on $n$ can be achieved by using first-order optimization frameworks, at the cost of a worse dependence on $\epsilon$. For example, by combining the normalized gradient method with the comparison-based gradient direction estimation algorithm in [2], we can obtain a method that finds an $\epsilon$-stationary point using $O(n/\epsilon^2)$ gradient queries. The same bound is also achieved in [3]. We will include a discussion of this $n$–$\epsilon$ tradeoff in the final version.
>
> Regarding your question on how to pick a single point from the list, we completely agree with you this is an important question, especially for downstream use. In many machine learning problems of interest, stationarity is closely aligned with global optimality. For example, in tensor decomposition [4], matrix completion [5], and regression with nonconvex regularization [6], it has been shown that global optimality can be obtained by finding a second-order stationary point, which is a generalization of stationary points. In these settings, a natural and principled heuristic strategy is to return the iterate with the smallest function value (that is observed by comparisons between them), which is close to optimal and therefore approximately stationary. We will add further discussion on this in the final version.
>
> Thank you for your question on the quantum oracle setting and the optimality of our quantum improvement. The quantum comparison oracle we consider is a natural quantum generalization of the classical comparison oracle, and is plausible in settings where the classical comparison oracle itself is efficiently computable by a classical circuit, in which case there exists a quantum circuit of the same size that gives the quantum comparison oracle.
>
> Regarding optimality, similar to the classical case, the quantum lower bounds for stationary points are not completely understood. For example, [1] shows that even with quantum gradient and Hessian oracles, at least $\Omega(\epsilon^{-1.5})$ queries are required to find an $\epsilon$-stationary point, but only when $n \ge \tilde{\Omega}(\epsilon^{-1.5})$. The $\epsilon$-dependence of our algorithm matches this lower bound. However, it is unclear if this dependence is tight when $n=o(\epsilon^{-1.5})$. This lower bound also has not ruled out the possibility that one can reduce the dependence on $\epsilon$ by increasing the dependence on the dimension $n$. We will add more discussion on the quantum oracle setup as well as relevant quantum lower bounds in the final version.
>
> [1] Carmon et al. Lower bounds for finding stationary points I.
>
> [2] Tao et al. Gradient Testing and Estimation by Comparisons.
>
> [3] Bergou et al. Stochastic Three Points Method for Unconstrained Smooth Minimization
>
> [4] Ge and Ma. Decomposing overcomplete tensors using sum-of-squares algorithms.
>
> [5] Ge et al. Escaping from saddle points—online stochastic gradient for tensor decomposition.
>
> [6] Loh and Wainwright. Regularized M-estimators with nonconvexity: Statistical and algorithmic theory for local optima.
>
> [7] Chenyi Zhang and Tongyang Li. Quantum Lower Bounds for Finding Stationary Points of Nonconvex Functions

---

> > ### Author Rebuttal · Reviewer_zvyp · 2026-03-31
> >
> > Thank you for answering all of our questions! We believe that our rating of a 5 is accurate.

---

> > > ### Author Response · Authors · 2026-04-01
> > >
> > > Thank you so much for your very positive feedback. We are delighted that our response fully resolved your concerns.

---

### Official Review · Reviewer_hw24 · 2026-03-04

**Soundness:** 4
**Presentation:** 3
**Significance:** 3
**Originality:** 3
**Overall Recommendation:** 5
**Confidence:** 4

**Summary:**

This paper investigates the task of visiting $\epsilon$-stationary points of non-convex functions via a comparison oracle. Indeed, since it is only assumed that the algorithms can use the comparison oracle, the algorithm cannot even test whether a point is a stationary point, let alone finding it. Hence, the guarantee that it visits an $\epsilon$-stationary point.

The main contribution of this work is an algorithm that, only via $\tilde O(n^2/\epsilon^{3/2})$ many comparison queries, it is able to visit an $\epsilon$-stationary point with constant probability. This result is obtained through the following points:

- First, they introduce a directional estimation algorithm ComparisonHessVec to approximate the direction of Hessian-vector products by comparing gradient directions at perturbed points ($x \pm ry$).

- Second, they provide a Hessian reconstruction algorithm ComparisonHE that recovers the normalized Hessian matrix by estimating individual column directions and determining their relative magnitudes through a "triangle-solving" geometric approach and using the ComparisonHessVec subroutine.

- Third, for the norm ratio calculation, their algorithm ComparisonRatio identifies the principal eigenvector up to an approximation error. The Davis–Kahan theorem is then used to bound estimation errors, allowing the algorithm to calculate the ratio between the Hessian and gradient norms.

- Finally, all these components integrate into a trust-region algorithm that guarantees a function decrease of $\Omega(\epsilon^{3/2})$ per step, ensuring the identification of an $\epsilon$-stationary point within $O(1/\epsilon^{3/2})$ iterations.

It is worth noting that the authors also extend these results to a setting where the access it to a quantum oracle where queries are issued in superposition. This allows bringing the dependence of number of queries on dimensionality from quadratic to linear, and all other dependencies (asymptotically) unaltered, that is $\tilde O(n/\epsilon^{3/2})$ many quantum comparison queries.

**Compliance With Llm Reviewing Policy:**

Affirmed.

**Final Justification:**

As explained in the rebuttal acknowledgment, the authors have provided very good and complete responses, and I decided to raise my score to 5.

**Key Questions For Authors:**

- Regarding the claim that ComparisonGE has an $O(n)$ overhead ($n$ comparisons needed to explore $n$-dimensional space), could the dependence drop under low-dimensional structure? Indeed, suppose there exists a $k$-dimensional subspace $U$ such that (approximately) $\nabla f(x)\in U$. If $U$ were known, we could restrict query directions in ComparisonGE to $U$, suggesting the direction-estimation cost might scale as $\tilde O(k)$ rather than $\tilde O(n)$.

- Also, would the above be any useful if the dominant curvature directions do not lie in the same $U$?

- Relatedly, suppose we aim for a weaker goal: finding a descent direction with constant probability rather than estimating the normalized gradient accurately. For instance, we could compare $f(x+ru)$ and $f(x-ru)$ for a random unit $u$ and stepping toward the better point. Would such a relaxation avoid the $\Omega(n)$ limitation? I would suspect not and this is probably related to the STP algorithm and its query complexity, but I am not sure.

**Limitations:**

yes

**Strengths And Weaknesses:**

Overall, I found the paper technically solid and enjoyable to read. The results build in a clean, modular way from the more restricted gradient-testing/estimation setting of Tao et al. (2026) to the more general comparison-only model for stationary points visiting. It is nice that you highlight how you are concretely using their ideas, and arriving to the results requires a number of clever steps.

Specifically, one part I particularly liked is Algorithm 5 and its case split: constructing three candidate moves (a normalized gradient step, a trust-region step on a normalized quadratic model, and a rank-one line-search step) and selecting the best is a neat way to dissect the problem and guarantee an iterate function decrease of $\Omega(\epsilon^{3/2})$.

More broadly, I found the main result genuinely surprising: despite having access only to comparative information, the algorithm can still guarantee visiting an $\epsilon$-stationary point with nontrivial probability and an almost tight complexity guarantee (see Questions).

There are also a few smaller clarity and consistency issues worth addressing (please let me know if I have missed something here):
- when the text says it “solves the triangle,” it would help to be explicit earlier on, e.g., that the law of sines is used and which directions/angles are known versus which norm ratios are being recovered.
- for completeness, I would also suggest adding pseudocode for ComparisonGE in the appendix, because, as written, I had to consult Tao et al. (2026) to fully reconstruct the routine (even though the overarching principle is clear already from your text).
- in Algorithm 1, line 3, gradients/vectors should be formatted consistently (boldface);
- the “Without loss of generality, …” paragraph in Section 4 left me slightly unsure why the testing step is needed (since Algorithm 2 does not appear to use which $||He_i||$ is largest), so it would be helpful to clarify whether this is only for analysis or is required for correctness;
- line 230 likely intends “is relatively small”;
- it is unclear what the $\xi_2$ tolerance is used for in Algorithm 5;
- finally, ComparisonGE is called with three arguments in Algorithm 1 but later with two (Algorithms 4 and 5)—if the lower bound $\gamma$ is implicit in later calls, that should be clarified.

---

> ### Author Rebuttal · Authors · 2026-03-31
>
> Thank you for your positive feedback and detailed comments!
>
> Thank you for the great question on reducing the overhead under low-dimensional structures. As you pointed out, if the subspace $U$ is known, the cost could indeed be $\tilde O(k)$. However, when $U$ is an unknown arbitrary subspace with no additional structure, $\Omega(n)$ comparison queries would still be needed to even estimate the normalized gradient of a single $x$. That said, under additional assumptions on $U$ (e.g., arising from structure in $f$), it is very possible to exploit this low-dimensionality algorithmically. We believe this is a natural direction for future work. If you have a particular setting or assumption in mind, we would be very interested in discussing it further during the discussion period.
>
> Regarding your question on the possibility to reduce the $n$ dependence by finding a descent direction with constant probability, we think it is a brilliant idea. The method you mentioned is exactly the updating formula in the STP algorithm [1]. Formally, [1] shows that if we set random unit $u$ as uniformly distributed on the unit sphere, the query complexity for finding a first-order stationary point is $O(n/\epsilon^2)$ (Algorithm 3.1 and Theorem 4.1). We believe this $\Omega(n)$ dependence is intrinsic in the worst case. Here is a brief proof sketch. Finding an $\epsilon$-stationary point is no easier than estimating the normalized gradient direction at a given point. For instance, consider the function $\langle g, x\rangle + ||x||^2/2$ for some unit vector $g$. Finding an $\epsilon$-stationary point of this function would recover an $\epsilon$-approximation $g$, which in turn requires $\Omega(n)$ queries even when $\epsilon$ is a constant and we have a function value oracle instead of a comparison oracle.
>
> Regarding your comment on the clarity and consistency issues, we greatly appreciate you for pointing them out, and we will fix them in the final version. Particularly, please find our clarifications regarding entries (1), (4), (6), and (7) below.
>
> (1). By “solve the triangle” (Line 337), we mean the following geometric procedure illustrated in Figure~3: We estimate the directions of $\nabla f(x)$ and $\nabla f(x+\mu v)$, denoted by $AB$ and $AC$, respectively. We also take $v$ to be the top eigenvector of our estimate $\widetilde H$, and use it to approximate the true leading eigenvector $u$ of $\nabla^2 f(x)$; the approximation error $||u - v||$ is controlled via the Davis–Kahan theorem. This allows us to approximate the direction of $\nabla^2 f(x)\cdot u$, which corresponds to the direction of $BE$. Next, we use $BE$ to approximate $BC$, since $\nabla f(x+\mu v) - \nabla f(x) = \nabla^2 f(x)\cdot u + o(\mu)$. By Lemma D.4, this approximation is sufficiently accurate. Given the directions of $AB$, $AC$, and $BC$, we can apply the law of sines to recover the ratio $||AB||/||BC||$, which corresponds (up to a constant factor) to the gradient–Hessian ratio. We will clarify this argument in the final version.
>
> (4). In the setting of Theorem 4.1, we assume that $||H||\ge\sqrt{L_2\epsilon}$, which tells us that there is an index $i$ such that $||He_i||$ is large (relative to $\epsilon$). Because in line 22 and 23 of Algorithm 2, in some corner cases, if the norm of some column vectors of the Hessian is too small to influence the distance between Hessian estimation and the real Hessian, we set it as $\vec0$. In this case, if the norm $||He_1||$ is too small, our estimation $\widehat{r_i}\sim||He_i||/||He_1||\triangleq 0$ may has uncontrollable error. So the existence of the largest norm column vector is required for correctness, the index is only for the convenience of writing.
>
> (6). $\xi_1$ and $\xi_2$ play essentially the same role, as both quantify the accuracy of gradient estimation. This is already noted in Appendix D.2 (Lines 1434–1435), where we treat them as analogous and use $\xi_1$ for notational simplicity. Since the overall error is controlled by the worse of the two, we set $\xi_1=\xi_2$ in Theorem 6.1.
>
> (7).  In Section 3, 4 and 5, we focus on the case that $\\|g\\|\ge\epsilon$. Hence, $\gamma$ should be set to $O(\epsilon)$.
>
>
>
> [1] El Houcine Bergou, Eduard Gorbunov, Peter Richtárik. Stochastic Three Points Method for Unconstrained Smooth Minimization.

---

> > ### Author Rebuttal · Reviewer_hw24 · 2026-04-02
> >
> > I thank the authors for the very good and complete responses, which I surely agree with. I will raise my score to 5.
> >
> > Regarding the possibility of improvement for "sparse" functions, I was thinking of something simple like $f(x) = g(Wx)$ where $W \in \mathbb R^{k \times n}$ is a projection matrix. Then, $\nabla f(x) = W^\top \nabla g(Wx)$ and the gradient would lie in $k$-dimensional row space of $W$, if I am not wrong. So, if one can learn the span of $W$ then it would be possible to solve the problem in $\mathbb R^k$ right? But there may be other examples, like low-rank matrix factorization, etc.
> >
> > I was also thinking that (but this is not properly thought through) in these cases you could first apply Johnson-Lindenstrauss lemma, i.e., define a random Gaussian matrix R \in $\mathbb R^{n \times m}$, with $m = O(k\log k)$ , and compare $f(x + Rz_1)$ to $f(x + Rz_2)$, for $z \in \mathbb R^m$. Now, because the random projection $R$ should preserve inner products and distances (by the JL lemma), the low-dimensional gradients behave closely to the real gradients. But maybe this does not work once the math is carried out.

---

> > > ### Author Response · Authors · 2026-04-03
> > >
> > > Thank you so much for your very positive feedback and for raising your recommendation to 5! We are delighted that our response resolved your previous questions and concerns.
> > >
> > > Thank you for your follow-up question on possible improvements for “sparse” functions. We agree that this is a very natural model for many machine learning problems and worth further investigation. We also agree that, once the span of $W$ has been identified, one can find an $\epsilon$-stationary point within this $k$-dimensional subspace with complexity independent of $n$, for example, $O(k^2/\epsilon^{1.5})$ using our algorithm or $O(k/\epsilon^2)$ using STP. However, we feel at least $\Omega(n)$ queries are needed to learn $W$ in the worst case. Consider the special case in which the underlying subspace has dimension $1$ and is spanned by a vector $w \in \mathbb{R}^n$. In this case, learning $w$ reduces to learning the gradient direction, which requires $\Omega(n)$ queries in the worst case even to achieve constant accuracy. Nevertheless, this approach may still lead to a total complexity of the form $O(n+\mathrm{poly}(k,\epsilon^{-1}))$, thereby decoupling the dependence on $n$ from that on $\epsilon$. We find this possibility very interesting and worth future investigation.
> > >
> > > We think it is a very interesting question whether the JL lemma could be useful in this setting. Could you please elaborate a bit more on the connection you have in mind between JL and the “sparse” function model? In particular, for a random projection $R$, we didn't completely follow why $Rz$ is in the row space of $W$. That said, we agree with your intuition that these low-dimensional gradients may closely track the true gradients. We would be very interested in discussing this idea further with you.

---

### Official Review · Reviewer_rMSt · 2026-03-11

**Soundness:** 3
**Presentation:** 3
**Significance:** 3
**Originality:** 3
**Overall Recommendation:** 5
**Confidence:** 3

**Summary:**

This paper studies finding stationary pointys by comparisons oracle. This paper provides an oracle complexity $\mathcal{O}(n^2\epsilon^{-1.5})$ to find the $\epsilon$-stationary point of a nonconvex function with Lipschitz continuous Hessian.
The author then provide quantum algorithms which take $\mathcal{O}(n\epsilon^{-1.5})$ quantum comparison oracle.

**Compliance With Llm Reviewing Policy:**

Affirmed.

**Final Justification:**

The authors have addressed my concern and I raise my score to 5.

**Key Questions For Authors:**

1. Although the oracle complexity enjoys optimal dependency on $\epsilon$, it seems that its dependncy on the dimension $n$ is not that good. Given the fact that the gradient, Hessian-vector, and the Hessian can be well estimated within $\mathcal{O}(n)$, $\mathcal{O}(n)$, and $\mathcal{O}(n^2)$ of the comparison oracle, it seems that the comparison oracle plays similar roles as the zeroth-order oracle. The authors may need to discuss [A, B] to check if their oracle complexity can be improved under both classical and quantum settings.

2. The paper only considers first-order stationary point. I am wondering if we can find second-order stationary point?

3. Please illustrate why we need to construct approximation to the normalized Hessian instead of the approximation of the Hessian.




[A]. First and zeroth-order implementations of the regularized Newton method with lazy approximated Hessians. Journal of Scientific Computing, 2025.

[B]. Quantum Speedups for Minimax Optimization and Beyond. NeurIPS, 2025.

**Limitations:**

Please refer to the question part.

**Strengths And Weaknesses:**

# Strengths:
* The method is very novel. This paper constructs several novel estimators of Hessian vector and normalized Hessian with only comparison oracles. These estimators are with high interests for the training of LLMs.
* The convergence results are strong. This paper shows that even with comparison oracles, one can achieve the optimal dependency on $\epsilon$ for finding the stationary point.
* This paper considers not only classical methods, but also improve its complexity when the quantum comparison oracle is achievable. Compared to the existing oracles used in quantum optimization, this quantum comparison oracle is cheaper and easier to realize.

# Weakness:
Please refer to the question parts.

---

> ### Author Rebuttal · Authors · 2026-03-31
>
> Thank you for your positive feedback and detailed comments!
>
> $\mathbf{Question\ 1}$: Although the oracle complexity enjoys optimal dependency on $\epsilon$, it seems that its dependncy on the dimension $n$ is not that good. Given the fact that the gradient, Hessian-vector, and the Hessian can be well estimated within $O(n)$, $O(n)$, and $O(n^2)$ of the comparison oracle, it seems that the comparison oracle plays similar roles as the zeroth-order oracle. The authors may need to discuss [A, B] to check if their oracle complexity can be improved under both classical and quantum settings.
>
> $\mathbf{Answer\ 1}$: Thank you for this great question. Indeed, following the methods and essentially the same underlying mechanism in [A,B], we have obtained a lazy trust-region method that finds an $\epsilon$-stationary point in $\widetilde O(n^{1/2}/\epsilon^{3/2})$ iterations. The method recomputes the normalized Hessian every $\widetilde\Theta(n)$ iterations. At each iterate $x_t$, the next point $x_{t+1}$ is computed using the normalized gradient at $x_t$ together with the gradient–Hessian norm ratio. A brief updating formula is as follows:
> $$
> x_{t+1}=\mathrm{min}_{||x-x_t||\le r}\{\langle\frac{g_t}{||g_t||}, x-x_t \rangle+\frac{||\tilde H||}{2||g_t||}(x-x_t)^\top\frac{\tilde H}{||\tilde H||}(x-x_t)\}
> $$
> where $g=\nabla f(x)$ and $\tilde H$ is the last computed Hessian. This lazy trust-region framework is well suited for comparison-based methods. Similar to the trust-region framework we used, it does not require any magnitude information of the function. Moreover, we have developed subroutines for estimating the normalized Hessian and gradient-Hessian ratio as Algorithm 2 and 4 in our paper. Nevertheless, these subroutines introduce errors, and careful treatment is needed for certain corner cases. For example, when estimating the gradient-Hessian ratio, any comparison based algorithm cannot handle the case where the Hessian has rank 1 and its only non-zero eigenvector is in the same direction of the gradient information-theoretically. Hence, a comparison-based algorithm does not follow directly from this framework.
>
> Nevertheless, we feel optimistic about this approach and achieving a $\widetilde O(n^2 + n^{3/2}/\epsilon^{3/2})$ query complexity, matching the zeroth-order query complexity of [A]. We will include a detailed discussion on this in our final version.
>
> $\mathbf{Question\ 2}$: The paper only considers first-order stationary point. I am wondering if we can find second-order stationary point?
>
> $\mathbf{Answer\ 2}$: Thank you for this great question. Our framework can indeed be extended to finding second-order stationary points (SOSP). In particular, our normalized Hessian estimation procedure already provides an approximate negative curvature direction. Building on this, we can augment Algorithm 6 with an additional candidate step that moves along the negative curvature direction with step size $r=\Theta(\epsilon)$. This step decreases the function value by $\Omega(\epsilon^{1.5})$ if current point is an $\epsilon$-FOSP but not an $(\epsilon,\sqrt\epsilon)$-SOSP. The analysis follows closely that of Appendix E.3. As a result, this modified algorithm finds an $(\epsilon,\sqrt\epsilon)$-SOSP using $\widetilde O(n^2/\epsilon^{1.5})$ queries. We will add this revised algorithm as well as the modified analysis in the final version.
>
> $\mathbf{Question\ 3}$: Please illustrate why we need to construct approximation to the normalized Hessian instead of the approximation of the Hessian.
>
> $\mathbf{Answer\ 3}$: Thank you for the insightful question. In the comparison setting, the algorithm only have access to relative ordering of function values across queried points. Consequently, it cannot access any magnitude information about derivatives at any order, including the function value, gradient, or Hessian. This necessitates working with a normalized Hessian, which depends only on scale-invariant directional information that remains accessible in this model. We will add more discussion on this in our final version.

---

> > ### Author Rebuttal · Reviewer_rMSt · 2026-04-01
> >
> > Thank you for answering all of my questions. I decide to raise my score to 5.

---

> > > ### Author Response · Authors · 2026-04-01
> > >
> > > Thank you so much for your very positive feedback and for raising your recommendation to 5. We are delighted that our response fully resolved your concerns.

---

### Decision · Program_Chairs · 2026-04-30

**Decision:**

Accept (regular)

**Comment:**

The paper considers the problem of finding stationary points in non-convex functions by comparing the values of two different points in the domain. The main result of the paper is an algorithm that finds a stationary point for an $n$-dimensional function using $O(n^2/\epsilon^2)$ comparisons. All reviewers and I recognize the novelty of the paper and the importance of the result. Thus, I believe it is a solid paper and should be considered for acceptance.